# Single-cell and spatial RNA sequencing reveal the spatiotemporal trajectories of fruit senescence

Xin Li [1,2,3], Bairu Li[1], Shaobin Gu[1], Xinyue Pang[4], Patrick Mason[2], Jiangfeng Yuan[1], Jingyu Jia[1], Jiaju Sun[1], Chunyan Zhao [5] ✉ & Robert Henry [2] ✉

The senescence of fruit is a complex physiological process, with various cell types within the pericarp, making it highly challenging to elucidate their individual roles in fruit senescence. In this study, a single-cell expression atlas of the pericarp of pitaya (*Hylocereus undatus*) is constructed, revealing exocarp and mesocarp cells undergoing the most significant changes during the fruit senescence process. Pseudotime analysis establishes cellular differentiation and gene expression trajectories during senescence. Early-stage oxidative stress imbalance is followed by the activation of resistance in exocarp cells, subsequently senescence-associated proteins accumulate in the mesocarp cells at late-stage senescence. The central role of the early response factor HuCMB1 is unveiled in the senescence regulatory network. This study provides a spatiotemporal perspective for a deeper understanding of the dynamic senescence process in plants.

Fruit, as a unique developmental organ in flowering plants, plays a crucial role in seed formation and plant reproduction. The growth and development process of fruit is generally divided into five stages: cell differentiation, cell enlargement, fruit development, ripening, and senescence[1]. Senescence, as the final critical stage in the fruit's life cycle, directly impacts the maintenance of fruit quality, as well as its market value and post-harvest lifespan. Fruit senescence is a complex oxidative and physiological process, accompanied by the metabolism of numerous substances[2]. In recent years, various theories have been proposed to explain the occurrence and development of senescence[2,3]. However, to date, the molecular mechanisms of senescence, especially the development of senescence, remain far from fully elucidated.

In recent years, rapidly advancing transcriptome sequencing technology has allowed the generation of large-scale gene expression data, enabling researchers to explore complex physiological processes within organisms based on spatiotemporal gene expression differences. Fruit is composed of different types of cells, but conventional bulk transcriptome sequencing can only detect the average gene expression levels[4,5], which obscures the characteristics of different cell populations. Consequently, it is not possible to analyze the cellular diversity and transcriptomic state heterogeneity within fruit tissues, which may be a significant reason why the overall patterns in the process of fruit senescence remain unclear.

The recent advancements in single-cell RNA sequencing (scRNA-seq) technology have provided an opportunity to systematically identify the transcriptional regulatory patterns and molecular differentiation trajectories of different cell types within multicellular biological tissues at the single-cell level[6,7]. This technology allows us to gain a deeper understanding of the changes and roles of different cell types during fruit senescence.

While scRNA-seq has been systematically used to identify cell populations in model plants like Arabidopsis and maize, it is important to note that databases and marker genes are still lacking for non-model plants like pitaya (*Hylocereus undatus, H. undatus*, also named

[1]College of Food and Bioengineering, Henan University of Science and Technology, Luoyang 471023, China. [2]Queensland Alliance for Agriculture & Food Innovation, Queensland Biosciences Precinct, The University of Queensland, St Lucia QLD 4072, Australia. [3]National Demonstration Center for Experimental Food Processing and Safety Education, Luoyang 471023, China. [4]College of Medical Technology and Engineering, Henan University of Science and Technology, Luoyang 471023, China. [5]Institute of Environment and Health, Jianghan University, Wuhan 430056, China. ✉e-mail: 13025569@qq.com; robert.henry@uq.edu.au

*Selenicereus undatus*), making it challenging to identify individual cell populations. Also, to date, researchers have utilized single-cell transcriptome sequencing technology to construct single-cell atlases for fruit and vegetables, such as strawberries (*Fragaria vesca*)[8] and lychees (*Litchi chinensis*)[9]; however, the tissues analyzed were leaves and shoot apices. There have been no reports utilizing scRNA-seq technology on the expression characteristics and variation patterns of different cell types during the senescence and decay processes of fruit. During fruit senescence and decay, it is worth exploring whether different types of fruit pericarp cells are involved in distinct cellular processes and possess distinct functions. These are important questions that warrant investigation.

The recent development of spatial transcriptomics methods has provided robust methods for addressing these questions[10,11]. Spatial transcriptomics technology generates datasets that not only include quantitative information on gene expression but also provide spatial distribution images of gene expression within tissues. Through correlation analysis with single-cell transcriptome data, it becomes possible to achieve spatial localization of different cell types within fruit pericarp tissues[10], thus supporting a comprehensive understanding of the functions of various cell populations within the pericarp during the senescence process.

*H. undatus* is a plant in the cactus family (Cactaceae) and is rich in sugars, plant proteins, vitamins, and water-soluble dietary fiber[12]. The fruit pericarp of *H. undatus* contains a high level of flavonoids, providing a solid foundation for its anti-senescence properties. In addition, when comparing the anti-senescence effects of the preservative Trypsin on various fruits and vegetables such as pitaya, mango, and cucumber, pitaya exhibited the optimal response to the preservative[4,5]. The thick and distinctly layered pericarp, coupled with its favorable response to preservatives, makes it an excellent candidate for studying the mechanisms of fruit senescence.

In this work, we conducted scRNA-seq and spatial transcriptomics analyses of the *H. undatus* fruit pericarp tissue. Through the application of four different algorithms, the data were correlated, enabling the construction of a comprehensive pericarp cell atlas. By categorizing these heterogeneous cell populations, we identified five major cell types and specific marker genes for each cell type. The expression localization of the senescence marker gene *HuSAG12* in the mesocarp was validated through in situ hybridization. Subsequently, we performed subpopulation analysis and pseudotime analysis on the two most significantly changing cell types, exocarp and mesocarp, during the fruit senescence process. This revealed the expression trends of pseudotime-related genes during senescence. Finally, we constructed a single-cell pseudotime regulatory network using the SCODE algorithm[13] and identified key genes within the network using plugins like CytoHubba.

The results of this study have elucidated the critical cell populations and the pseudotime trajectories of gene expression during postharvest fruit senescence. They reveal a strategy for early warning of decay by modifying the early response system in fruit senescence. This research provides ideas and technologies to reduce post-harvest losses and extend storage periods, thus playing a significant role in advancing fruit and vegetable industries.

## Results

### Generation of a cell atlas of *H. undatus* pericarp during senescence

The acquisition of single-cell transcriptomes from pericarp tissues posed a technical challenge, particularly when compared to more tender tissues like roots[14] or seedlings[15]. To systematically elucidate the gene expression patterns within the pericarp of *H. undatus* during the senescence process, we meticulously extracted pericarp tissues from both mature and senescent stages, followed by the application of droplet-based single-cell RNA sequencing (scRNA-seq) to generate a comprehensive transcriptomic atlas (Fig. 1a). Notably, the pericarp of *H. undatus* is characterized by its high content of lignin and polysaccharides, necessitating the optimization of protocols for protoplast isolation and impurity removal. Due to the fragility of protoplasts, a filtration step was employed to eliminate damaged cellular fragments and organelles, after which the purified protoplasts were loaded into the 10× Genomics Chromium Controller. In the CK (mature) and Post (senescent sample after storage, details see "Methods" section) samples, 80% and 77% of viable cells were retrieved, respectively.

Subsequently, scRNA-seq libraries were constructed on an Illumina platform and sequenced (Supplementary Table 1). A total of 6738 individual cells were successfully profiled in the CK group, while 9179 individual cells were profiled in the Post group. After stringent filtering, successful analyses were conducted on 5646 cells from the CK group and 7670 cells from the Post group (Supplementary Table 1). From the 422 million reads obtained from the pericarp samples, 91.8% to 95.4% were successfully mapped to the *H. undatus* reference genome. Furthermore, the expression of a median of 708 genes in the CK samples and 777 genes in the Post samples was detected using the *H. undatus* reference [Pitaya Genome and Multiomics Database (PGMD)].

To generate a comprehensive cellular atlas of *H. undatus* pericarp senescence, mature samples (CK group) were merged with post-storage senescent samples (Post group) for cell clustering and annotation. A total of 27,735 genes were identified from the single-cell transcriptome data and categorized into 13 distinct cellular clusters. Uniform Manifold Approximation and Projection (UMAP) was employed for visualizing and exploring these cellular clusters (Fig. 1b). It should be noted that these cellular clusters exhibited differences in their relative abundances between the CK group and the Post group. Specifically, cellular clusters 2, 3, 4, 7, 9, 10, and 11 were predominantly composed of cells from the CK group, while cellular clusters 0, 1, 5, 6, 8, and 12 contained a higher proportion of cells from the Post group (Supplementary Table 2).

To annotate each cellular cluster, we identified cluster-enriched genes with significantly higher expression levels in specific cellular clusters compared to all other cellular clusters (Supplementary Data 1). Among the 13 clusters, a total of 38 marker genes were screened. HU05G00061 was a marker gene in both Cluster 10 and Cluster 12 (Fig. 1c).

### Spatial transcriptome sequencing of the pericarp of *H. undatus* fruit and cell-type recognition

To achieve the spatial localization of the 13 cell clusters obtained from the single-cell transcriptome and thereby identify each cell cluster, we conducted spatial transcriptome analysis on CK samples using the 10× Visium platform (Fig. 2a).

The technology employed in this study involved placing frozen tissue sections on an array of 5000 spots, with each spot measuring $6.5 \times 6.5\,mm^2$. Each spot had a diameter of $55\,\mu m$ and contained approximately 100 million oligonucleotides. These oligonucleotides possessed specific positional barcodes and were attached to oligo (dT) primers (Fig. 2a). For the frozen sections of fruit pericarp, we further optimized enzymatic control of tissue structure permeabilization, specifically tailored for *H. undatus* (Supplementary Fig. 1). Out of the 325 million reads obtained from the pericarp samples, 98.0% of the barcodes were considered valid, ensuring data accuracy. The average number of reads per spot was 91,253, enabling comprehensive spatial transcriptome analysis.

The spatial transcriptome sequencing generated 3566 valid spots and 17,849 valid genes. Among them, 3652 genes (Supplementary Data 2) exhibited significant spatial variation between different tissues (Supplementary Fig. 2A, B). By applying Seurat's SNN algorithm to the dataset, beads were effectively classified into 7 clusters within 5 regions (Fig. 2b). These regions included exocarp

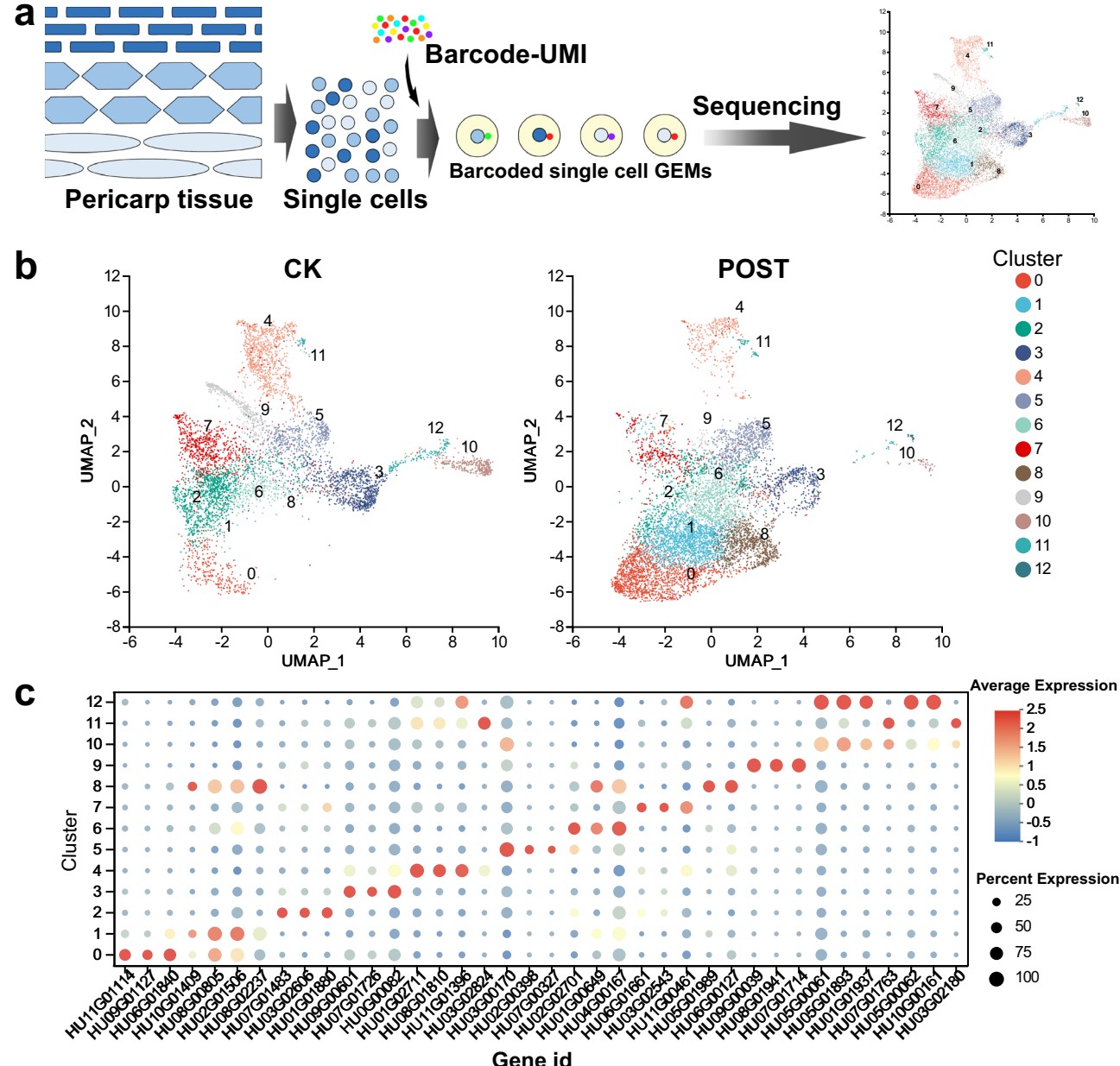

**Fig. 1 | Generation of a *H. undatus* pericarp cell atlas. a** Flowchart of experiments in this study. Different colors represent Barcode and UMI (Unique Molecular Identifier) sequences, where barcodes are used for cell differentiation, and UMIs are used for transcript differentiation. In 10× Genomics reagents, there are a total of 4 million Barcode variations, and UMIs consist of 10 nucleotides, allowing for up to 1,048,576 unique combinations. **b** UMAP visualization of 13 cell clusters in CK and Post group of *H. undatus* pericarp samples. Each dot denotes a single cell. Colors denote corresponding cell clusters. **c** Expression patterns of representative cluster-specific marker genes on UMAP. Dot diameter indicates the proportion of cluster cells expressing a given gene. The color scale represents the gene expression levels, with red indicating high expression and blue indicating low expression.

(EX, cluster 4), mesocarp (ME, clusters 1, 2, and 7), endocarp (EN, cluster 5), endocarp fibers (ENF, cluster 3), and vascular bundles (VB, cluster 6). The UMAP plot of the spatial transcriptome vividly depicted 5 distinct components of the pericarp (Supplementary Fig. 2C). Similar to the methods used in scRNA-seq analysis, each pericarp region's top marker gene was identified by comparing their expression differences with other regions. The spatial localization maps (left) and violin plots (right) of marker gene expression for each cluster showed that gene HU08G01266 specifically localized to EX, gene HU08G02237 specifically localized to ME, gene HU06G02555 was specifically localized to EN, gene HU07G02077 was specifically localized to ENF, and gene HU10G00163 was specifically localized to VB (Fig. 2c).

## Integrating microarray-based spatial transcriptomics and single-cell RNA sequencing reveals tissue architecture in pericarp of *H. undatus*

We employed two categories of four different methods to integrate and analyze spatial transcriptomic and single-cell transcriptomic data. The objective was to elucidate the spatial localization information of various cell clusters identified in the single-cell spectra and further reveal the composition of different cell types.

Based on analysis using SingleR (Single Cell Recognition) and SciBet (Single Cell Identificator Based on E-test), we observed that cell clusters 0, 1, 6, and 8 from the scRNA-seq data clustered together, suggesting that they belong to the ME component. Furthermore, cell clusters 2, 3, 5, 7, and 9 in the scRNA-seq data exhibited similar

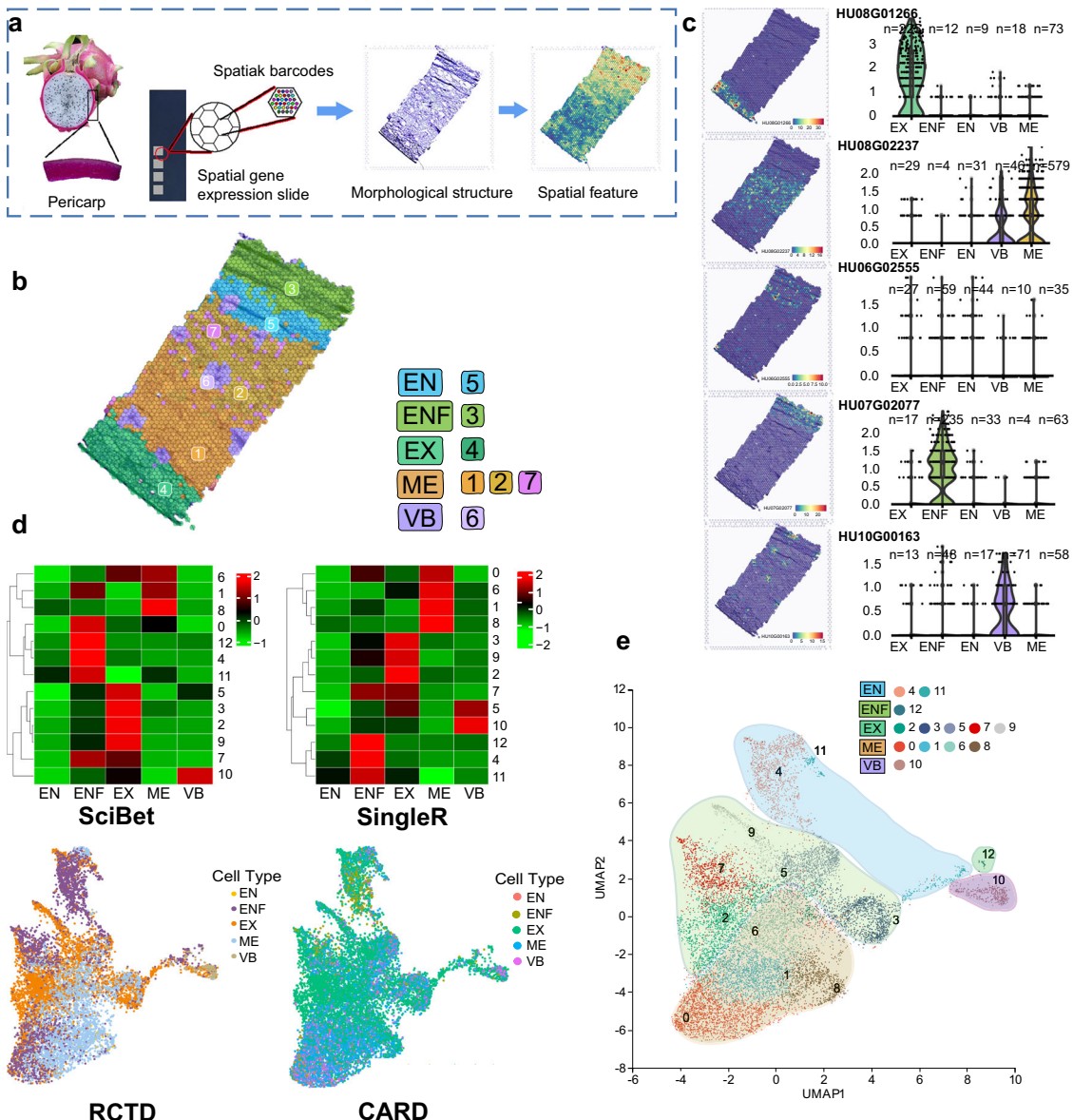

**Fig. 2 | Reconstruction of a cellular atlas for mature pericarp of *H. undatus* using spatial transcriptomics. a** Workflow for sampling and sequencing *H. undatus* pericarp on the 10× Visium platform. **b** Illustrations of cell types discovered on glass slides, overlaid on corresponding H&E-stained images. Clusters are named based on the spatial positioning of cell types. **c** Spatial localization maps (left) and violin plots (right) showing the expression of marker genes for different cell types. The violin plot showed six data nodes for each set of data, arranged from largest to smallest, namely the maximum value (upper edge), the upper quartile, the median, the lower quartile, and the minimum value (lower edge). The sample sizes (n number) are shown on each panel. **d** Association analysis between spatial transcriptomics and single-cell transcriptomics using four algorithms: SciBet, SingleR, RCTD, and CARD. **e** UMAP plot displaying 13 clusters of single cells classified into 5 cell types after spatial transcriptomics identification.

clustering patterns, indicating a likely association with the EX component, with cell cluster 7 possibly also belonging to the ENF component. Cell cluster 10 was classified as the VB component, although there is some uncertainty regarding whether cell cluster 5 also belongs to the VB component. Cell clusters 4, 11, and 12 showed clustering patterns consistent with the ENF component. The clustering data for cluster 12 was only concentrated in the ENF, indicating that cluster 12 belonged to ENF first, but it remains uncertain whether any of them belonged to the EN component (Fig. 2d).

The results obtained from regional cell type deconvolution (RCTD) and cell annotation in regional decomposition (CARD) were generally consistent, with the spatial localization of EX, ME, and VB cells matching the results from SingleR and SciBet (Fig. 2d and Supplementary Fig. 2D, E). However, there were differences in the localization of ENF and EN cells between these methods, and there was also

inconsistency between SingleR and SciBet (Fig. 2d and Supplementary Fig. 2D, E).

In the violin plot of Supplementary Fig. 2F, the marker gene of cluster 12 was only independently expressed in cluster 12. It is difficult to find the marker genes of cluster 4 that were only independently expressed in cluster 4. They were all co-expressed in clusters 4 and 11, and even had a certain expression level in cluster 12. The same applied to the marker genes of cluster 11. In addition to Hu03G02180 being specifically expressed in cluster 11, other marker genes were co-expressed in clusters 11 and 4 and even co-expressed in cluster 12. Therefore, when classifying the cell types of each cell cluster, we grouped cluster 4 and cluster 11 together. Due to the clear classification of cluster 12 as ENF, clusters 4 and 11 were classified as the second-highest-scoring EN components in the results of SciBet and SingleR (Fig. 2d).

Based on the results from the four algorithms and considering the results of violin plot of marker genes and the close proximity of cell clusters of the same type on the UMAP plot, the 13 cell clusters identified from the scRNA-seq data were categorized into five distinct cell types. Cell clusters 0, 1, 6, and 8 were classified as mesocarp (ME) cells. Cell clusters 2, 3, 5, 7, and 9 were designated as exocarp (EX) cells. Cell cluster 10 was assigned as vascular bundle (VB) cells. Cell clusters 4 and 11 were categorized as endocarp (EN) cells, while cell cluster 12 was labeled as endocarp fiber (ENF) cells (Fig. 2e).

The integration analysis of spatial transcriptomics and single-cell transcriptomics has successfully identified the cell types in *H. undatus* pericarp and constructed a comprehensive pericarp cell atlas.

## Subcluster analysis distinguished cells belonging to different samples in key components of *H. undatus* pericarp

A statistical analysis of cell proportions in the CK and Post-samples revealed significant changes in cell composition in the senescent pericarp after storage. The proportion of cells belonging to the exocarp (EX) component significantly decreased from 64.97% in the CK group to 25.97% in the Post group (Fig. 3a). Conversely, the proportion of cells in the mesocarp (ME) component significantly increased from 10.87% in the CK group to 67.46% in the Post group (Fig. 3a). These findings were further supported by UMAP visualization, clearly demonstrating the significant increase in ME cell

populations and the significant decrease in EX cell populations (Supplementary Fig. 3A).

Microscopic images clearly illustrated the changes in different layers of *H. undatus* pericarp. In the CK group samples, beneath the smooth and thick waxy layer, there were three to four layers of exocarp cells tightly arranged, with each layer exhibiting a relatively orderly structure. Moving inward, there was a large area of thin-walled mesocarp cells, densely packed with small gaps between them. Further inward was the endocarp layer, where cell arrangement was also relatively orderly (Fig. 3b). In the Post group samples, the wax layer became thinner and wrinkled, closely followed by an exocarp cell layer with increased gaps between cells. The mesocarp cell layer was significantly thinned, with only three to four layers of thin-walled cells maintaining their normal cell morphology. The remaining mesocarp cells experienced significant dehydration, resulting in noticeable wilting and wrinkling. The innermost endocarp cell layer appeared more disorganized, with significantly enlarged intercellular spaces (Fig. 3b).

Considering the significant changes observed in the EX and ME components during fruit senescence, the subsequent analysis focused on exploring the cellular subpopulations within these two components. After re-clustering EX and ME cells, five distinct subgroups were identified (Fig. 3c) and further categorized as CK subgroups (Clusters 1 and 2) and Post subgroups (Clusters 0, 3, and 4) (Fig. 3d). To

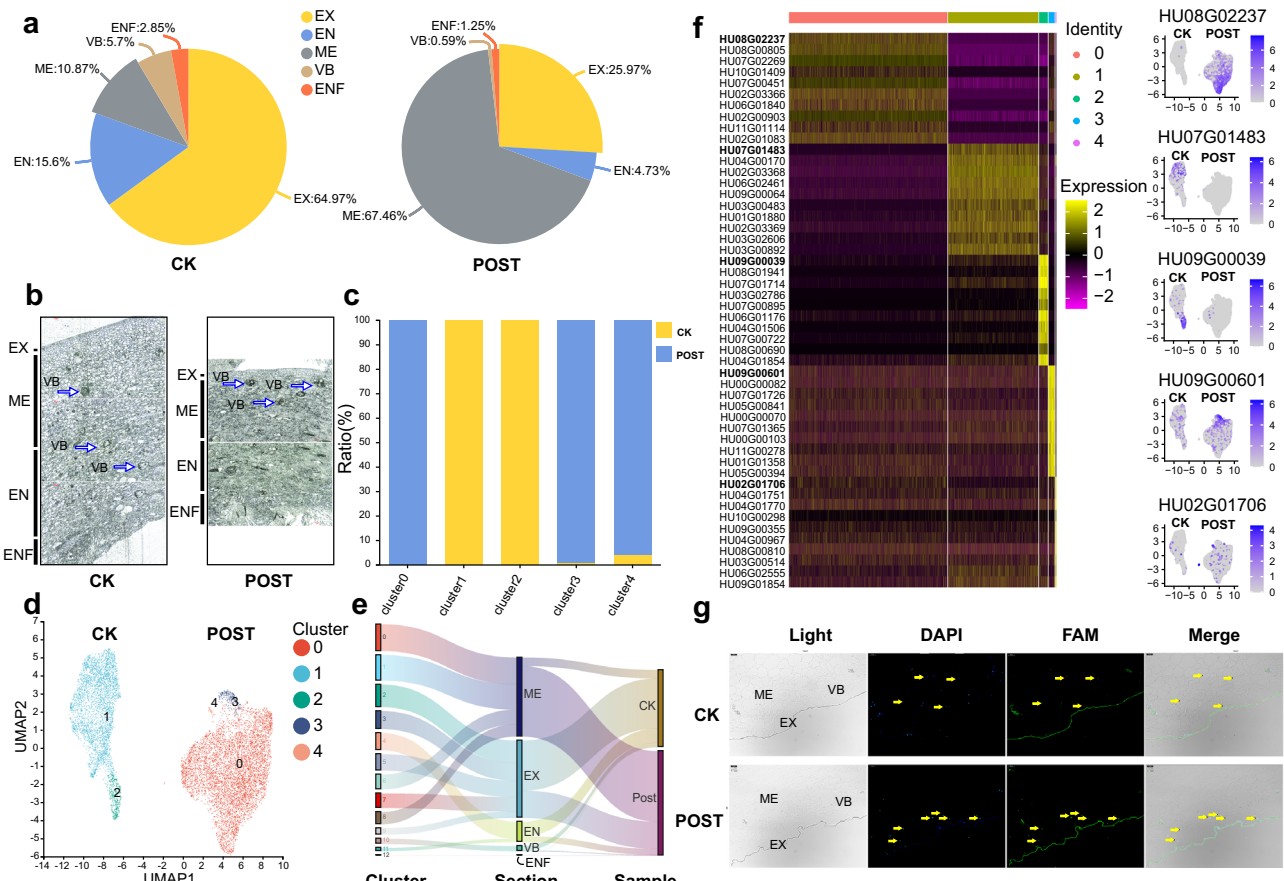

**Fig. 3 | Subcluster analysis for exocarp and endocarp cells. a** Pie charts depicting the percentage composition of various cell types in CK and Post samples. **b** Optical microscope images of CK and Post samples. Three independent experiments were repeated with similar results. **c** Bar charts illustrating the statistical proportions of CK and Post sample cells within each subcluster of cells. **d** UMAP plots for cells of various subclusters. **e** Sankey diagram showing the distribution of clusters in components and these cells gathered in components in CK and Post samples. Colors according to Fig. 1b. **f** Heatmap and UMAP of top 1 marker genes of

5 subclusters from the exocarp and endocarp cells, with the top1 gene ID emphasized. The UMAP plot in Fig. 3f illustrated the expression localization of the top 1 gene of subclusters 0, 1, 2, 3, 4 from the heatmap in each subcluster, consistent with the distribution of CK and Post in Fig. 3c. **g** RNA FISH indicated that the predominant location of *HuSAG12* was in the mesocarp. Three independent experiments were repeated with similar results. Components of EX, ME, and VB were labeled in the light field image. *HuSAG12* probes were labeled with FAM (green). Nuclei were stained with DAPI (blue). Scale bar: 40 μm.

comprehensively depict the relationships among the 13 clusters from scRNA-seq and these five components, as well as their distribution in CK and Post samples, we employed a Sankey diagram. Notably, the diagram clearly illustrates the specificity of the EX component to CK samples and the ME component to Post samples. Certainly, a small fraction of the EX components were attributed to the Post samples (Fig. 3e). Furthermore, a comparison of the expression levels of the top 100 marker genes within each subgroup confirmed distinct gene expression profiles among the five identified subgroups (Fig. 3f).

Based on the scRNA-seq profiles, the UMAP overview in CK and Post samples showed the expression localization of the top gene in each of the five subgroups (Fig. 3f and Supplementary Data 3). Hu08G02237 exhibited expression specificity in the Post sample, mainly located in the mesocarp. The spatial transcriptomics results also confirmed the specific localization of this gene in the fruit's mesocarp. This result was further validated by RNA–FISH, indicating that the expression of Hu08G02237 was primarily localized in the mesocarp, whether from CK or Post samples (Fig. 3g). Moreover, the RNA–FISH results clearly showed that the expression level of Hu08G02237 was higher in Post samples compared to CK samples (Fig. 3g). This finding was consistent with the results obtained from single-cell transcriptome sequencing (Hu08G02237 expression in CK and Post was 0.0395 and 2.395, respectively).

## Pseudotime analysis revealed the time conversion trajectories of mature and senescent cells in the mesocarp and exocarp

As is well-known, in leaves, the differentiation rate at the edges is slower than in the central parts[16]. However, during the senescence process of the pericarp, the differentiation patterns of different parts of the pericarp have remained unclear. To clarify this issue, we selected the exocarp and mesocarp cells in *H. undatus* pericarp, which exhibited the most significant changes during senescence, and conducted pseudotime analysis.

The gene expression changes were plotted along pseudotime, and these genes were divided into four clusters (Supplementary Fig. 3B). For Cluster 3, genes exhibited early high expression along the pseudotime axis and were enriched for GO and KEGG entries associated with oxidative stress, chitin metabolism processes, and carbohydrate derivative metabolism processes (Supplementary Data 3). In the relevant clusters in the exocarp, they were upregulated, indicating an initial cellular stress response in the exocarp. Next, genes in Clusters 2 and 4 showed higher expression in the middle of the pseudotime axis, also corresponding to cells in the exocarp. Their functions primarily involved phenylpropanoid biosynthesis pathways downstream flavonoid biosynthesis and cellular stress resistance (Supplementary Data 3). The senescence pathways associated with Cluster 2 can be summarized in a DAG (Supplementary Fig. 3C). For example, the GO biological process "flavonol biosynthetic process (GO:0051555)" is a child term of two terms: "flavonol metabolic process (GO:0051554)" and "flavonol biosynthesis process (GO:0051553)" (Supplementary Fig. 3C). Genes in Cluster 1 were primarily upregulated in the later stages along the pseudotime axis, with high expression in mesocarp clusters 0, 1, 6, and 8, related to fruit senescence (represented by HuSAG12, gene ID HU08G02237) and seed development (represented by HuSUS, gene ID HU02G00890) (Supplementary Data 4).

To investigate the developmental trajectories of mature and senescent cells, gene sets were constructed based on the results mentioned above, including 52 senescence-related genes, 642 resistance-related genes, 1913 reactive oxygen species (ROS)-related genes, and 258 phenylpropanoid pathway-related genes (Supplementary Data 5).

We conducted pseudotime and embedding heatmap analysis for genes in the four gene sets mentioned above. To clarify the pseudotime expression patterns of each gene set, eight key pseudotime nodes, consisting of the first and last genes in the four clusters of the overall pseudotime graph, were selected. These eight genes served as critical pseudotime nodes for the overall senescence pseudotime and were plotted together with the genes from the gene sets mentioned above, elucidating the positions of gene expression for each gene set within the overall pseudotime. The results demonstrated that both resistance-related genes and phenylpropanoid pathway-related genes involved in resistance, as well as senescence-related genes and ROS-related genes involved in senescence, exhibited temporal expression patterns along the pseudotime (Supplementary Fig. 3D). Furthermore, the heatmap revealed the pseudotime expression patterns of some functionally unknown genes, such as HU05G00200 in resistance-related genes, identified as an uncharacterized protein LOC104893067 in the database, and HU07G02057 in senescence-related genes, identified as a hypothetical protein GH714_016062 in the database. However, based on their early response to senescence, these genes are likely to play important roles in fruit senescence. Taken together, our results reveal a differentiation continuum of pericarp cells during senescence.

To find cells with a high correlation with the above-mentioned key gene functions and make the trajectory of fruit senescence clearer, gene set scoring was used to screen target cells[17,18]. The results showed that, compared to the CK group, the Post group had a significant increase in cells related to senescence and ROS, while cells related to resistance and the phenylpropanoid pathway significantly decreased (Fig. 4a). Based on the gene set scoring results, 9019 cells highly correlated with the mentioned genes were selected. These cells were further subjected to subcluster analysis, resulting in 13 subclusters (Supplementary Fig. 3E), with subclusters 0, 3, 4, 8, 10, and 11 primarily consisting of cells from the CK sample, while the remaining subclusters were mainly composed of cells from the Post sample, as depicted in the uMAP plot (Supplementary Fig. 3F).

The developmental trajectories of 13 subpopulations consisting of 9019 cells highly correlated with senescence were subsequently delineated. The results revealed three major trajectories, with cells from the mature (CK group) and senescent (Post group) samples aligning along pseudotime paths (Fig. 4b).

This ordering was captured by latent time (Fig. 4b). As depicted in Fig. 4b, mature cells (MC) and senescent cells (SC) occupied distinct branches. Mature cells from the CK sample were distributed at a relatively early pseudotime stage, situated in the Pre-branch, and categorized as State 1. Mature cells with a later differentiation time resided in Branch 1, primarily comprising cells with high expression of resistance genes, referred to as resistance cells (RC), and were classified as State 3. A small subset of early-stage senescent cells from the Post sample was classified as State 3, while the majority of cells from the Post sample at various stages of senescence exhibited the latest differentiation time and predominantly expressed genes indicative of a senescent state, referred to as senescent cells (SC) and classified as State 2 (Fig. 4b). The results showed that, overall, cells from the CK sample exhibited lower differentiation compared to cells from the Post sample, with cells from the Post sample being in a more open differentiation state (Fig. 4b).

When considering spatial information within the developmental trajectory, we found that the cells in the pre-branch included both EX and ME cells (Fig. 4b and Supplementary Fig. 4A–C). The resistance cells in Branch 1 exhibited a distinct spatial distribution pattern, primarily belonging to the CK sample, mainly located in EX, and with a later differentiation time (Fig. 4b and Supplementary Fig. 4D). On the other hand, the senescence cells in Branch 2, primarily from the Post sample, were mainly located in ME and had the latest differentiation time (Fig. 4b and Supplementary Fig. 4E).

Based on the gene expression characteristics, cells were categorized into three states according to RNA velocity (Fig. 4c). The

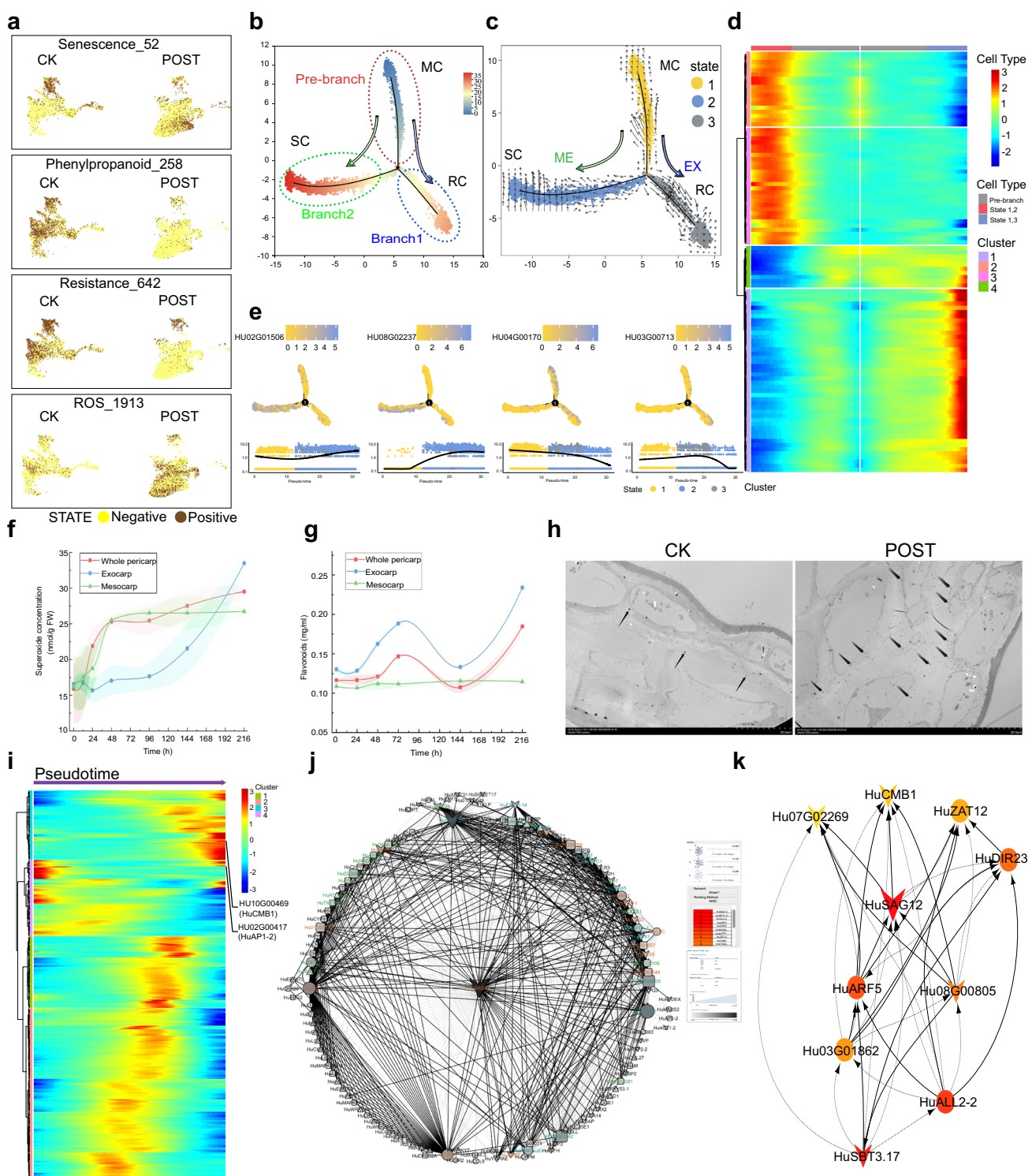

STATE ● Negative ● Positive

results of RNA velocity analysis showed that MC (mature cells in CK samples) formed trajectory sources leading to both RC (resistant cells in the EX of CK samples) and SC (senescent cells in the ME component of Post samples) directions (Fig. 4c). RNA velocity dynamics also revealed stable attractor states for different cells. The stable attractor state under mature conditions was represented by MC cells. Additionally, RC cells represented the attractor state under resistant conditions, while SC in the Post samples became the attractor state in the late storage period (Fig. 4c).

Furthermore, we assessed the gene expression patterns along the pseudotime based on the differentiation branching points from the

Pre-branch to Branch 1 and from the Pre-branch to Branch 2 within EX and ME cells. Cluster analysis revealed four expression clusters representing differentiation patterns of four classes of genes (Fig. 4d). As expected, KEGG terms related to senescence, such as amino acid metabolism, fatty acid degradation, sphingolipid metabolism, amino sugar and nucleotide sugar metabolism, were enriched in cells from clusters 2 and 3, which were pointing towards Branch 2 (Fig. 4d). These clusters represented gene expression changes in cells transitioning from a fresh state to senescence. Cluster 2 was enriched with genes involved in oxidative phosphorylation pathways, likely responsible for senescence induction.

**Fig. 4 | Senescent trajectories of exocarp cells and mesocarp cells. a** Gene set scoring results plot showcasing cells highly correlated with target genes in CK and Post samples. **b** Latent time showed the internal clock of cells. Different colors of latent time represent different differentiation times, with darker shades of red indicating earlier times and darker shades of blue indicating later times. **c** RNA velocity analysis mapped three cellular states on the pseudotime plot. The colors represent different cell states. The direction of the black arrows represents the potential trajectories of the cells, and the length of the arrows represents the strength of the trends. **d** Clustering of differentially expressed genes along a pseudotime progression of EX cells and ME cells. **e** Visualization of the gene expression patterns of top genes in the clusters of Fig. 4d mapped onto the pseudotime trajectory. Pseudotime mapping of each gene with expression curves below. The color of each point represents different cellular states, and the horizontal axis represents time progression from left to right. The figure illustrates the gene's expression changes across three different states of cells over time. **f**, **g** Changes in endogenous superoxide anion and flavonoid concentrations within pericarp during the post-harvest storage in CK and Post samples. Data are presented as mean values ± SD. The style of connecting is spline. The area under curve is filled. **h** Accumulation of endogenous $H_2O_2$ within the pericarp of CK and Post samples. Three independent experiments were repeated with similar results. Arrows indicate the deposition of cerium peroxide ($Ce[OH]_2OOH$ and $Ce[OH]_3OOH$) formed after $CeCl_3$ staining, representing the deposition of $H_2O_2$. Scale bar: 20 μm. **i** Clustering of 529 transcription factors-encoding genes along a pseudotime progression. **j** The Gene Regulatory Network (GRN) was inferred from the dynamic expression of top 100 genes at pseudo-temporal branching points and 529 transcription factors integrated dynamically expressed across senescence differentiation pseudotime with a parameter cutoff of 2.0. Solid and dotted lines represent positive and negative regulation, respectively. Node size corresponds to the predicted connectivity. Nodes from clusters obtained via MCODE are labeled with different colors. Nodes that are specifically upregulated during senescence in the EX and ME sections were also depicted as inverted triangles and triangles. Nodes co-expressed in EX and ME were represented as circles. **k** Hierarchical layout of 10 hubs in Fig. 5J. The nodes were ranked and colored by cytoHubba.

On the other hand, KEGG terms associated with resistance, including "Cutin, suberine and wax biosynthesis," "Phenylpropanoid biosynthesis," "Pentose and glucuronate interconversions," and more, were significantly enriched in cells from clusters 1 and 4, which were pointing towards Branch 1 (from state 1 to state 3) and contained nearly all of the resistance cells. Cluster 4 was enriched with genes highly expressed in pathways related to phenylpropanoid biosynthesis, including flavonoid synthesis, as well as the pentose and glucuronate interconversions pathway. This pathway is crucial in plants and contributes to maintaining the structural integrity of cell walls, defense against pathogens, and detoxification of foreign substances. It provides precursors for the biosynthesis of structural components of the cell wall, such as hemicelluloses and pectin, which serve as physical barriers against pathogens. This pathway also contributes to the synthesis of secondary metabolites with antimicrobial properties. Genes in cluster 4 primarily play roles in plant defense, antimicrobial functions (HuTL1, thaumatin-like protein 1b), stress resistance (HuP-GIP1, polygalacturonase inhibitor-like), and participation in plant defense responses (HuALL2-2, hypothetical protein CDL15_Pgr017890). Genes in cluster 1 were enriched in pathways such as "Cutin, suberine and wax biosynthesis" and "Plant-pathogen interaction," laying the foundation for resistance in exocarp cells. Clusters 1 and 4 represented gene expression changes from fresh to stress-resistant cells (Supplementary Data 6).

Mapping the expression changes of marker genes onto the pseudotime trajectory revealed that the gene HU04G00170, involved in defense responses, remained highly expressed in mature cells at the pre-branch and in resistance cells at branch 1. The gene HU03G00713, specifically expressed in resistance cells, was preferentially distributed along branch 1 of the trajectory (Fig. 4e and Supplementary Data 7). In contrast, the senescence marker genes HU08G02237 and HU02G01506 exhibited increased density and expression levels at the terminal points of these trajectory branches (Fig. 4e).

### ROS and flavonoid phenotypic changes and their regulatory mechanisms in mature and senescent cells

To validate the temporal trajectory of fruit senescence as determined by pseudotime analysis, the levels of ROS and flavonoids in the exocarp and mesocarp of the fruit were examined during the senescence process. The results revealed a significant increase in the accumulation of endogenous ROS in the fruit pericarp as senescence progressed. The accumulation of ROS was primarily concentrated in the mesocarp and endocarp regions (Supplementary Fig. 4F). At day 0, ROS accumulation was detected in both the mesocarp and endocarp, likely due to cellular damage during tissue sectioning, while the exocarp exhibited very low levels of ROS. However, after 6 days of storage, ROS accumulation was observed in the exocarp as well, and by this time, the mesocarp and endocarp had already exhibited relatively high levels of ROS (Supplementary Fig. 4F). It's worth noting that DCF can detect various ROS species, including superoxide anions, hydrogen peroxide ($H_2O_2$), and ROO radicals[19], and so further separate measurements were performed for superoxide anions and $H_2O_2$ to clarify the pattern of ROS accumulation.

Further analysis of superoxide anions revealed that their accumulation was primarily observed in the mesocarp. Superoxide anion response occurred very early, with a significant increase in superoxide anions detected in the mesocarp as early as 12 h into storage. The exocarp maintained low levels of superoxide anions throughout storage until a burst of superoxide anions was observed in the late stages of fruit decay (Fig. 4f).

The results of flavonoid detection indicated that the flavonoid response in the pericarp occurred later than the response of other ROS, such as superoxide anions. The overall flavonoid levels in the pericarp began to increase on the second day, reached the first peak on the third day, decreased during the mid-term of storage, and then increased again to the highest levels during the late stages of fruit decay. When the exocarp and mesocarp were sampled separately, the flavonoid levels in the exocarp followed a pattern consistent with the overall pericarp changes and were significantly higher than the overall levels. In contrast, the mesocarp showed minimal fluctuation in flavonoid levels during the fruit senescence process (Fig. 4g). Specific staining for $H_2O_2$ in electron microscopy showed a significant increase in $H_2O_2$ accumulation in the pericarp of the Post group (Fig. 4h, right panel) compared to the CK group (Fig. 4h, left panel). Hydrogen peroxide primarily accumulated in the exocarp, but at lower levels and with a more uniform distribution, without the characteristic clustering seen in oxidative bursts following pathogen infection (Supplementary Fig. 4g).

The gene regulatory network underlying the pericarp senescence process was inferred using SCODE (Supplementary Data 8). By integrating the dynamic expression of the top 100 genes at the pseudotime branch points and pseudotime-related transcription factors (Fig. 4I), we revealed a complex network governing pericarp senescence (Supplementary Data 8). We obtained three clusters through the Cytoscape plugin "MCODE", which respectively contained 13, 15, and 15 nodes (Supplementary Data 8). Notably, proteins encoded by genes such as the senescence marker gene HuSAG12, resistance-related gene HuDIR23, auxin response factor HuARF5, and transcription factor HuCMB1 were all assigned to Cluster 2 (Fig. 4j).

To explore the potential regulatory mechanisms of the senescence process, the first objective was to identify hubs within the network of pseudotime-related genes (PRGs). Investigating the interactions among PRGs will help in understanding the roles of PRGs in pericarp cell senescence.

In the CytoHubba plugin within the Cytoscape software, the widely recognized senescence marker gene *HuSAG12* was identified as the most highly connected hub regulatory factor (Fig. 4k). Interestingly, using 12 algorithms such as MNC and closeness in cytoHubba, nine algorithms yielded nearly identical top 10 nodes (Supplementary Data 8). Apart from the top-ranked *HuSAG12*, which was specifically upregulated in the mesocarp during *H. undatus* senescence, *HuSBT3.17*, *HuCMB1*, and HU08G00805 also exhibited mesocarp-specific upregulation during the *H. undatus* senescence process. The network also highlighted two transcription factors among the top 10 nodes, *HuCMB1* (HU10G00469, MADS-box protein CMB1), and *HuARF5* (HU02G01225, auxin response factor 5). In the Gene Regulatory Networks (GRNs) focusing solely on transcription factors, *HuCMB1* and *HuARF5* also ranked among the top three crucial nodes (Supplementary Fig. 4H). Furthermore, the pseudotime heatmap revealed another transcription factor, HuARF5, among the top ten, exhibiting early to mid-stage expression during senescence. Hormone profiling showed changes in various hormone levels during the *H. undatus* senescence process (Supplementary Fig. 4I). Combining these results with the pseudotime analysis suggested that auxin might be a key hormone associated with senescence in *H. undatus*.

In order to further elucidate the changes in flavonoids and other metabolites in different components during the senescence process, mass spectrometry was employed to detect the metabolite levels in *H. undatus* at four storage time points. A data-independent acquisition (DIA) method, sequential window acquisition of all theoretical fragment-ion spectra (SWATH), was utilized for data analysis[20,21]. Using this method, the instrument deterministically fragmented all precursor ions within the predefined m/z range in a systematic and unbiased fashion.

A total of 3015 metabolites were identified in both negative and positive ion modes (Supplementary Data 9). Among them, 697 metabolites were matched to the HMDB database (Supplementary Fig. 5A), including 153 lipids and lipoid molecules (21.95%), 135 organoheterocyclic compounds (19.37%), 94 phenylpropanoids, and polyketides (13.49%). (Supplementary Data 8). PLS-DA results showed that compounds identified in the exocarp, mesocarp, and endocarp could be clearly distinguished in both positive and negative ion modes (Supplementary Fig. 5B, C).

To explore the differences in metabolites among different components, the top 30 metabolites were identified based on VIP > 1 and $p$ < 0.05 (Supplementary Fig. 5D and Supplementary Data 9). The Venn diagram illustrated the differences in metabolites among the three main components—exocarp (EX), mesocarp (ME), and endocarp (EN)—with no compound overlapping among the differentially regulated metabolites in all three components (Supplementary Fig. 5E). KEGG enrichment analysis results revealed the pathways involved in the metabolites of these three components. The differentially regulated metabolites in the exocarp component were mainly enriched in the "Phenylpropanoid biosynthesis" pathway compared to the other two components, while metabolites in the mesocarp component were mainly enriched in pathways such as "Biosynthesis of cofactors," "Arginine biosynthesis," and "Histidine metabolism" (Supplementary Fig. 5F).

### Functional validation of HuCMB1 during the senescence process through VIGS

To validate the accuracy of scRNA-seq and pseudotime heatmap analysis, four genes with high expression at different stages of the senescence process were selected for RT-qPCR analysis. In addition to the hub genes identified in the gene regulatory network (GRN), namely *HuCMB1, HuSAG12*, and *HuMED32*, considering their high expression in both early and late senescence stages, the gene *HuERD6-2*, which showed elevated expression in the mid-senescence-period-according-to the pseudotime heatmap, was also included.

The RT-qPCR results revealed that the *HuCMB1* gene exhibited specific high expression in the mesocarp and showed elevated expression in the mid and late stages of senescence ($p$ < 0.01). The well-known senescence marker gene *HuSAG12* exhibited extremely high expression only in the late senescence stage of the mesocarp component ($p$ < 0.05). *HuERD6-2* was found to be highly expressed in the exocarp during the mid-senescence stage ($p$ < 0.01), and the gene *HuMED32* showed high expression in the late-senescence stage of the exocarp component ($p$ < 0.05) (Fig. 5a, Supplementary Data 10).

To further confirm the role of *HuCMB1* in fruit senescence, RNA-silenced lines of the *HuCMB1* gene were constructed through virus-induced gene silencing (VIGS) (Fig. 5b). In comparison to the control, the fruit exhibited a darker color, dried and developed lesions more rapidly, and showed an accelerated senescence process after *HuCMB1* silencing (Fig. 5c). The rate of weight loss during senescence slightly increased (Fig. 5D) ($p$ > 0.05), and there was a significant reduction in the early-stage flavonoid biosynthesis (24 h, $p$ < 0.05), resulting in an overall smoother bimodal curve of flavonoid accumulation (Fig. 5e). RT-qPCR results demonstrated a significant downregulation of *HuCMB1* gene expression in the RNA-silenced lines, along with a substantial decrease in the expression of *HuSAG12* gene. The expression of the *HuERD6-2* and *HuMED32* genes showed a slight up or downregulation in the RNA-silenced lines, respectively (Fig. 5f).

In summary, our results revealed the developmental trajectory of pericarp cells during senescence and provided insights into the interactions between the senescence-related genes and resistance-related genes that exhibited specific localization in the mesocarp and exocarp during the developmental processes of senescence and resistance cells (Fig. 6).

## Discussion

Nowadays, there are over a thousand reports using single-cell transcriptomics (scRNA-seq), with the majority of these articles focused on studying animal cells[22–24]. Currently, there are various platforms for scRNA-seq, including BD Rhapsody (BD Biosciences, USA), and Chromium (10× Genomics, USA), among others, providing multiple possibilities for high-throughput analysis[25]. Drop-based high-throughput and low-cost cell processing platforms, such as Drop-seq or the Chromium 10x platform, dominate the field of plant single-cell transcriptomics. Using these platforms, studies have been conducted on various plant tissues, including Arabidopsis roots and leaves, rice stems, and sheaths[26,27].

However, regardless of whether the research subject is animals or plants when examining all the single-cell transcriptomic articles, the analyses typically begin with cell type annotation. The reason behind this is that the data obtained from single-cell transcriptomics encompass a vast number of cells, and the cell clusters defined by expression characteristics ultimately belong to mathematical groupings[22]. Such cell groups only acquire biological significance for further downstream analysis after undergoing cell type annotation. For plant studies, cell type annotation poses even greater challenges[14,15].

Some databases provide information on marker genes for certain plant species, such as Arabidopsis and rice[28,29]. However, information on marker genes for cacti, a family of plants that includes *H. undatus*, remains limited. Additionally, most marker genes have been identified from root cells and flowers, leading to a lack of cell type information related to pericarp in existing databases, which mainly cover root epidermis and epidermal cells. In this current work, we found that the marker gene HU08G02237 from cell cluster 1 in *H. undatus* matches the marker gene SAG12 (senescence-associated gene 12) from Arabidopsis petiole in the PlantCellMarker database[30,31]. Furthermore, the marker gene HU05G01893 was identified as matching the marker gene LOC_Os12g30150 (calmodulin depedent protein kinases) from the *Oryza sativa*in the

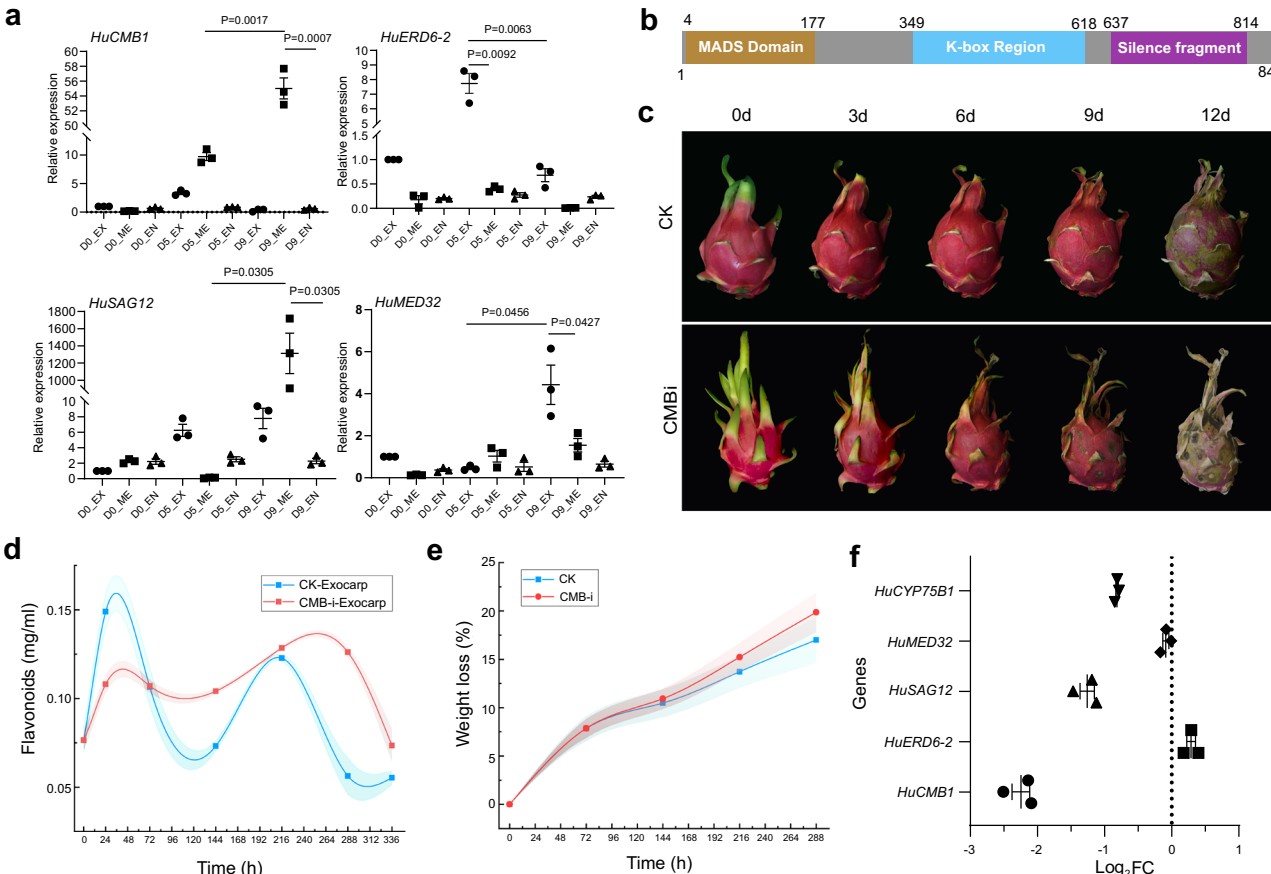

**Fig. 5 | Silencing of *HuCMB1* led to faster senescence of *H. undatus*. a** Expression of four senescence-related genes at different components of pericarp and time points of senescence. *n* = 3 biologically independent samples. Data are presented as mean values ± SD. A paired two-tailed *t*-test was used for all statistical analyses. No adjustments were made for multiple comparisons.* represents *p* < 0.05, and ** represents *p* < 0.01. **b** Schematic showing the domain structure of *HuCMB1* gene. **c**–**e** Changes in fruit phenotype (**c**), weight loss rate (**d**), and flavonoid levels in the exocarp (**e**) after silencing the *HuCMB1* gene. *n* = 5 biologically independent samples used in (**c**–**e**). **d**, **e** The style of connect is spline. The area under the curve is filled. **f** RT-qPCR was used to analyze the expression of *HuCMB1*, *HuERD6-2*, *HuSAG12*, and *HuMED32* in *H. undatus* after 9 days of storage under *HuCMB1* gene silencing. **d**–**f** *n* = 3 biologically independent replicates for each experiment. Data are presented as mean values ± SD. No adjustments were made for multiple comparisons.

PlantCellMarkerdatabase[32,33]. This suggested that HU05G01893 may be located in the vascular bundle region of *H. undatus* pericarp.

At the same time, it was observed that even among marker genes, the specificity of expression varies across different clusters. Meanwhile, we observed that even among marker genes, the expression specificity varied across clusters. Some clusters had marker genes with higher expression specificity within the cluster, such as marker genes HU07G01483, HU03G02606, and HU01G01880 in Cluster 2, and marker genes HU09G00039, HU08G01941, and HU07G01714 in Cluster 9. However, some clusters had marker genes with lower expression specificity, as they were also expressed at higher levels in several other clusters. For example, marker genes HU10G01409, HU08G00805, and HU02G01506 in Cluster 1 had relatively high expression in Clusters 0, 6, and 8. Similarly, marker genes HU02G02701, HU01G00649, and HU04G00167 in Cluster 6 exhibited higher expression in Clusters 1 and 8. Multiple marker genes are expressed at high levels in these four clusters 0, 1, 6, and 8. These findings strongly suggest that cell clusters 0, 1, 6, and 8 may belong to the same cell type (see "Results" section). However, due to the lack of information on other cell types in the pericarp, such as the exocarp, mesocarp, and endocarp, we currently cannot obtain a more comprehensive understanding of cell types from existing databases. Therefore, it is necessary to acquire clearer spatial information and cell type-specific expression patterns.

During the post-harvest storage of fruit, understanding the gene expression changes in different cell types within the pericarp is crucial

for comprehending the senescence process. However, specific gene expression changes within various cell types of the pericarp have largely remained unexplored. Our objective was to construct a spatial cellular map of the pericarp and identify cell types associated with the 13 cell clusters obtained from scRNA-seq analysis. To achieve this goal, we performed spatial transcriptome sequencing (stRNA-seq) on cross-sections of *H. undatus* pericarp using the 10× Visium platform (see "Results" section). This approach enabled us to investigate the spatial distribution of gene expression within the pericarp and gain deeper insights into the cellular composition of this tissue.

Currently, stRNA-seq technologies can be categorized into three main types: laser capture microscopy-based methods (LCM-seq, Geo-seq), imsenescence-based methods (FISSEQ, MERFISH, osmFISH, BaristaSeq, STARmap), and in situ capture sequencing methods (Visium, HDST, slide-seq, DBiT-seq, Seq-Scope, Stereo-seq)[34]. Microdissection-based methods can achieve single-cell or even subcellular resolution but have limitations in terms of the number of detected genes. Imsenescence-based methods mostly allow single-cell resolution but are constrained by long image acquisition times and the need for complex equipment. The resolution of most in situ capture sequencing methods is usually limited by the diameter of the capture points. These stRNA-seq technologies with single-cell resolution can infer subcellular structures and address cellular heterogeneity.

However, in the context of plant research, the successfully applied stRNA-seq technologies are primarily the commercial platforms 10X

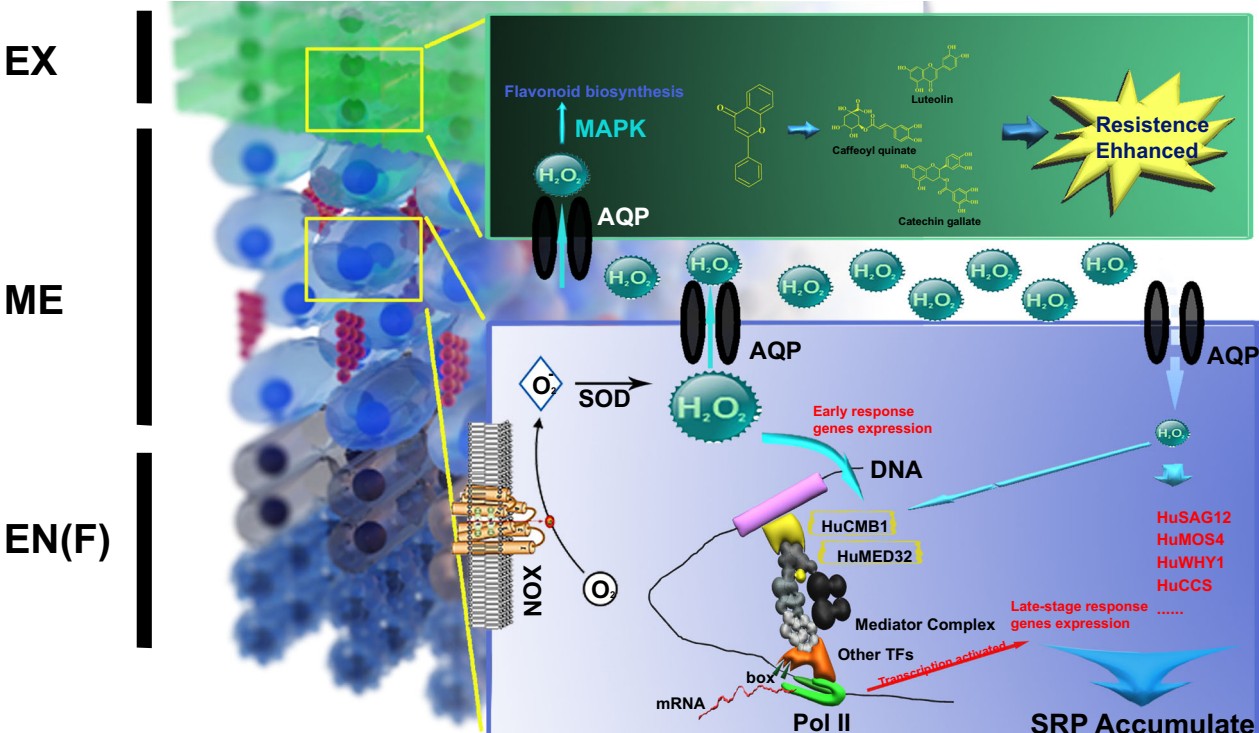

**Fig. 6 | Hypothesis of senescence regulatory trajectories in the exocarp and mesocarp cells.** Our findings indicated that the senescence process of the fruit exhibited a distinct spatiotemporal transition. The activation of ROS signals in the mesocarp occurred initially, followed by a significant upregulation of resistance genes in the exocarp cells. In the mid to late stages of fruit senescence, senescence-related genes like *HuSAG12* showed high expression in the mesocarp. The early

responsive transcription factor HuCMB1 held a central position in the senescence regulatory network created using the SCODE algorithm. EX represents the exocarp; ME represents the mesocarp; EN(F) represents the endocarp and endocarp fibers. SRP represents senescence-related proteins. SOD stands for superoxide dismutase. AQP denotes aquaporin proteins. Pol II represents RNA polymerase II. TF refers to transcription factor.

Genomics' Visium and BGI's Stereo-seq. For instance, Xia et al. used Stereo-seq to examine Arabidopsis leaves, distinguishing morphologically distinct cell subtypes (upper and lower epidermis, spongy, and palisade mesophyll cells) with similar transcriptomic features[35]. For large-scale plant tissues requiring the detection of numerous potentially functional genes, high-throughput stRNA-seq technologies are foundational. Visium's spatial chips can capture tissue slices spanning several millimeters, such as Arabidopsis inflorescences[36] and early floral organs of orchids[37].

Despite the promising applications of current stRNA-seq technologies in plants, challenges persist in terms of sampling difficulties and data processing. The spatial transcriptomic technique utilized in this study underwent optimization across multiple steps, including fixation, probe capture, and library construction[19,20], and it was successfully applied to perform spatial transcriptome sequencing of *H. undatus* pericarp tissue (see "Results" section). Next, we aim to address the challenges in the data processing of stRNA-seq through a correlation analysis with scRNA-seq.

Single-cell RNA sequencing (scRNA-seq) can systematically identify cell populations within tissues, but characterizing their spatial organization remains challenging. In this study, a microarray-based spatial transcriptomics approach was combined with scRNA-seq generated from the same sample. This method utilizes an array of spots to reveal spatial patterns of gene expression, with each spot capturing the transcriptome of multiple neighboring cells. Various analytical methods were employed to decipher the diverse cell types obtained from individual cells of *H. undatus* pericarp, enabling a precise characterization of cell composition across different tissue regions.

SingleR and SciBet cluster and annotate cells by annotation of marker genes and supervised learning[38,39]. Because the components of

the spatial transcriptome of the *H. undatus* pericarp were clearly located, it was accurate to evaluate the cell type obtained in scRNA-seq through the data set of the spatial transcriptome by using SingleR and SciBet. RCTD and CARD were also used in this work, both of which are deconvolution algorithms[40,41]. In fact, there were two or more types of cells in a part of the spot in the spatial transcriptome. Because of the assumptions of RCTD and CARD, when using the RCTD and CARD algorithm, the components with a high proportion of cells in the whole sample, such as EX (42.50%, Supplementary Table 2) and ME (43.47%, Supplementary Table 2), or the component VB with clear location, which was significantly different from other components, were more accurate. For components with fewer cells and less distinct partitioning compared to other components (EN and ENF), the calculation results of RCTD and CARD had significant errors.

After the identification of cell types within the 13 cell clusters, both the statistical analysis of cell proportions and microscopic observations revealed that the exocarp (EX) and mesocarp (ME) were the two cell types undergoing the most significant changes during fruit pericarp senescence (see "Results" section). Subcluster analysis segregated the EX and ME cells from both samples, attributing them to CK and Post samples, respectively. In both scRNA-seq and spatial transcriptomic results, Hu08G02237 (*HuSAG12*) was identified as the marker gene for the mesocarp. RNA-FISH results were consistent with both single-cell transcriptomics and spatial transcriptomics, showing that the expression of the senescence marker gene *HuSAG12* was primarily localized in the mesocarp. In the Post sample, due to a significant amount of cell death, there were fewer cell nuclei stained with DAPI, resulting in a lower number of cells showing RNA expression with FAM staining. On the other hand, the cells stained with FAM in the Post sample exhibited significantly higher fluorescence intensity compared to the CK sample, indicating an upregulation of expression

in the senescing sample, in line with the results from single-cell and spatial transcriptomics (see "Results" section).

Pseudotime analysis is widely employed in the field of scRNA-seq and is used to study the temporal patterns of gene expression during cell development and dynamic processes. By constructing a topological map of cell development or a pseudotime axis, it becomes possible to model and infer the developmental trajectories of cells, thereby revealing crucial changes in gene expression and regulatory networks. In the study of early senescence processes, pseudotime analysis can be utilized to identify genes associated with senescence, uncovering their expression patterns and temporal characteristics during cell development. The application of this method contributes to a deeper understanding of the mechanisms underlying early fruit senescence and provides insights and targets for strategies and interventions aimed at fruit decay prevention.

Two pseudotime analyses were conducted. In the first pseudotime analysis, two cell types, EX and ME, were selected for computation. In the pseudotime analysis of all cells in EX and ME, we conducted GO and KEGG analysis of pseudotime-correlated genes (Supplementary Data 3) and identified significant enrichment in four functions: senescence, resistance, phenylpropanoid pathways, and ROS metabolism. This implies that these genes play a crucial role in the senescence trajectory. Therefore, four gene sets associated with these four functions were established, and 9,019 cells highly correlated with these genes were selected through gene set scoring. Subsequently, pseudotime analysis of these functionally defined cells clarified the differentiation trajectory of cells during the senescence process (see "Results" section). Thus, the first clustering analysis revealed the major functions of target cells, and the second clustering analysis, focusing on highly correlated cells with the identified functions, elucidated the cell differentiation trajectory. Pseudotime analysis has provided a clear timeline for fruit senescence. Initially, there is a reduction in the cells' antioxidant capacity, leading to the accumulation of reactive oxygen species (ROS), primarily occurring in the mesocarp cells. The signaling then propagates to the exocarp cells, where cells perceive stress and initiate the high-level expression of enzymes involved in the phenylpropanoid biosynthetic pathway. This promotes the synthesis of compounds like flavonoids, enhancing cell wall strength and increasing the exocarp cells' antioxidant capacity (see "Results" section). However, as the storage time prolongs, cells become unable to maintain their antioxidant capacity, resulting in a significant imbalance of ROS within the cells, pushing the cells into a senescent state. At this point, mesocarp cells start accumulating senescence marker proteins, such as HuSAG12.

The pseudotime heatmap demonstrated that *HuCMB1*, highly expressed in the late stages of senescence, already showed some expression in the earliest stages of senescence. Thus, while the GRNs suggested that *HuCMB1* was regulated by *HuSAG12* (see "Results" section), there were no reported interactions between the two. While some research had mentioned the WHIRLY protein's role in regulating gene expression related to defense and senescence by binding to gene promoters, and the Arabidopsis WHY1 (single-stranded DNA-binding protein) was found to accumulate in the cell nucleus, altering RNA polymerase II (RNAP II) and suppressing the transcription of senescence-related transcription factor WRKY53[42], the role of RNA polymerase II in plant cell senescence remained unclear. In a 2023 paper published in Nature, Debès et al. clarified the role of MED32 in regulating senescence in five model species, including humans and mice[43]. The pseudotime heatmap in this study showed that *HuMED32* began to express in the early stages of senescence, potentially providing evidence for MED32's involvement in senescence in plants (see "Results" section). A hypothesis was proposed that *HuCMB1* might have already begun to function in the early stages of pericarp cell senescence, potentially regulating *HuMED32* (HU08G00932, a mediator of RNA polymerase II transcription subunit 32), which affected

senescence speed and had not been previously recognized[43]. The pseudotime heatmap in this study also demonstrated that transcription factors WHY1/2 within the resistance-related genes not only exhibited high expression at the end of senescence but also showed a strong response in the early stages of senescence, offering theoretical support for WHY1/2's functions in repair, cell senescence, and resistance, as described in the literature[42,44,45].

There has been controversy surrounding the role of the senescence marker gene *SAG12* in senescence. In different plants such as Arabidopsis and rice, there are conflicting pieces of evidence regarding whether SAG12 positively or negatively regulates senescence, particularly leaf senescence[46,47]. This could be attributed to the fact that *SAG12* expression occurs in the mid to late stages of senescence, influenced by numerous genes expressed in the early stages of senescence, making its role in senescence complex and challenging to elucidate. The carboxyl-terminal region of the HuCMB1 protein can act as a transcription activation domain (see "Results" section). In this study, when silenced, the HuCMB1 protein significantly influences the senescence process of *H. undatus*, indicating its crucial role in the regulation of transcriptional activation in this context. In the silenced lines of *HuCMB1*, the expression of *HuSAG12* was significantly downregulated. However, phenotypically, the *HuCMB1*-silenced lines exhibited a markedly accelerated senescence. Therefore, we speculated that there are two possibilities: either *HuSAG12* is negatively correlated with senescence in *H. undatus*, or HuSAG12, as a late-response protein in senescence, is interfered with by early-response factors such as HuCMB1. Further investigations are needed to elucidate the exact role of SAG12 in fruit senescence and to understand the interplay between HuCMB1 and HuSAG12 in this context.

The well-recognized senescence marker gene *SAG12* was found to exhibit high expression in the later stages of senescence. If the senescence status of the fruit is only discovered at this point, it is already too late. In this study, pseudotime analysis has allowed the identification of early response genes, which may have a simpler and clearer role in senescence. If the early-response genes discovered in this study were used as senescence-reporting genes, it would be more effective for monitoring fruit senescence. Importantly, through pseudotime analysis, not only were genes reported to regulate senescence in animals discovered, such as *MED32*[43] and its upstream transcriptional regulator CMB1[47], but also some functionally unknown early-response senescence genes (HU05G00200, HU07G02057, etc.) were uncovered. The inferred gene expression pseudotime patterns provide valuable resources for identifying early regulatory factors and key genes involved in fruit senescence.

RNA velocity analysis provides an independent method to study differentiation dynamics. It calculates changes in mRNA abundance for each individual cell and uses these changes to predict its future transcriptional state[48]. In the UMAP cell state space, this analysis provided trajectory field vectors pointing toward differentiation directions for all cells, summarized into trajectories leading towards developmental directions under different treatment conditions (see "Results" section). Based on the results of RNA velocity analysis, we proposed a hypothesis that MCs (mature cells) in CK samples formed a trajectory source, sensing cell dehydration in the early storage period and responding to $H_2O_2$ signals, leading to the direction of RCs (resistant cells) in EX of CK samples. In the late storage period, they sensed significant cell dehydration and formed a trajectory source that received senescence signals leading to the direction of SCs (senescent cells) in the ME fraction of Post samples. This result aligned with the pseudotime tree results (see "Results" section).

While there are several theories about the mechanisms of senescence, such as the free radical senescence theory, telomere senescence theory, mtDNA mutation theory, and carbonyl toxicity senescence theory, the free radical senescence theory has received substantial experimental evidence. This theory suggests that the primary cause of

organism senescence is the oxidative damage inflicted by free radicals on cellular biomolecules (nucleic acids, lipids, proteins, etc.), leading to impaired cellular function, decline, or loss of cellular function, ultimately resulting in organismal senescence[2,49].

However, at what point does the excessive accumulation of these damaging ROS begin? Some studies suggest that the accumulation of ROS serves as a trigger for cell senescence[49,50]. Fruit, when subjected to oxidative damage, experiences a decline in physiological functions, with reduced resistance, and becomes more susceptible to pathogen infections[2]. Hydrogen peroxide, as a crucial signaling molecule, possesses dual physiological functions. Low concentrations of $H_2O_2$ and other ROS molecules in cells are likely induced by signals of cell senescence, which, through the MAPK pathway, amplify the senescence signals, activate the synthesis of antioxidants in the phenylpropanoid pathway, and actively counteract the senescence process in fruit. When ROS levels become excessively high, they trigger subsequent senescence-related physiological processes[2,49]. This current work further supported the role of $H_2O_2$ as a signaling molecule by observing the accumulation of trace amounts of $H_2O_2$ in the exocarp during fruit senescence under transmission electron microscopy (see "Results" section).

In comparison to the negative ion mode, compounds identified in the positive ion mode from both exocarp (EX) and mesocarp (ME) exhibited clearer temporal differences. With prolonged storage time, the metabolites in senescent samples diverged more from those in mature samples taken on day 0, with the greatest difference observed on day 9 (see "Results" section). When disregarding the time factor and categorizing all metabolites from different components into three groups (EX, ME, and EN), the inter-group difference analysis revealed a significant proportion of flavonoid compounds in both the differentially regulated metabolites between EX and EN, as well as between EX and ME (33.33% and 36.67%, respectively) (see "Results" section). Furthermore, some compounds with excellent antioxidant and other biological activities, not previously identified in *H. undatus*, were detected (Supplementary Data 8). Examples include flavonol compound Herbacetin, a natural flavonoid from flaxseed known for its antioxidant, anti-inflammatory, and anticancer activities[51,52]. Additionally, Esculin, a coumarin glucoside found in various plants, exhibited anti-inflammatory effects[53,54]. In contrast to EX, ME primarily contained amino acids, cofactors, and other primary metabolites, indicating that EX may be responsible for providing resistance, while ME maintains the normal physiological status of *H. undatus* during fruit senescence (see "Results" section). The results of metabolite changes from UPLC-MS/MS supported the single-cell transcriptome analysis in this study, providing evidence for unraveling the mechanism of *H. undatus* senescence.

Furthermore, researchers have proposed that antioxidant compounds can be defined as anti-senescence substances. They can selectively kill senescence cells or act as modifiers of the senescence phenotype by regulating the senescence phenotype[50,55]. In this context, early-response oxidative stress molecules associated with phenylpropanoids, which have antioxidant activity, can potentially serve as therapeutic targets for delaying cell senescence.

It has been reported that metabolites in the pericarp, including those involved in flavonoid biosynthesis, flavone and flavonol biosynthesis, and phenylpropanoid biosynthesis, vary during the maturation process in other species such as avocado[56,57]. These products may be produced at different stages of fruit maturation and were likely generated by different types of cells, representing distinct stages of fruit ripening[56,57]. The results of this study provide theoretical support for understanding the roles of different types of cells in fruit maturation and senescence.

The results of this study clearly showed the bimodal trend of flavonoid accumulation during the senescence process (see "Results" section). The significant impact of silencing the key transcription

factor HuCMB1, which responds early to senescence, on flavonoid synthesis was evident. *HuCYP75B1*, previously identified as a crucial gene for flavonoid synthesis in *H. undatus*[5], was significantly down-regulated in the HuCMB1-silenced lines in this work (see "Results" section), reinforcing the hypothesis that HuCMB1 likely accelerates fruit senescence by inhibiting flavonoid synthesis, thereby reducing resistance. On the other hand, the expression of HuERD6-2, a protein responding to stress such as dehydration, showed a slight upregulation in the HuCMB1-silenced lines, indicating that it is not regulated by CMB1. Instead, due to reduced fruit resistance, it experiences more pronounced stress stimulation, leading to increased expression.

This study proposed that flavonoids may serve as potential indicators of the senescence process. Flavonoids exhibit excellent ROS scavenging activity, and ROS is closely associated with senescence[52,58]. The anti-senescence effects of flavonoids in animals have been confirmed[59], and the correlation between changes in flavonoid levels and plant senescence processes has been suggested[60]. However, the bimodal pattern of flavonoids as a marker for fruit or even plant senescence has not been reported. This might be attributed to the single-cell transcriptome analysis in this study, which addressed the issue of concealed cellular heterogeneity, highlighting the correlation between flavonoid changes in the exocarp and senescence (see "Results" section). This provided insights into the study of the mechanisms underlying fruit and other plant tissue senescence.

The pseudotime analysis in this study revealed several transcription factors involved in the senescence process of *H. undatus*, including HuCMB1, HuARF5, and HuAP1-2. Similarly, in other fruit such as strawberries, scRNA-seq studies have uncovered transcription factors associated with the infection process of *Fragaria vesca* leaves by *Botrytis cinerea*, including WRKY75 and NAC042[8]. The transcription factor interaction network obtained in our work also included multiple WRKY family transcription factors, such as WRKY75 (see "Results" section). However, in the WRKY family, HuWRKY40 and HuWRKY53 played more central roles in the senescence process of *H. undatus*. Regarding the NAC family, HuNAC002 emerged as more crucial in the senescence of *H. undatus* (Supplementary Data 9). While the scRNA-seq of *Fragaria vesca* has revealed early response genes after infection with *Botrytis cinerea*[8], the senescence response genes identified in this study differ. The variations between the results of this work and the transcriptome of *Fragaria vesca* leaves during *Botrytis cinerea* infection could be attributed to species differences between *H. undatus* and *Fragaria vesca*, tissue differences between fruit and leaves, or distinctions between the processes of senescence and disease susceptibility.

Analyzing the temporal trajectory of post-harvest fruit cell differentiation is crucial for advancing research into the mechanisms of post-harvest preservation of fruit. This study found that the senescence processes of fruit exhibited a very clear timeline of ROS-resistance- senescence, and also identified a spatial transition of cell function along the mesocarp-exocarp-mesocarp. Transcription factors of the MADS family such as HuCMB1 were found to be highly expressed at different stages of senescence and in different types of peel cells, indicating their likely involvement in distinct biological processes of different cell types. While the MADS family has been previously reported to be involved in flower development, this study revealed their participation in fruit resistance and senescence, particularly by exploring the temporal trajectory of *HuCMB1* gene regulation in senescence, providing evidence for understanding the mechanism of fruit senescence. Thus far, preservation research has primarily focused on macroscopic environmental factors and signaling pathways post-harvest. However, the results of this study suggested that future fruit preservation strategies may involve controlling and utilizing the functionality of specific cell populations to achieve anti-senescence effects. This research will contribute to a better understanding of the roles of different types of fruit cells in the senescence process, thereby

promoting more in-depth research into post-harvest fruit senescence mechanisms at the single-cell level. In conclusion, single-cell studies on fruit are still in their early stages, and future research on a broader range of fruit species will provide more robust evidence for understanding the mechanisms underlying fruit senescence.

## Methods

### Preparation of mature and senescent samples

Fruit from the 35th days after artificial pollination (DAAP) of *H. undatus* (Vietnam No. 1) was collected from Miaoshui Base, Luoge Road, Ruyang County, Luoyang City, Henan Province, China. The harvested fruit were used as mature samples. They were stored under conditions of 25 °C and 85% relative humidity. During the storage period, the levels of superoxide anion and flavonoids in the exocarp were monitored every three days. The senescent sample was identified typically on the ninth day, characterized by the eruption of superoxide anions, the second peak in flavonoid levels, and the simultaneous appearance of pronounced dehydration-induced wrinkling or disease spots on the fruit phenotype.

### Sample collection and preparation of single-cell suspensions

According to the 10× Genomics user guide, strict requirements are imposed on the purity and concentration of native protoplasts. Single-cell preparation was modified from the procedure previously reported and described as follows[7]. In brief, mature and senescent pericarp of *H. undatus* were separately taken, and the tissues were digested in enzyme solution devoid of RNase (1% cellulase, 1% hemicellulase, 0.3% pectinase, 0.6 M mannitol, 100 mM MES, 20 mM KCl, 20 mM CaCl$_2$, and 10% bovine serum albumin) for 0.5 h at room temperature. The enzyme solution was passed through a 100 μm cell strainer, centrifuged, resuspended in an isotonic CWP solution (400 mM mannitol, 0.1% BSA), passed through a 40 μm cell strainer, and centrifuged to collect cell pellets, which were washed one to three times. Cell viability was observed and counted with a final concentration of 0.1x Trypan Blue staining. Cell concentration was adjusted to between 400 and 1200 cells/μl.

### scRNA-seq library construction

The suspension of *H. undatus* cells was loaded onto the 10× Genomics ChromiumTM system, resulting in the formation of single-cell microreactor systems encapsulated within oil droplets, known as GEMs (Gel Bead-in-Emulsion). All remaining steps, including library construction, were carried out according to the manufacturer's standard protocol (Chromium Single Cell 3′ v3.1)[7]. Quantification of the sequencing library was performed using a high-sensitivity DNA chip on the Bioanalyzer 2100 (Agilent) and the Qubit high-sensitivity DNA quantification instrument (Thermo Fisher Scientific). Sequencing of the library was performed on the NovaSeq6000 platform (Illumina), generating paired-end reads of 150 base pairs.

### scRNA-seq data processing

The reads were processed using the Cell Ranger 4.0 pipeline with default and recommended *parameters*. The Illumina sequencing outputs in FASTQ format were aligned to the *H. undatus* reference genome from the Pitaya Genome and Multiomics Database (PGMD) using the STAR algorithm[61]. Subsequently, UMI (Unique Molecular Identifier) counting and filtering of non-cell-related barcodes were performed to generate a gene barcode matrix for each individual sample. This resulted in the creation of a gene barcode matrix containing cell barcodes and gene expression counts. Next, this output was imported into the Seurat (v3.2.0) R package for quality control and downstream analysis of single-cell RNA sequencing data[62]. Unless otherwise specified, all functions were run using default parameters. Cells with a high proportion of mitochondria and cells containing dual cells in a droplet were filtered out. Standard panels with three quality criteria were used to filter low-quality cells and determine the number of detected transcripts (count of unique molecular identifiers), the number of detected genes, and the percentage of reads mapped to mitochondrial genes (filtered based on quartile threshold criteria). Normalized data (using the NormalizeData function in the Seurat package) were extracted for a subset of variable genes. In determining variable genes, we also controlled the close relationship between variability and mean expression levels.

### Cell clustering and annotation

After computing the shared nearest neighbor graph[62], we applied the Louvain Method[63] to perform graph-based clustering on the PCA-reduced data, thereby clustering the cells. For sub-clustering, the same scaling, dimensionality reduction, and clustering procedures were applied to specific datasets, typically limited to one cell type. The FindClusters function of seurat software was used to complete dimensionality reduction clustering analysis of cell sub-clusters[64,65]. For each cluster, we used the Wilcoxon Rank-Sum test to identify significantly differentially expressed genes when compared to the remaining clusters. SCINA[66] and known marker genes were employed for cell type identification. In this study, PCA[67] and UMAP[68] were used as dimensionality reduction algorithms.

### Differential expression analysis and functional enrichment

Differential gene expression analysis was performed on mature and senescent *H. undatus* pericarp using the "FindMarkers" function in Seurat. To annotate each cell cluster, we identified cluster-enriched genes with significantly higher expression levels in specific cell clusters compared to all other cell clusters. These genes needed to meet the following criteria: expression in the specific cell cluster in over 40% of cells, expression in the remaining cell clusters in less than 20% of cells, statistical significance ($q \leq 0.01$), and a high fold change ($\log_2 FC \geq 1$). Ranked in descending order based on $\log_2 FC$, the top three genes from each cluster were selected as marker genes. Dot plots of marker gene expression for each cluster were generated using the DotPlot function in Seurat (modified as needed). The same function, using the split.by parameter (modify as needed), was also used to visualize the expression of key genes in fresh and overripe *H. undatus* pericarp for each cluster. The AddModuleScore function was employed to score the target gene set. Based on the scoring results, cells highly correlated with this gene set were determined using quartile-based methods.

### Pseudotime analysis

Pseudotime trajectory analysis was conducted using the DDRTree algorithm within the Monocle[69] software package. The trajectory was visualized as a tree structure, including tips and branches. Log-normalized data from the Seurat object were imported into Monocle. Gene expression profiles in Monocle were used to identify genes with specific expression across cell groups and assess the statistical significance of these findings[70]. Cells were ordered along the trajectory and visualized in a reduced-dimensional space. Pseudotime analysis was performed on *H. undatus*-related genes, grouping genes with similar expression trend. Branch points were selected to determine genes contributing to developmental trajectory branches. Differential gene testing functions were used to filter for pseudotime-dependent or branch-dependent genes. Significantly branch-dependent genes were visualized using the plot-genes branched-heatmap function.

### RNA velocity analysis

As previously reported, RNA velocity was calculated based on spliced and unspliced counts and analyzed using cells present in the pseudotime ordering[48]. We estimated RNA velocity using scVelo (https://scvelo.org), a method for developmental trajectory analysis[71]. This estimated changes in RNA abundance over time by calculating the ratio of spliced to unspliced mRNA within individual cells and inferred

the likely direction of cell differentiation in the next steps. The velocity field was projected onto the pseudotime space generated by Monocle 2[72].

## Visualization of pseudotime-related gene regulatory networks

The top 100 genes along the Monocle 2 pseudotime branch points and transcription factors from the pseudotime-related differential genes were selected. The expression levels of these 100 branch point-related genes and 529 transcription factors was fitted to smooth spline curves using a Vector Generalized Linear and Additive Model (VGAM) to describe the gene expression trends along the pseudotime. Normalize pseudotime for each cell from 0 to 1. Gene Regulatory Networks (GRNs) were inferred using SCODE[13]. To obtain reliable relationships, we ran SCODE 50 times and averaged the results. The regulatory relationships between genes were exported to a file named "meanA.txt," where positive and negative values represent upregulation and downregulation, inferring activation and inhibition relationships, respectively. Load the arranged file containing source, target, SCODE values, and regulation columns into Cytoscape for visualization[65,73].

## Spatial transcriptomic (stRNA-seq) experiments

Like the CK samples in single-cell transcriptome, mature *H. undatus* peel was taken for spatial transcriptome sequencing. All *H. undatus* samples were embedded in cold OCT before cryosectioning. The tissue sections were placed on frozen Visium tissue optimization slides (10× Genomics) and Visium Spatial Gene Expression slides (10× Genomics). Subsequently, the tissue sections were fixed in cold methanol and stained according to the Visium Spatial Gene Expression User Guide (10× Genomics) or the Visium Spatial Tissue Optimization User Guide (10× Genomics). Tissue clearing was performed for 3 min based on the tissue optimization protocol[36,37].

Libraries were prepared following the Visium Spatial Gene Expression User Guide and loaded at a concentration of 300 pM onto the NovaSeq 6000 system (Illumina) with the NovaSeq S4 reagent kit (200 cycles, 20027466, Illumina) to achieve a sequencing depth of approximately 250 M read-pairs per sample or higher. The sequencing scheme for the samples was as follows: Read 1: 28 cycles; i7 index read, 10 cycles; i5 index read, 10 cycles; Read 2: 91 cycles.

For tissue sections attached to capture areas, fixation, histological staining, and tissue clearing were performed. After capturing transcripts with surface probes, reverse transcription was carried out overnight. Following surface cDNA synthesis, the tissue was removed, and spatial barcode mRNA-cDNA hybrids from each capture area were released from the array, collected into tubes, and subjected to library preparation. Subsequently, ST libraries were generated and sequenced on the Illumina platform.

## stRNA-seq data analysis

We processed, aligned, and summarized UMI counts for each spot on the Visium spatial transcriptomics chip using the Space Ranger software from 10× Genomics (version v. 1.2.2), with the reference genome being the [Pitaya Genome and Multiomics Database (PGMD)]. The original UMI count point matrix, images, point image coordinates, and scale factors were imported into R. The point matrix was filtered to retain only points that covered the tissue sections. Normalization of the raw UMI counts was performed using a negative binomial regression approach (SC Transform)[74]. To analyze high-resolution spatial RNA-seq data, unbiased and graph-based clustering of spatial features was performed using the Louvain method[63]. Visualization of clustering was achieved in a two-dimensional map using Uniform Manifold Approximation and Projection (UMAP)[75]. The FindMarkers function in Seurat was employed to identify Differentially Expressed Genes (DEGs) between two different samples or clusters using a likelihood ratio test. Essentially,

DEGs with $|\log_2 FC| > 0.25$ and $Q$-value $\leq 0.05$ were considered as significantly differentially expressed genes.

## Integration analysis of stRNA-seq and scRNA-seq

To compare clusters identified from single-cell sequencing and our stRNA-seq data, we employed the following analysis methods: SingleR[38], SciBet[39], RCTD[40], and CARD[41] to pinpoint highly correlated cell clusters. The annotation in SingleR was performed for each single cell independently[38]. First, a Spearman coefficient was calculated for single-cell expression with each of the samples in the reference dataset. The correlation analysis was performed only on variable genes in the reference dataset. Next, multiple correlation coefficients per cell type according to the named annotations of the reference dataset were aggregated to provide a single value per cell type per single cell. The SingleR open-source R package is maintained on GitHub R package and available from https://github.com/dviraran/SingleR. Code to reproduce the figures is also available in the GitHub repository. The Training-test split, cross-validation, and supervised cell type annotation by SciBet were performed as described by Li et al.[39]. SciBet selected marker genes using an entropy test and then assigned cells to their respective cell types using multimodal distribution models and maximum likelihood estimation applied using the default number of marker genes. All the functions mentioned above were implemented in the R package SciBet, which can be downloaded at http://scibet.cancer-pku.cn. An online version of SciBet is also available on this website, which is based on JavaScript. All codes used for benchmarks are available at https://github.com/PaulingLiu/scibet. In the heatmaps generated by SingleR and SciBet, clusters were grouped based on the correlation data between single-cell transcriptome and spatial transcriptome clusters. As for the method RCTD, we followed the guidelines on the RCTD GitHub repository: https://raw.githack.com/dmcable/spacexr/master/vignettes/spatial-transcriptomics.html. We set doublet_mode = 'full'[40]. As for the CARD, in scRNA-seq, B was denoted as the G by K cell type-specific expression matrix for the informative genes, where each element represented the mean expression level of an informative gene in a specific cell type. The expression matrix B was commonly referred to as the reference basis matrix. In the spatial transcriptomics data, X was denoted as the G by N gene expression matrix for the same set of informative genes measured on N spatial locations. V was denoted as the N by K cell type composition matrix, where each row of V represented the proportions of the K cell types on each spatial location. The objective was to estimate V given both X from the spatial transcriptomics data and B constructed from the scRNA-seq data[41].

## RNA fluorescence in situ hybridization (RNA-FISH)

After dehydration, the tissue blocks were embedded in paraffin wax and sectioned using a microtome (LEICA RM2016, Germany) into 5 μm thick slices. Deparaffinization was carried out twice in xylene, followed by dehydration in 100%, 95%, 80%, and 70% ethanol solutions. The samples were then fixed in 4% paraformaldehyde (RNA-free). Subsequently, a humid chamber was prepared using 5× SSC (pH 7.5) and formamide (1:1). The samples were treated with 30% $H_2O_2$ in pure methanol (1:9) for 10 min, followed by the addition of 0.25% hydrochloric acid and incubation at room temperature for 15 min. Proteinase K was applied to cover the samples, and the samples were incubated in a molecular hybridization oven at 37 °C for 20 min. A 0.1 M glycine wash solution was used for a 1-min wash to terminate the action of proteinase K. After each of these steps, the samples were rinsed 2–3 times with DEPC water for 1 min each time. The tissues were fixed with 4% paraformaldehyde (PFA) for 10 min. Probes were applied to cover the slices, followed by hybridization at 65 °C for 48 h. The slices were washed three times with 2× SSC buffer. Subsequently, staining with 1 μg/ml DAPI was performed for 5 min at room temperature. Images were obtained using a fluorescence upright microscope (Leica

DM2500, Germany), and fluorescence intensity analysis was conducted using ImageJ. The fluorescent oligonucleotide probe for HU08G02237 was synthesized by Gefan Biotechnology Co., Ltd. (China)[76]. The probe sequence was as follows: 5'-FAM-CAAAGCCAUC-CAUUGCUCGUGCUGAUCGACCAUGGAUGGCUCAUC -3'.

## Endogenous $H_2O_2$ localization by transmission electron microscopy

The $H_2O_2$ content in *H. undatus* pericarp was detected using histochemical methods[76–78]. Tissue blocks ($2 \times 2 \times 2$ mm) were excised from fresh and decaying *H. undatus* pericarp. These tissue blocks were suspended in an aqueous solution of 5 mM $CeCl_3$ (Sigma, UK) in glutaraldehyde buffer and allowed to incubate for 2 h at 28 °C to visualize $H_2O_2$ accumulation within the tissues. Subsequently, the samples were rinsed with phosphate buffer and dehydrated with a gradient of ethanol solutions (30%, 50%, 70%, and 80%). Afterwards, the samples were transferred to 90% and 95% acetone solutions and treated for 15 min, respectively. Finally, samples were treated twice with pure acetone for 20 min each time. The samples were then treated with a mixture of Spurr embedding agent and acetone (V/V = 1/1 and 3/1) for 1 and 3 h, respectively, and finally with pure embedding agent overnight. Thin sections (70–90 nm) of the samples were obtained using a LEICA EM UC7 ultramicrotome. Deposition of cerium peroxide was monitored using an H-7800 transmission electron microscope operating at 80 kV[76].

## Morphological microscopic observation of pericarp

Preliminary observations of fresh *H. undatus* pericarp was conducted using a Nikon Eclipse Ci optical microscope equipped with a calibrated eyepiece micrometer. Images were captured using Image-Pro Plus software[78], and analysis was performed to determine the sizes of epidermal and subcutaneous tissue cells, as well as the thickness of the epidermal and subcutaneous tissue layers.

## Flavonoid and ROS assessment

For the control group, fresh *H. undatus* pericarp was cut into $0.8 \times 1.8$ cm strips and placed in 10 mM fluorescent probe $H_2DCFDA$. Incubation was carried out in the dark at 25 °C for 0.5 h. Afterward, the samples were rinsed with PBS buffer. Subsequently, slices were taken from a position 0.3 cm from the outer edge of the strips and immediately mounted on temporary slides for fluorescence microscopy observation.

For the experimental group, *H. undatus* pericarp samples placed for 3, 6, and 9 days were used, following the same procedures as the control group. Pericarp of *H. undatus* without any treatment was prepared as temporary slides for observation as the negative control.

Fluorescence of ROS stained with $H_2DCFDA$ was measured using a fluorescence microscope (Leica DM2500, Germany) with an excitation wavelength of 488 nm and emission wavelengths between 515 and 540 nm. The detection of superoxide anions was performed using the hydroxylamine oxidation method[49]. After fully shaking the supernatant of the fruit peel homogenate, 65 mM PBS (pH = 7.8), and 10 mM hydroxylamine hydrochloride (V/V = 2/1/1) for 30 min, p-aminobenzene sulfonic acid (17 mM) and α-naphthylamine (7 mM) (V/V = 2/2) were added in the reaction system. The mixture was rapidly shaken and then reacted for 15 min. The absorbance was measured at 530 nm. Total flavonoid contents were determined using the aluminum chloride colorimetric method[79]. The flavonoid extracts or rutin standard solution (>98%; CAS: 153-18-4; 10–100 μg/mL) were mixed with $NaNO_2$ solution (94 mM) and incubated at 25 °C for 6 min. Al $(NO_3)_3$ solution (54 mM) was then added. The mixture was thoroughly mixed and allowed to stand for another 6 min. Subsequently, NaOH (435 mM) was added to each extract and incubated at room temperature for 10 min. The absorbance was measured at 510 nm.

## Metabolites detection by mass spectrometry

**Sample preparation and extraction.** The freeze-dried pericarp was crushed using a mixer mill (MM 400, Retsch) with a zirconia bead for 1.5 min at 30 Hz. 100 mg powder was weighted and extracted 3 h at 60 °C with 0.6 ml 70% aqueous methanol. Following centrifugation at 10, 000 g for 10 min, the extracts were filtrated (SCAA-104, 0.22 μm pore size) before ultraperformance liquid chromatography-tandem mass spectrometry (UPLC-MS/MS) analysis[44].

**UPLC and high-resolution MS settings.** Liquid chromatography separation was performed using a Shimadzu Nexera UPLC system (Shimadzu, Kyoto, Japan), configured in binary 30 A pumps, SIL- 30AC autosampler and a CTO-30AC column oven. All the components were eluted onto a Phenomenex Kinetix C18 column ($2.1 \times 100$ mm, 100 A, 1.7 μm) fitted with a C18 guard column (2.0 mm I.D. × 4.0 mm; Phenomenex Luna). The column oven was set at 50 °C, and the autosampler was cooled at 8 °C. The flow rate was 0.4 min/mL. The mobile phase A (MPA) was 1% ACN, 0.1% Formic acid (V/V), and the organic phase B (MPB) was 90% ACN, 1% formic acid. The gradient elution program was as follows: an isocratic elution of 10% MPB for the initial 1.0 min with column flow diverted to waste. Column flow was returned in line with MS followed by a linear gradient elution of 10–50% MPB from 1 to 8 min, and then followed by a linear gradient elution of 50–99% MPB from 8 to 10.5 min; after holding the composition of 90% MPB for the next 2.5 min, the column was returned to its starting conditions till the end of the gradient program at 15 min for column equilibration.

MS analysis was performed using an AB Sciex X500B mass spectrometer (Concord, Ontario, Canada), which operated in negative or positive ionization mode with a TurboV ion source and ESI TwinSpray electrode. The source conditions were set as follows: ion-spray voltage floating 5.5 kV (4.5 kV neg mode), declustering potential 40 V. collision energy 5 V, turbo spray temperature 500 °C, nebulizer gas (Gas 1) 50 psi, heater gas (Gas 2) 50 psi, and curtain gas 30 psi. Continuous recalibration was carried out every 1 h by injecting and analyzing the appropriate X500B calibration mix with the aid of the automated calibration delivery system. All the parameters were controlled and run by Sciex OS v3.0software (Sciex, Concord, Ontario, Canada). Data analysis was processed with MS-DIAL (Version 4.9.221218) (http://prime.psc.riken.jp/compms/msdial/main.html)[80].

The acquisition using SWATH consisted of a full scan, followed by a Q1 isolation strategy. The full scan covered a mass range of m/z 100-1000 with an accumulation time of 100 ms. Variable SWATH windows were developed based on test injections of pooled samples and optimized using the SWATH Variable Window Calculator v1.2 (Sciex). Dynamic collision energy with 5 V spread was used, and accumulation time set to 0.04 s, and mass range of windows acquired set to 50–1000 m/z.

## RNA isolation and first-strand cDNA synthesis

Total RNA was extracted from EX, ME, or EN samples of *H. undatus* pericarps using RNAprep Pure Micro Kit (DP432, TIANGEN, Beijing, China), followed by cDNA synthesis with the TransScript® One-Step gDNA Removal and cDNA Synthesis SuperMix kit (AH311-02, TransGen, Beijing, China)[4,5].

## Virus-induced gene silencing (VIGS) of *HuCMB1*

The fragment (178 bp) of *HuCMB1* was cloned from the *H. undatus* cDNA and ligated to the pTRV2 vector[81]. pTRV1, pTRV2, and pTRV2-HuCMB1 were transformed into the *A. tumefaciens* strain GV3101 (forward primer: 5'GCTCTAGAACGAGATCAAATTGGGAAGCC3', reward primer: 5'CGGGATCCTAAAAGCCACGATATTCTGA3'). The infection protocol was carried out according to Zhang et al.[82]. Pericarps of *H. undatus* were immersed in bacterial suspension with

OD600 of 0.6 and then cultivated at 25 °C. The experiments were performed in five biological replicates.

## Gene expression analysis by RT-qPCR

The quantitative real-time polymerase chain reaction (RT-qPCR) was performed as reported by Li et al.[4]. The gene *β-actin* of *H. undatus* was used as the internal control[83]. Method of $2^{-\Delta\Delta Ct}$ was employed to calculate the relative copy numbers of the genes[84,85]. All primers used for RT-qPCR were shown in Supplementary Data 10. Three biological replicates were performed.

## Statistical analysis

Perform statistical analysis using OriginPro 2021 (Version 9.8.0.200). The intergroup differences in weight loss rates between CK and CMB-silenced lines were analyzed using one-way analysis of variance (ANOVA). The differences in gene expression and flavonoid levels among different components at a single time point were assessed using paired-sample *t*-tests for significance analysis. Values are denoted as significant ($p < 0.05$) or highly significant ($p < 0.01$). Supervised projection to latent structure discriminant analysis (PLS-DA) was carried out to dissect the overall variance of metabolites and the composition differences of the samples. Metabolites were screened based on the combination of p (corr) and variable importance in the projection (VIP) values from the PLS-DA[5]. The two-sided Wilcoxon Rank Sum test was used in the statistical analysis of supplementary Data 1, 2, 4, and 10. Bonferroni was used for data adjustment, and the number of multiple comparisons depends on the number of all genes in the rds data ($n = 20357$ in supplementary Data 1, 4, and 10; $n = 14,623$ in supplementary Data 2). The *p*-values were calculated based on a one-sided hypergeometric model, and the *p*-adjust method (BH) was used in the GO and KEGG analysis in the supplementary Data 3 and 6. The two-sided *t*-test was used to obtain the *p*-values and the *p*-adjust method (BH) was used in the analysis of the metabolites in Supplementary Data 8. The parameters used in the Cytoscape plugin "MCODE" were shown as follows: Node Score Cutoff: 0.2; Haircut: true; Fluff: false; K-Core: 2; Max. Depth from Seed: 100 (Supplementary Data 9).

## Reporting summary

Further information on research design is available in the Nature Portfolio Reporting Summary linked to this article.

## Data availability

The scRNA-seq and stRNA-seq data used in this study have been deposited in the NCBI SRA database with BioProject numbers "PRJNA974579" and "PRJNA1002459 ". The data of transmission electron microscopy (TEM) have been deposited at the EBI bioimage archive with BioStudies accession number S-BSST1358. The metabolomics data have just been deposited to the EMBL-EBI MetaboLights database with the identifier MTBLS9688. The complete dataset will be accessed here https://www.ebi.ac.uk/metabolights/MTBLS9688. Source data are provided in this paper.

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

## Acknowledgements

We are grateful for the free online platform of Majorbio I-Sanger Cloud Platform (www.i-sanger.com). We would also like to acknowledge J.A. Imlay for discussion and comments on the draft; and W. Ma for technical assistance. This work was supported by the Australian Research Council Grant CE (No. 200100015, to R.H.) and the National Natural Science Foundation of China (No. 22136006; No. 21976073, to C.Y.Z.).

## Author contributions

Conceptualization: X.L. Investigation: B.R.L., X.Y.P., and J.F.Y. Methodology: S.B.G., P.M., and J.J.S. Writing—original draft: X.L., B.R.L., X.Y.P., and J.Y.J. Writing—editing: C.Y.Z. and R.H., with inputs from all authors. Visualization: X.L., and X.Y.P. Supervision: R.H. Funding acquisition: C.Y.Z. and R.H.

## Competing interests

The authors declare no competing interests.
