## [Peer Review File · Nature Communications]

Single-Cell and Spatial RNA Sequencing Reveal the Spatiotemporal Trajectories of Fruit SenescenceReviewer #1 (Remarks to the Author):

It's original and interesting to combining Single-cell RNA sequencing and spatial transcriptome together to elaborate the changes and mechanisms of fruit senescence, and which has significance for postharvest preservation of pitaya. However, still there are some problems bothering me, which are as below.

1. Reactive oxygen induced plant senescence is a common opinion documented by many researches. For pitaya have strong oxidation resistance with high betaine substances in its fruit peel, did authors have some new findings in pitaya fruit senescence process that is different to other plants?
2. The assignment of cellular identity of each cell cluster is fully speculative, without experimental data support. Some marker genes used for the classification of cell clusters were not specific expressed in corresponding cell clusters (Figure 1C). Besides, these marker genes were lack of reference supports, the functional annotations or gene names were also absent. The expression pattern of these marker genes in pitaya fruit pericarp and whether they are expressed in specific cell types require specific experimental evidence to verify.
3. Line 518-536: Lots of discussion were presented in the Results section, considering to move them to the Discussion section.
4. The discussion section is not the repeat description of results, and the whole essay needs to be rewritten, considering to discussion with new findings or debates to other reports.
5. The ordering of Figure 2 is chaotic. Figure 2B, the color of cell layers, cell clusters and names should be consistent. For example, background color of graphic symbol 'EN' should be blue with the same of 'cluster 5' and also the color of cells in the section.
6. For single-cell sequencing, 13 cell clusters were classified, but only 7 cell clusters were identified by spatial transcriptome. How to explain the cell clusters classified in Single-cell sequencing were inconsistently with which in spatial transcriptome.
7. Line 276-278: What data used here for re-clustering EX and ME cells? How to divide clusters 1 and 2 to CK subgroups and 0, 3, 4 to Post subgroups, what's your basis?
8. Line 219: Please explain the classification basis of cell clusters 2, 3, 5, 9 and 10, which cannot be interpreted from the MIA analysis results graph.
9. Line 231: As shown in Figure 2E, cell cluster 2 belongs to the same region as cell clusters 0, 1, 6, and 8, but they are divided into different cell types. Please explain the basis for this.
10. Front size in figures is too small.
11. Line 184-189, abbreviation word mentioned above should be used here for instead of exocarp, mesocarp...
12. Line 785: "for" was redundant.
13. Line 821:"trends" should be "trend".
14. Line 831:"estimates" should be "estimated".
15. Line 832: "infers" should be "inferred".
16. Line 844: "between" should be "among".
17. Line 845: "respectively" should be added to the end of "among".
18. Line 860: "a at" was redundant.
19. Line 885: "as" should be added to the end of "considered".

Reviewer #2 (Remarks to the Author):

In this manuscript entitled "Single-Cell and Spatial RNA Sequencing Reveal the Spatiotemporal Trajectories of Fruit Senescence", thorough integrating spatial transcriptomics and single-cell RNA sequencing, the authors constructed a single-cell expression atlas of the pericarp pitaya (*Hylocereus undatus*) and established cellular differentiation and gene expression trajectories during fruit senescence. Although the authors provide interesting data analysis results, most of them are descriptive and not rigorous enough to support the conclusions.

Major issues

1. Cell type annotation is the first important step in single cell RNA sequencing data analysis and the foundation for all subsequent analyses. To annotate the 13 clusters identified in single-cell transcriptomic data, the authors utilized three categories of five different methods (SciBet,

SingleR, MIA, RCTD, and CARD) to perform the association analysis between spatial transcriptomics and single cell transcriptomics. However, it is very confusing that the association analysis results by five methods are not consistent.

For examples, from line 207 to line 209, the authors reported "Based on analysis using SingleR (Single Cell Recognition) and SciBet (SingleCell Identifier Based on E-test), we observed that cell clusters 0, 1, 6, and 8 from the scRNA-seq data clustered together, suggesting that they belong to the ME component." However, Cluster 0 in SciBet method is highly associated with ENF, not the ME, though cluster 0 in Single R method is associated with ME (Figure 2D). In addition, in MIA method, all 13 clusters are highly associated with ENF and EX, but the authors classified cluster 2,3,5,9 as EX. Similarly, the authors assigned cluster 10 as VB, and cluster 5 as EX (in line 232), but the two clusters are highly associated with VB in Single R method (Figure 2D). Thus, the cell type annotation results are little confusing and the annotation basis is not rigorous enough to support the conclusion, and the authors should solve this problem.

2. From line 261 to line 273, in Figure 3B, the authors described the microscopic images of different layers of *H. undatus* pericarp in CK group and Post group. However, we find it difficult to obtain the effective information from the images. The authors should label different cell layers and mark the corresponding characteristics in these images.

3. In line 276, the authors re-clustered EX and ME cells, and got five distinct subgroups. But the authors did not perform cell type recognition of these subgroups. From line 286 to line 289, the authors described that "Based on the scRNA-seq profiles, the UMAP overview of the 13 clusters in CK and Post samples showed the expression localization of the top gene in each of the five subgroups (Fig. 3F). Among them, Hu08G02237 exhibited strong expression specificity in the mesocarp." However, this statement is confusing. In Figure 3F, heat map and UMAP only showed that Hu08G02237 is the marker gene of subcluster 0, not the mesocarp. In addition, I cannot get the significance of the re-clustering analysis.

4. The pseudotime analysis is confusing. The pseudotime analysis is usually performed in cell populations with developmental associations. In line 378, the authors selected 9019 cells from all cells based on gene set scoring results and performed subcluster analysis and developmental trajectories analysis. The 9019 cells contained multiple cell types and their developmental relationships are complicated. This analysis may be not scientific and meaningful.

Minor issues

5. More validated expression evidence for the marker genes of the identified cell types should be provided. The RNA FISH images in Figure 3G are not clear enough to understand.

6. The spatial transcriptomics was performed only on one pericarp section. Lacking the biological replicates.

7. The discussion part of this manuscript is similar to the result part.

8. The figure legend of Figure 5 is too simple.

9. The abbreviation that the first time appears should be explained. For example, in line 113, the CK and Post samples should be explained.

Reviewer #3 (Remarks to the Author):

This work is the first single cell-expression atlas of a fleshy fruit pericarp during the senescence process. There is a lack of cell type information in fleshy fruit studies: there is no information about specific fruit cells, specifically senescence-related ones, so far. The authors performed technical optimization for further analysis of tissues. Thus, the work meets the expected standards in the field (fruit ripening and senescence) giving specific and key information for the fruit senescence process.

Some points should be incorporated or clarified by the authors according to my review:

Introduction

-Check if the genus *Hylocereus* has changed to *Selenicereus*

-Why use pitaya fruit as a model for this pioneering study? There are other fruits that have been well studied in their maturation and senescence process, and besides the reasons said by the authors (lines 76-80), other major reasons should be stated.

Results

- Define CK and Post samples (the first time that are mentioned, line 113). For instance, give information about the days after anthesis and the main physiological characteristics of the fruit developmental stages used.
- Line 143: cluster 5 seems to belong to the Post group since it has 666 vs. 530 of the CK group according to Table S2.
- Line 148: How do you select the 38 marker genes? Maybe a brief explanation about it could contribute to the reading.
- Line 182: Are you sure Fig. S2C refers to the 5 distinct components?
- Is the clustering analysis (lines 207-216) according to higher transcriptional levels of the cell clusters? It should indicate the clustering criteria.
- Lines 225, 227: The figure S2F does not exist.
- Line 231: cell cluster 2 is in the ME not EX cells (Fig. 2E).
- Line 242: after storage? Indicate storage conditions (in Methods)
- Lines 281-282: indicate (and discuss) that a proportion of the EX component goes to Post samples (Fig. 3E).
- Line 463: "after 6 days" 6 days of what? Please briefly explain the conditions.
- Line 531: Fig. 4L ?
- Fig. 3 legend: in (F) indicate the meaning of identities (0, 1, 2, 3, 4)
- Fig. 5 legend: authors could add the meanings of the acronyms (EX, ME, EN(F),)

Discussion

- The discussion seems very technical: authors explain how the work was done but it lacks discussion with other works or previous information.
- Authors could include in the Discussion section the work of Bai et al. 2022, Hort. Res. (<https://doi.org/10.1093/hr/uhab055>) in terms of defense- and senescence-related genes (single cell atlas in Botrytis-infected strawberry leaves).
- Line 673: authors mention "initiate seed development" What results or references exist for that? Seed development starts before mesocarp senescence in most fleshy fruit.

Methods

- What is the origin of fruit samples? Do the authors know about the growing conditions of the pitaya fruit used? The authors should give more related information.
- Line 799: define "fresh" and "overripe" stages in biological or physiological terms (e.g., Days After Anthesis, etc).

(details)

- Line 432: change "Pentose" to "pentose"
- Line 751: change "hours" to "h"
- Line 860: delete "a" before "at"

Reviewer #4 (Remarks to the Author):

The authors used microarray-based spatial transcriptomics and single-cell RNA-sequencing to reveal tissue architecture in pericarp of *Hylocereus undatus*, constructed the single-cell expression atlas of the pericarp of pitaya, revealing exocarp and mesocarp cells undergoing the most significant changes during the fruit senescence process. The research is interesting, while some points should be taken in consideration.

1. Whether the ripening process of pitaya fruit is the same as tomato that has post-ripening stage. Is postharvest storage post-ripening or senescence? How to define senescence of pitaya fruit? What are the criteria or indicators of senescence for this fruit? The authors should give some explanation, as this is important for all the experiments in this article.
2. Line291: Validation experiments, one gene is not enough, and should take one gene every cell group for gene or RNA localization, of course, if possible, combine to do some fusion of fluorescent protein for cellular localization. If the authors only use one gene to do this, it should be studied in depth, such as using gene editing technology to verify whether the gene is functional. If gene editing is not possible, transient gene regulation techniques can be considered.

3. Line959: In the ROS detection experiment, ROS would erupt on the wound surface during fruit cutting. How to ensure that ROS detected were not caused by the wound?
4. Line91: "using the Cytoscape plugin SCODE". Correct citation (original references) should be given to software, databases, etc., and such issues should be examined in whole article.
5. Lines62-64: "Also to date, there have been no reports on the expression characteristics and variation patterns of different cell types during the senescence and decay processes of fruit." The authors should check carefully to give this statement, make sure that there was definitely no report in other fruit.
6. Discussion: The authors need add some discussion about currently available fruit single cell sequencing or spatiotemporal transcriptome researches. Therefore, it can better reflect the innovation of the author's research.

1 **List of Responses**

5 **REVIEWER COMMENTS**

7 Reviewer #1 (Remarks to the Author):

It's original and interesting to combining Single-cell RNA sequencing and spatial
transcriptome together to elaborate the changes and mechanisms of fruit senescence,
and which has significance for postharvest preservation of pitaya. However, still there
are some problems bothering me, which are as below.

1. Reactive oxygen induced plant senescence is a common opinion documented by
many researches. For pitaya have strong oxidation resistance with high betaine
substances in its fruit peel, did authors have some new findings in pitaya fruit
senescence process that is different to other plants?

**Response:** Based on the single-cell transcriptome analysis, we further investigated the
metabolic changes in the exocarp, mesocarp, and endocarp of *H. undatus* at four
different time points during the senescence process using ultraperformance liquid
chromatography tandem mass spectrometry (UPLC-MS/MS) analysis. A data-
independent acquisition (DIA) method, sequential window acquisition of all theoretical
fragment-ion spectra (SWATH) was used to analyze the data (Kang et al., 2020; Castro-
Moretti et al., 2023). By using this method, the instrument deterministically fragmented
all precursor ions within the predefined m/z range in a systematic and unbiased fashion.

The description of the results and discussion have been added to the Results and
Discussion sections as follows:

**Figure R1 Metabolic analyses of different components in the pericarp of *H. undatus*.**
(Added as Figure S5 in the manuscript)

(A). Classification of metabolites identified in the pericarp by HMDB. (B and C). PLS-DA analysis
of metabolites identified in different samples under positive and negative ion modes, respectively.
(D). Expression and VIP values of top 30 differentially expressed metabolites among the three
components (VIP ≥ 1.0), from left to right: Exocarp vs. Mesocarp, Mesocarp vs. Endocarp, Exocarp
vs. Endocarp. “*” and “**” mean $p < 0.05$ and $p < 0.01$ in the t-test, respectively. (E). Venn diagram
of overlapping differentially expressed metabolites among the three components. (F). KEGG
enrichment analysis of differentially expressed metabolites among the three components.

**Contents added in the “Results” section:**

A total of 3,015 metabolites were identified in both negative and positive ion
modes (Table S10). Among them, 697 metabolites were matched to the HMDB
database (fig. S5A), including 153 lipids and lipid molecules (21.95%), 135
organoheterocyclic compounds (19.37%), 94 phenylpropanoids, and polyketides
(13.49%) (Table S10). PLS-DA results showed that compounds identified in the
exocarp, mesocarp, and endocarp could be clearly distinguished in both positive and
negative ion modes (fig. S5BC).

To explore the differences in metabolites among different components, the top 30
metabolites were identified based on VIP > 1 and $p < 0.05$ (fig. S5D and Table S10).
The Venn diagram illustrated the differences in metabolites among the three main
components—exocarp (EX), mesocarp (ME), and endocarp (EN)—with no compound
overlapping among the differentially regulated metabolites in all three components (fig.

S5E). KEGG enrichment analysis results revealed the pathways involved in the
metabolites of these three components. The differentially regulated metabolites in the
exocarp component were mainly enriched in the "Phenylpropanoid biosynthesis"
pathway compared to the other two components, while metabolites in the mesocarp
component were mainly enriched in pathways such as "Biosynthesis of cofactors,"
"Arginine biosynthesis," and "Histidine metabolism" (fig. S5F).

62 **Contents added in the “Discussion” section:**

In comparison to the negative ion mode, compounds identified in the positive ion
mode from both exocarp (EX) and mesocarp (ME) exhibited clearer temporal
differences. With prolonged storage time, the metabolites in senescent samples diverged
more from those in mature samples taken on day 0, with the greatest difference
observed on day 9 (fig. S5B). When disregarding the time factor and categorizing all
metabolites from different components into three groups (EX, ME, and EN), the inter-
group difference analysis revealed a significant proportion of flavonoid compounds in
both the differentially regulated metabolites between EX and EN, as well as between
EX and ME (33.33% and 36.67%, respectively) (fig. S5D). Furthermore, some
compounds with excellent antioxidant and other biological activities, not previously
identified in *H. undatus*, were detected (Table S10). Examples include the flavonol ,
Herbacetin, a natural flavonoid from flaxseed known for its antioxidant, anti-
inflammatory, and anticancer activities. Additionally, Esculin, a coumarin glucoside
found in various plants, exhibited anti-inflammatory effects. In contrast to EX, ME
primarily contained amino acids, cofactors, and other primary metabolites, indicating
that EX may be responsible for providing resistance, while ME maintains the normal
physiological status of *H. undatus* during fruit senescence (fig. S5F). The results of
metabolite changes from UPLC-MS/MS supported the single-cell transcriptome
analysis in this study, providing new evidence for unraveling the mechanism of *H.*
*undatus* senescence.

84 **References**

- [1] Kang, D., et al. Comparative analysis of constitutes and metabolites for traditional
Chinese medicine using IDA and SWATH data acquisition modes on LC-Q-TOF
MS. *Journal of pharmaceutical analysis*, **10**, 588-596 (2020). doi:
10.1016/j.jpha.2019.11.005
- [2] Castro-Moretti, F. R., et al. A metabolomic platform to identify and quantify
polyphenols in coffee and related species using liquid chromatography mass
spectrometry. *Front. Plant Sci.* **13**, 1057645 (2023). doi:
10.3389/fpls.2022.1057645
- [3] Qiao, Y., et al. Herbacetin induces apoptosis in HepG2 cells: Involvements of ROS
and PI3K/Akt pathway. *Food Chem Toxicol.* **51**, 426-433 (2013). doi:
10.1016/j.fct.2012.09.036.
- [4] Yang, X. D., Chen, Z., Ye, L., Chen, J., & Yang, Y. Y. Esculin protects against
methionine choline-deficient diet-induced non-alcoholic steatohepatitis by
regulating the Sirt1/NF- κ B p65 pathway. *Pharm Biol.* **59**, 922-932 (2021). doi:
10.1080/13880209.2021.1945112.

2. The assignment of cellular identity of each cell cluster is fully speculative, without
experimental data support. Some marker genes used for the classification of cell clusters
were not specific expressed in corresponding cell clusters (Figure 1C). Besides, these

marker genes were lack of reference supports, the functional annotations or gene names
 were also absent. The expression pattern of these marker genes in pitaya fruit pericarp
 and whether they are expressed in specific cell types require specific experimental
 evidence to verify.

Response: We really appreciated your comments. Yes, we can only infer the gene's
 specificity in various cell clusters based on its expression levels within each cluster. In
 this study, we initially validated the senescence marker gene *HuSAG12*'s specific
 expression in the mesocarp through RNA-FISH. However, recognizing the limitations
 of this experiment, we supplemented our findings with RT-qPCR. Using RT-qPCR, we
 assessed the expression changes of four key senescence-related genes, including
 *HuSAG12*, *HuCMB1*, *HuMED32*, and *HuERD6-2*, at three time points during *H.*
 *undatus* senescence in the exocarp, mesocarp, and endocarp (Figure R2A).

 The results of RT-qPCR indicated that the expression of the four genes showed
 specific localization in different components, consistent with the findings from single-
 cell transcriptomics (Table R1). The expression changes at different stages of
 senescence aligned with the pseudotime heatmap results from scRNA-seq (Fig. R2B).
 The description of the results was incorporated into the Results section, as follows:

 The expression of the gene *HuCMB1* was highest in the EX component and was
 significantly elevated in the mid and late stages of senescence ($p < 0.01$). The well-
 established senescence marker gene *HuSAG12* exhibited exceptionally high expression
 solely in the late senescence phase of the ME component ($p < 0.05$). *HuERD6-2*, on the
 other hand, showed high expression in the EX component specifically during the mid-
 senescence period ($p < 0.01$). The gene *HuMED32* demonstrated elevated expression
 in the late senescence phase of the EX component ($p < 0.05$) (Fig. R2AB and Fig. 5A,
 Table R1 and S12).

**Figure R2 Expression of senescence related genes** (Figure R2A was added as Figure 5A in the
 manuscript. Fig. R2B was shown in the Fig. 4I and S3D) **A**, Expression of four senescence-related
 genes at different components of pericarp and time points of senescence by RT-qPCR. * represents
 $p < 0.05$, and ** represents $p < 0.01$; **B**, Heatmap showed the expression of senescence related genes
 along a pseudotime progression.

**Table R1 Expression of senescence related hub genes in components of pericarp in scRNA-**
 **seq profiles.** (Shown as sheet 2 of the Table S12 in the manuscript)

Compon ents	Clust ers	Gene id	Gene Name	avg_Log 2FC	pct.1(%)	pct.2(%)	Signific ant	Regul ate
ME	1	HU08G02	HuSAG	1.200329	85.2	31.8	yes	up

		237	12	43				
	8			3.718588 37	98.5	35	yes	up
	0			- 1.025316 5	52.6	37	yes	down
	2			- 3.825551 1	11.2	43.2	yes	down
	3			- 2.194015 5	18.7	41.5	yes	down
EX	5			- 2.173106 1	21.6	41	yes	down
	7			- 2.234789 3	17.8	41.1	yes	down
	9			- 3.909085 5	4.3	40.5	yes	down
	4			- 2.201000 1	15.9	41.7	yes	down
EN	11			- 0.898904 9	23.6	39.5	yes	down
VB	10			- 3.083416 5	12	40	yes	down
	1			-0.51863	12.3	20.5	yes	down
	6			- 0.521777 5	13.4	19.9	yes	down
ME	8			- 0.597675 9	13	19.8	yes	down
	7	HU05G02 127	HuERD 6-2	0.968398 2	38.3	17.7	yes	up
EX	3			- 0.407859 5	9.4	20.4	yes	down
	5			- 0.462027 8	11.5	20.1	yes	down
EN	4			1.286084 3	40.7	17.1	yes	up
	0			0.504605 14	83.7	57.6	yes	up
ME	1	HU10G00 469	HuCM B1	0.869305 3	81	58.3	yes	up
	6			0.487063 41	78.6	59.9	yes	up
	8			0.582794 52	83.5	59.9	yes	up

	3			-	0.628560	41.4	63.6	yes	down
	5			2	-0.273793	54.8	62.1	yes	down
EX	7			8	0.390901	55.1	62	yes	down
	9			1	1.411496	32.4	62.5	yes	down
VB	10			2	1.796187	22.1	62.6	yes	down
EN	4			4	1.827219	27.7	64.9	yes	down
	11				-2.382495	18.9	62.2	yes	down
EX	2	HU08G00 932	HuME D32		0.225712 06	12.5	8.1	yes	up

Furthermore, the current databases for plants are still incomplete, with some marker genes lacking functional annotations or gene names. We also conducted additional VIGS experiments. By silencing the senescence-related key gene *HuCMB1* identified in this study, we monitored the temporal changes in flavonoid levels in the pericarp of *H. undatus* after *HuCMB1* silencing. Combining phenotypic changes, we obtained preliminary validation of the function of the *HuCMB1* gene. Furthermore, through RT-qPCR experiments, we confirmed that *HuMED32* was regulated by *HuCMB1* and plays a crucial regulatory role in senescence. For a detailed description and analysis of the specific results, please refer to the main text of the manuscript.

Figure R3 Silencing of *HuCMB1* led to faster senescence of *H. undatus*. (Figure R3 was added as Figure 5 in the manuscript)

**A**, Expression of four senescence-related genes at different components of pericarp and time points
of senescence. * represents $p < 0.05$, and ** represents $p < 0.01$; **B**, Schematic showing the domain
structure of HuCMB1; **C-E**, Changes in fruit phenotype (C), weight loss rate (D), and flavonoid
levels in the exocarp (E) after silencing the HuCMB1 gene. **F**, RT-qPCR was used to analyze the
expression of *HuCMB1*, *HuERD6-2*, *HuSAG12*, and *HuMED32* in *H. undatus* after 9 days of storage
under *HuCMB1* gene silencing.

3. Line 518-536: Lots of discussion were presented in the Results section, considering
to move them to the Discussion section.

**Response:** Yes, we agree with your perspective. The contents that should not be in
the Results section but belongs in the Discussion and we it has been moved accordingly.
This includes discussions on marker genes and the key genes identified through GRN
screening.

4. The discussion section is not the repeat description of results, and the whole essay
needs to be rewritten, considering to discussion with new findings or debates to other
reports.

**Response:** Yes, we agree with your suggestions. The previous Discussion section was
not sufficiently in-depth and lacked comparisons and analyses with relevant
preliminary results. The entire article has undergone substantial rewriting, with the
following key changes:

- 1) An introduction to the gene set scoring analysis method has been added, along with
corresponding references.
- 2) The analysis approach for the pseudotime section has been explained in more detail,
clarifying the complex ideas behind the two pseudotime analyses for better reader
comprehension.
- 3) The discussion section now includes additional information on the metabolite
content of each component detected by mass spectrometry. This covers differences
in metabolites among components and comparative analyses with known
metabolites in *H. undatus*.
- 4) Comparative analyses with single-cell analysis results of *Fragaria vesca* infection
with *Botrytis cinerea*, as found in the literature, have been incorporated.
- 5) A summary of currently available single-cell sequencing and spatial
transcriptomics technologies has been added. The achievements in the plant field
using these technologies have been discussed and compared with our work.
- 6) VIGS and RT-qPCR experiments have been included, providing preliminary
evidence that HuCMB1 is involved in the fruit senescence process and can regulate
flavonoid synthesis.
- 7) Combining the background from previous research and the results of this study,
the discussion section now includes a comparative analysis of the impact of
flavonoids on senescence.

5. The ordering of Figure 2 is chaotic. Figure 2B, the color of cell layers, cell clusters
and names should be consistent. For example, background color of graphic symbol 'EN'
should be blue with the same of 'cluster 5' and also the color of cells in the section.

**Response:** We have reorganized Figure 2 in the manuscript to ensure a neat order.
Regarding Figure 2B, we initially inverted the colors corresponding to cell clusters in

the legend for better color contrast. We agree with your suggestion that, for better reader understanding, the colors of cell clusters and the legend should correspond. We have made the necessary adjustments (Figure R4). Additionally, considering your 8th comment, we have removed the MIA image. Please refer to the response below your 8th comment for specific explanations. Thanks again.

Figure R4 Reconstruction of a cellular atlas for mature pericarp of *H. undatus* using spatial transcriptomics. (Shown as revised Figure 2 in the manuscript)

(A). Workflow for sampling and sequencing *H. undatus* pericarp on the 10X Visium platform. (B). Illustrations of cell types discovered on glass slides, overlaid on corresponding H&E-stained images. Clusters are named based on the spatial positioning of cell types. (C). Spatial localization maps (left) and violin plots (right) showing the expression of marker genes for different cell types. (D). Association analysis between spatial transcriptomics and single-cell transcriptomics using four algorithms: SciBet, SingleR, RCTD, and CARD. (E). UMAP plot displaying 13 clusters of single cells classified into five cell types after spatial transcriptomics identification.

6. For single-cell sequencing, 13 cell clusters were classified, but only 7 cell clusters were identified by spatial transcriptome. How to explain the cell clusters classified in Single-cell sequencing were inconsistently with which in spatial transcriptome.

Response: Unlike single-cell transcriptomics, which can detect the gene expression

profile of each cell, the current 10× Visium spatial technology is based on barcode transcripts for RNA sequencing. This preserves the original transcripts and enables spatial BCR repertoire analysis, allowing us to detect gene expression levels while obtaining positional information on gene expression within tissues. Each capture area has an area of 6.5 mm × 6.5 mm, with approximately 5,000 spots, and each spot may contain more than one cell. Compared to single-cell transcriptomics, spatial transcriptomics has a slightly lower resolution. The main reason for the difference in the number of cell clusters obtained by the two technologies is this lower resolution. Therefore, we employed algorithms such as SingleR, SciBet, CARD, and RCTD to establish the association between the two transcriptome datasets. This approach allows us to reveal the spatial positioning of individual cell clusters and analyze the transcription profiles of various cell types at the single-cell level.

7. Line 276-278: What data used here for re-clustering EX and ME cells? How to divide clusters 1 and 2 to CK subgroups and 0, 3, 4 to Post subgroups, what's your basis?

第 276-278 行: 这里使用什么数据来重新聚类 EX 和 ME 单元? 如何将簇 1 和 2 划分为 CK 子组, 将 0、3、4 划分为 Post 子组, 您的依据是什么?

Response: After completing the visualization of cell types (Figure 2E), we created cell sets for the five cell types, namely EX cell set, ME cell set, EN cell set, etc. Subsequently, we selected the EX and ME cell sets for subpopulation analysis.

The dimensionality reduction clustering analysis of cell subpopulations utilized the FindClusters function of the Seurat software (Chen et al., 2020; Zhang et al., 2021) to cluster specified cell groups into five subclusters (Table S5). Single-cell sequencing technology marks cells with barcodes, where each cell has its unique barcode (a nucleotide sequence). Through barcode information, we can trace each cell back to its origin, distinguishing whether the cell is from the CK or Post sample (specific barcode information is provided in Table R2). The cell count statistics resulted in the findings shown in Figure 3C, indicating that cells from subclusters 1 and 2 mainly originated from the CK group, while cells from subclusters 0, 3, and 4 predominantly originated from the post group (Table S5).

Table S5 Information about the cellular sources for 5 cell subclusters.

Subcluster	0	1	2	3	4	Sum
CK	0	3892	386	2	2	4282
Post	6848	8	0	264	46	7166

Table R2 Example of Specific Information for Barcode in these subclusters.

Barcode	Sample	Subcluster
Post_AAACCCAAGGCAGCTA-1	Post	0
Post_AAACCCACAATACCCA-1	Post	0
Post_AAACCCATCCGGCAAC-1	Post	0
Post_AAACCCATCGCAATGT-1	Post	0
Post_AAACGAAAGCTGCCAC-1	Post	0
Post_AAACGAACATGACACT-1	Post	0
Post_AAACGAAGTCCAAGAG-1	Post	0

Post_AAACGAAGTTGCGGCT-1	Post	0
Post_AAACGCTAGCGTCAAG-1	Post	0
Post_AAACGCTGTGCATCTA-1	Post	0
CK_GATGGAGGTTTACCTT-1	CK	2
CK_GATGTTGAGGCCTAGA-1	CK	2
CK_GATTCTTGTGTCATTG-1	CK	2
CK_GATTTCTGTTGCATGT-1	CK	2
CK_GCAGCTGCATCATGAC-1	CK	2
CK_GCAGGCTTCGTGGCTG-1	CK	2
CK_GCATTAGCAGTCGCAC-1	CK	2
CK_GCCGTGACACCAGCCA-1	CK	2
CK_GCGATCGCACTGAGGA-1	CK	2
CK_GGAACCCTCTTCTGTA-1	CK	2

References

[1] Chen Y. P. et al. Single-cell transcriptomics reveals regulators underlying immune cell diversity and immune subtypes associated with prognosis in nasopharyngeal carcinoma. *Cell Res.* **30**, 1024-1042. (2020). DOI: 10.1038/s41422-020-0374-x

[2] Zhang, T. Q., Chen, Y., Liu, Y., Lin, W. H. & Wang, J. W. Single-cell transcriptome atlas and chromatin accessibility landscape reveal differentiation trajectories in the rice root. *Nat. Commun.* **12**, 2053 (2021). DOI: 10.1038/s41467-021-22352-4

8. Line 219: Please explain the classification basis of cell clusters 2, 3, 5, 9 and 10, which cannot be interpreted from the MIA analysis results graph.

Response: MIA (Multimodal Integration Analysis) is an analytical method that integrates single-cell expression profiles with spatial transcriptomics data through multimodal analysis. It calculates the overlap between regionally differential genes in spatial transcriptomics and marker genes identified for cell types in single-cell transcriptomics. MIA employs hypergeometric distribution to infer the enrichment level of specific cell types in particular tissue regions, considering all genes as the background to calculate p-values. As shown in Figure 1C, the expression localization of marker genes is not absolute but relatively higher in certain cell clusters and lower in others. The regional differential genes in spatial transcriptomics exhibit a similar pattern. Due to the statistical principles involved, MIA's calculation results may be more accurate with larger sample sizes. However, given the limited number of samples in this study, the results of MIA may be unreliable and are not suitable for the analysis of this manuscript. The MIA results have been removed.

Currently, single-cell transcriptomics is a relatively new technology, and the number of papers utilizing spatial transcriptomics is limited. Therefore, algorithms suitable for joint analysis of the two omics are still in the early stages of development, with limited application. In the future, we are willing to explore the application areas of the MIA algorithm by analyzing it with more single-cell data, identifying its optimal use cases.

9. Line 231: As shown in Figure 2E, cell cluster 2 belongs to the same region as cell clusters 0, 1, 6, and 8, but they are divided into different cell types. Please explain the basis for this.

Response: I apologize for the misalignment of the labeled numbers in the figure. The cluster represented by the number 2 in the figure is located within the curve range of EX, but the labeled number 2 shifted to the ME area. We have corrected the label number 2 to its accurate position (Fig. R5). Thank you for bringing this to our attention.

Figure R5 Corrected UMAP plot (A) and corrected partition plot (B).

10. Font size in figures is too small.

Response: All the fonts in the figures have been adjusted as much as possible to make them large enough for readers to see clearly.

11. Line 184-189, abbreviation word mentioned above should be used here for instead of exocarp, mesocarp...

Response: Appreciated the reminder. These words have been replaced by the abbreviations mentioned above.

12. Line 785: "for" was redundant.

Response: The word "for" has been deleted.

13. Line 821: "trends" should be "trend".

Response: The word "trends" has been replaced by "trend".

14. Line 831: "estimates" should be "estimated".

Response: The word "estimates" has been replaced by "estimated".

15. Line 832: "infers" should be "inferred".

Response: The word "infers" has been replaced by "inferred".

16. Line 844: "between" should be "among".

Response: Although the network showed the regulatory relationship of many genes, each relationship was an interaction between two genes, so we prefer to use "between".

to express the interaction between two genes.

17. Line 845: “respectively” should be added to the end of “among”.

Response: The word “respectively” has been added to the end of this sentence.

18. Line 860: “a at” was redundant.

Response: Sorry, they are redundant. They have been deleted.

19. Line 885: “as” should be added to the end of “considered”.

Response: The word “as” has been added.

Reviewer #2 (Remarks to the Author):

In this manuscript entitled “Single-Cell and Spatial RNA Sequencing Reveal the
Spatiotemporal Trajectories of Fruit Senescence”, thorough integrating spatial
transcriptomics and single-cell RNA sequencing, the authors constructed a single-cell
expression atlas of the pericarp pitaya (*Hylocereus undatus*) and established cellular
differentiation and gene expression trajectories during fruit senescence. Although the
authors provide interesting data analysis results, most of them are descriptive and not
rigorous enough to support the conclusions.

Major issues

1. Cell type annotation is the first important step in single cell RNA sequencing data
analysis and the foundation for all subsequent analyses. To annotate the 13 clusters
identified in single-cell transcriptomic data, the authors utilized three categories of five
different methods (SciBet, SingleR, MIA, RCTD, and CARD) to perform the
association analysis between spatial transcriptomics and single cell transcriptomics.
However, it is very confusing that the association analysis results by five methods are
not consistent.

For examples, from line 207 to line 209, the authors reported “Based on analysis using
SingleR (Single Cell Recognition) and SciBet (SingleCell Identifier Based on E-
test), we observed that cell clusters 0, 1, 6, and 8 from the scRNA-seq data clustered
together, suggesting that they belong to the ME component.” However, Cluster 0 in
SciBet method is highly associated with ENF, not the ME, though cluster 0 in Single R
method is associated with ME (Figure 2D). In addition, in MIA method, all 13 clusters
are highly associated with ENF and EX, but the authors classified cluster 2,3,5,9 as EX.
Similarly, the authors assigned cluster 10 as VB, and cluster 5 as EX (in line 232), but
the two clusters are highly associated with VB in Single R method (Figure 2D). Thus,
the cell type annotation results are little confusing and the annotation basis is not
rigorous enough to support the conclusion, and the authors should solve this problem.

Response: Considering that the five methods belonged to three major categories of
algorithms, even if they fell under the category of deconvolution algorithms, such as
SingleR and SciBet, their calculation principles differed. Therefore, the results obtained
from these five algorithms might not have been identical. Regarding cluster 0, SingleR
showed a high correlation of cluster 0 with ME. In the results from SciBet, the
correlation of cluster 0 with ME was the second highest among the five components.
Additionally, the results from CARD and RCTD also indicated a high correlation of

cluster 0 with the ME component. Another factor for assessing the correlation between
clusters and components was the position of clusters in the UMAP plot, where clusters
belonging to the same component would have been close together. Therefore,
considering these factors, cluster 0 was assigned to the mesocarp (ME) component.

Regarding MIA, we agree with your opinions. MIA is an analytical method that
integrates single-cell expression profiles and spatial transcriptomic data through
multimodal analysis. It calculates the overlap between spatial transcriptomic region-
specific differentially expressed genes and marker genes for cell types identified in
single-cell transcriptomics. It uses the hypergeometric distribution to infer the
enrichment of specific cell types in particular tissue regions, and it calculates p-values
with all genes as the background. From Figure 1C, it could be observed that the
expression localization of marker genes was not absolute but relatively high in certain
cell clusters and lower in others. The situation was similar for region-specific
differentially expressed genes in spatial transcriptomics. Due to the statistical principles
involved, MIA's calculation results might have been more accurate with larger data
sample sizes. However, given the limited sample size in this study, the results from
MIA were considered unreliable and were excluded from the analysis. Currently,
single-cell transcriptomics is still a relatively new technology, with fewer publications
utilizing spatial transcriptomics. Therefore, algorithms suitable for joint analysis of two
omics data types are still in the early stages of development, and their applications are
not yet widespread. In the future, we are willing to explore the applicability of the MIA
algorithm by applying it to more single-cell data and identifying its optimal use cases.

2. From line 261 to line 273, in Figure 3B, the authors described the microscopic images
of different layers of *H. undatus* pericarp in CK group and Post group. However, we
find it difficult to obtain the effective information from the images. The authors should
label different cell layers and mark the corresponding characteristics in these images.

Response: Yes, annotating the various cellular layers helps readers understand the
information in the images. The various cellular layers of the CK and Post groups have
been labeled in Figure 3B. Descriptions of the characteristics of each layer have been
provided in the Results section.

**Figure R6 Optical microscope images of CK and Post samples** (Figure R6 was shown as revised
Figure 3B in the manuscript).

3. In line 276, the authors re-clustered EX and ME cells, and got five distinct subgroups.
But the authors did not perform cell type recognition of these subgroups. From line 286
to line 289, the authors described that “Based on the scRNA-seq profiles, the UMAP
overview of the 13 clusters in CK and Post samples showed the expression localization
of the top gene in each of the five subgroups (Fig. 3F). Among them, Hu08G02237
exhibited strong expression specificity in the mesocarp.” However, this statement is
confusing. In Figure 3F, heat map and UMAP only showed that Hu08G02237 is the
marker gene of subcluster 0, not the mesocarp. In addition, I cannot get the significance
of the re-clustering analysis.

**Response:** Really appreciated for your kindly comments. We apologize for not clearly
explaining our research approach. We conducted two pseudotime analyses. In the first
pseudotime analysis, we selected two cell types, EX and ME, for computation.
However, the pseudotime trajectory was not clear. In the results of the pseudotime
analysis for all EX and ME cells, we performed GO and KEGG analysis of pseudotime-
correlated genes (Figure S3C, Table S5) and identified significant enrichments in
senescence, resistance, phenylpropanoid pathway, and reactive oxygen species
metabolism. This implied that these genes play crucial roles in the senescence trajectory.
Thus, four gene sets related to these four functions were established. By gene set
scoring, 9,019 cells highly correlated with these genes were selected. Subsequently,
pseudotime analysis of these functionally clarified cells demonstrated a clearer
developmental trajectory during senescence (Figure 4B). Therefore, the first
pseudotime analysis revealed the major functions of target cells, while the second
pseudotime analysis, focusing on cells highly correlated with the identified functions,
revealed the cellular developmental trajectory.

Regarding the localization of Hu08G02237 in the subcluster, this conclusion was
derived from the cell tracing results of the subgroups described below. I apologize for
any confusion in our previous manuscript. The explanation of the tracing results and
relevant data tables has been included in the manuscript for better understanding.

For this subcluster analysis, only cells from the exocarp and mesocarp were
selected. We supplemented the basis for the cell sources in the subcluster analysis. The
cell subgroup dimensionality reduction clustering analysis in this study used the
FindClusters function of the Seurat software (Chen et al., 2020; Zhang et al., 2021),
clustering the specified cell groups into 5 subclusters. The traceability statistics of cell
numbers have been added to the supplementary materials of the manuscript as Table
S5. Single-cell sequencing technology marked cells using barcodes, with each cell
having its own barcode (a nucleotide sequence). Through barcode information, the
origin of each cell could be traced, indicating whether the cell was from the CK or Post
sample (specific barcode information is available in Table R3). The results of cell
counting were presented in Figure 3C, where cells in subclusters 1 and 2 primarily
originated from the CK group, and cells in subclusters 0, 3, and 4 primarily originated
from the Post group (Table S5).

Based on the traceability results from the aforementioned barcodes, 99.81% of cells
in subclusters 1 and 2 $[(3892+386)/(3892+386+8)]$ originated from the CK sample,
and 99.94% of cells in subclusters 0, 3, and 4 $[(6848+264+46)/(6848+264+46+2+2)]$
originated from the Post sample (Table S5). The UMAP results showed that cells from

CK and Post samples formed two major clusters, with CK cells clustering on the left
 and Post cells on the right.

For cells specifically from the mesocarp (ME), the traceability analysis showed a
 total of 5788 cells, with 5174 cells (89.39%) originating from the Post sample. The
 UMAP results indicated that Hu08G02237 exhibited specific expression in the Post
 sample, mainly located in the mesocarp. However, considering that there were also
 some exocarp cells in the Post sample, the expression specificity statement was
 modified to: Hu08G02237 exhibited expression specificity in the Post sample, mainly
 located in the mesocarp.

**Table S5 Information about the cellular sources for 5 cell subclusters of ME and EX.**

Subcluster	0	1	2	3	4	Sum
CK	0	3892	386	2	2	4282
Post	6848	8	0	264	46	7166

**Table R3 Example of Specific Information for Barcode in these subclusters.**

Barcode	Sample	Subcluster
Post_AAACCCAAGGCAGCTA-1	Post	0
Post_AAACCCACAATACCCA-1	Post	0
Post_AAACCCATCCGGCAAC-1	Post	0
Post_AAACCCATCGCAATGT-1	Post	0
Post_AAACGAAAGCTGCCAC-1	Post	0
Post_AAACGAACATGACACT-1	Post	0
Post_AAACGAAGTCCAAGAG-1	Post	0
Post_AAACGAAGTTGCGGCT-1	Post	0
Post_AAACGCTAGCGTCAAG-1	Post	0
Post_AAACGCTGTGCATCTA-1	Post	0
CK_GATGGAGGTTTACCTT-1	CK	2
CK_GATGTTGAGGCCTAGA-1	CK	2
CK_GATTCTTGTGTCATTG-1	CK	2
CK_GATTTCTGTTGCATGT-1	CK	2
CK_GCAGCTGCATCATGAC-1	CK	2
CK_GCAGGCTTCGTGGCTG-1	CK	2
CK_GCATTAGCAGTCGCAC-1	CK	2
CK_GCCGTGACACCAGCCA-1	CK	2
CK_GCGATCGCACTGAGGA-1	CK	2
CK_GGAACCCTCTTCTGTA-1	CK	2

**References**

- [1] Chen Y. P. et al. Single-cell transcriptomics reveals regulators underlying immune
 cell diversity and immune subtypes associated with prognosis in nasopharyngeal
 carcinoma. *Cell Res.* **30**, 1024-1042. (2020). DOI: 10.1038/s41422-020-0374-x
 [2] Zhang, T. Q., Chen, Y., Liu, Y., Lin, W. H. & Wang, J. W. Single-cell
 transcriptome atlas and chromatin accessibility landscape reveal differentiation
 trajectories in the rice root. *Nat. Commun.* **12**, 2053 (2021). DOI: 10.1038/s41467-

4. The pseudotime analysis is confusing. The pseudotime analysis is usually performed in cell populations with developmental associations. In line 378, the authors selected 9019 cells from all cells based on gene set scoring results and performed subcluster analysis and developmental trajectories analysis. The 9019 cells contained multiple cell types and their developmental relationships are complicated. This analysis may be not scientific and meaningful.

Response: Yes, we performed two pseudotime analyses, and we apologize for not clearly explaining the research approach. In summary, the first pseudotime analysis revealed the primary functions of the target cells. The second pseudotime analysis, focusing on cells with high correlation to the selected relevant functions, elucidated the trajectory of cell differentiation. The description of the approaches for the two pseudotime analyses has been added to the Discussion section for better reader understanding. The specific approaches are outlined as follows.

In the first pseudotime analysis, we selected the EX and ME cell types for computation. However, the pseudotime trajectory was not clear. Upon analyzing the pseudotime results for all EX and ME cells, we conducted GO and KEGG analyses for pseudotime-related genes, revealing significant enrichment in four functions: senescence, resistance, phenylpropanoid pathway, and reactive oxygen metabolism. This suggested a crucial role for these genes in the senescence trajectory. Subsequently, four gene sets were established, and through gene set scoring, we filtered out 9,019 cells highly correlated with these gene sets.

The majority of these 9,019 cells were EX and ME cells (Table R4). To ensure the completeness of the results, we retained other cell types highly correlated with the target gene set without deleting them. On the other hand, the selection of such highly correlated cells effectively removed cells unrelated or minimally related to the target gene set, resulting in a clearer pseudotime analysis of the cell differentiation trajectory.

In the second pseudotime analysis, based on the analysis results from gene set scoring, cells more relevant to the senescence process were selected, and irrelevant cells were excluded. The pseudotime analysis of the filtered 9,019 cells with high correlation to senescence provided a clearer differentiation trajectory (Fig. 4B). The relevant literature on gene set scoring has been included in the manuscript (Ren et al., 2021; Hao et al., 2021).

The early-response genes to senescence were identified through the pseudotime heatmap, representing the most crucial findings of this study. We supplemented the experiments with RT-qPCR and VIGS, using RT-qPCR to confirm the specific expression of key genes in selected components and the high expression period during the senescence process. The VIGS experiments provided preliminary validation of the functions of the selected senescence-related genes. This indicates the effectiveness of the pseudotime analysis results in this study. For images related to the VIGS results and RT-qPCR results for core senescence-related genes, please refer to the following (Fig. R8).

Table R4 Cell Count for each cluster in gene set scoring results

Components	Clusters	Num of cells high related with gene sets			
		Resistance	Senescence	Phenylpropanoids	ROS
	0	47	240	207	848
	1	92	540	56	536
ME	6	166	164	167	351
	8	55	573	35	188
Sum of ME cells		360	1517	465	1923
	2	640	89	847	442
	3	311	231	423	326
EX	5	161	212	203	126
	7	552	171	497	251
	9	121	69	271	30
Sum of EX cells		1785	772	2241	1175
Sum of ME and EX cells		2145	2289	2706	3098
Percent of EX and ME cells in all components		64.43%	68.76%	81.28%	93.06%
EN	4	958	738	535	212
	11	136	147	46	11
Sum of EN		1094	885	581	223
ENF	12	14	19	4	0
VB	10	76	136	38	8

References

[1] Ren, X., et al. COVID-19 immune features revealed by a large-scale single-cell transcriptome atlas. *Cell*. **184**, 1895-1913 (2021). doi: 10.1016/j.cell.2021.01.053

[2] Hao, Y. H., et al. Integrated analysis of multimodal single-cell data. *Cell*. **184**, 3573-3587.e29 (2021). doi: 10.1016/j.cell.2021.04.048.

Minor issues

5. More validated expression evidence for the marker genes of the identified cell types should be provided. The RNA FISH images in Figure 3G are not clear enough to understand.

Response: Really appreciated for your kindly comments. We agree with your perspective. In Figure 3G, the bright-field microscopy images serve as the foundation for understanding the spatial localization of gene expression in RNA-FISH images. We have added labels for the various cellular layers in the images, allowing readers to comprehend the stained cell nuclei in both DAPI and FAM-stained images, as well as the positioning of RNA for marker genes.

**Figure R7. RNA FISH indicated that the predominant location of *HuSAG12* was in the**
 **mesocarp.** (Shown as revised Figure 3G in the manuscript)
 Components of EX, ME, and VB were labelled in the light field image. *HuSAG12* probes were
 labeled with FAM (green). Nuclei were stained with DAPI (blue). Scale bar: 40 μ m.

 Based on the current FAM staining results, we can observe the expression
 localization of *HuSAG12* RNA, although the effectiveness is not optimal. To further
 validate the gene expression localization, we employed RT-qPCR to confirm the
 expression changes of the four key genes related to senescence that were identified in
 this study during the senescence process (Fig. R8).

 The results from RT-qPCR demonstrated that the specific localization of the four
 genes in different components aligned with the findings from single-cell
 transcriptomics (Fig. R8A and Table R5). Moreover, the expression changes at different
 stages of senescence were consistent with the pseudotime heatmap results from scRNA-
 seq (Fig. R8A and B). The description of the results has been integrated into the Results
 section, as outlined below:

 The expression of the gene *HuCMB1* was highest in the EX component and
 significantly elevated in the mid and late stages of senescence ($p < 0.01$). The well-
 established senescence marker gene *HuSAG12* exhibited exceptionally high expression
 solely in the late senescence phase of the ME component ($p < 0.05$). *HuERD6-2*, on the
 other hand, showed high expression in the EX component specifically during the mid-
 senescence period ($p < 0.01$). The gene *HuMED32* demonstrated elevated expression
 in the late senescence phase of the EX component ($p < 0.05$) (Fig. R8AB and Fig. 5A,
 Table R5 and S12).

Figure R8 Expression of senescence related genes (Figure R8A was added as Figure 5A in the manuscript. Figure R8B was shown in the Fig. 4I and S3D). **A**, Expression of four senescence-related genes at different components of pericarp and time points of senescence by RT-qPCR. * represents $p < 0.05$, and ** represents $p < 0.01$; **B**, Heatmap showed the expression of senescence related genes along a pseudotime progression.

Table R5 Expression of senescence related hub genes in components of pericarp in scRNA-seq profiles. (Shown as sheet 2 of the Table S12 in the manuscript)

Components	Clusters	Gene id	Gene Name	avg_Log2FC	pct.1(%)	pct.2(%)	Significant	Regulate
ME	1			1.200329 43	85.2	31.8	yes	up
	8			3.718588 37	98.5	35	yes	up
	0			- 1.025316 5	52.6	37	yes	down
EX	2			- 3.825551 1	11.2	43.2	yes	down
	3			- 2.194015 5	18.7	41.5	yes	down
	5	HU08G02 237	HuSAG 12	- 2.173106 1	21.6	41	yes	down
	7			- 2.234789 3	17.8	41.1	yes	down
EN	9			- 3.909085 5	4.3	40.5	yes	down
	4			- 2.201000 1	15.9	41.7	yes	down
VB	11			- 0.898904 9	23.6	39.5	yes	down
	10			- 3.083416 5	12	40	yes	down
ME	1	HU05G02	HuERD	-0.51863	12.3	20.5	yes	down

	6	127	6-2	- 0.521777 5	13.4	19.9	yes	down
	8			- 0.597675 9	13	19.8	yes	down
EX	7			0.968398 2	38.3	17.7	yes	up
	3			- 0.407859 5	9.4	20.4	yes	down
	5			- 0.462027 8	11.5	20.1	yes	down
EN	4			1.286084 3	40.7	17.1	yes	up
ME	0			0.504605 14	83.7	57.6	yes	up
	1			0.869305 3	81	58.3	yes	up
	6			0.487063 41	78.6	59.9	yes	up
	8			0.582794 52	83.5	59.9	yes	up
EX	3			- 0.628560 2	41.4	63.6	yes	down
	5	HU10G00	HuCM	-0.273793	54.8	62.1	yes	down
	7	469	B1	- 0.390901 8	55.1	62	yes	down
	9			- 1.411496 1	32.4	62.5	yes	down
VB	10			- 1.796187 2	22.1	62.6	yes	down
EN	4			- 1.827219 4	27.7	64.9	yes	down
	11			-2.382495	18.9	62.2	yes	down
EX	2	HU08G00	HuME	0.225712 06	12.5	8.1	yes	up
		932	D32					

In this study, early response genes were identified through the pseudotime heatmap. We conducted additional VIGS experiments to provide preliminary validation of the functions of the selected senescence-related genes (Fig. R9). By silencing the RNA of the identified key senescence gene, *HuCMB1*, through VIGS, we monitored the temporal changes in flavonoid levels in *H. undatus* pericarp after *HuCMB1* silencing. Combining these results with phenotypic changes, we achieved a preliminary validation of the functionality of the *HuCMB1* gene. Additionally, through RT-qPCR experiments, we confirmed that *HuMED32* was significantly regulated by *HuCMB1*, playing a crucial regulatory role in senescence (Fig. R9 and Fig. 5). For a detailed description and analysis of the results, please refer to the main text of the manuscript.

Figure R9 Silencing of *HuCMB1* led to faster senescence of *H. undatus*. (Figure R9 was added as Figure 5 in the manuscript)

A, Expression of four senescence-related genes at different components of pericarp and time points of senescence by RT-qPCR. * represents $p < 0.05$, and ** represents $p < 0.01$; **B**, Schematic showing the domain structure of *HuCMB1*; **C-E**, Changes in fruit phenotype (**C**), weight loss rate (**D**), and flavonoid levels in the exocarp (**E**) after silencing the *HuCMB1* gene. **F**, RT-qPCR was used to analyze the expression of *HuCMB1*, *HuERD6-2*, *HuSAG12*, and *HuMED32* in *H. undatus* after 9 days of storage under *HuCMB1* gene silencing.

6. The spatial transcriptomics was performed only on one pericarp section. Lacking the biological replicates.

Response: Really appreciated for your kindly comments. Yes, we agree with your opinion. For biological experiments, having repeated samples is highly meaningful. Unfortunately, due to budget constraints, we only prepared a single pericarp slice. In our future work, we will strive to secure more funding support to use repeated samples for increased data accuracy.

Regarding the data in this study, the spatial transcriptome primarily provided spatial information, while the cellular transcriptome was based on the higher-resolution single-cell transcriptome sequencing data. The interplay between these two transcriptomes mutually reinforced and contributed to more reliable results in this study.

Furthermore, currently, spatial transcriptome data is generally considered stable and reliable. Some published articles utilizing spatial transcriptome technology have also employed single samples. For instance, Fu et al. analyzed the developmental process of maize kernels, using one sample each for 12 DAP and 24 DAP (Fu et al., 2023). Du et al. studied six developmental stages of poplar stem tissues, using one sample for each stage (Du et al., 2023). Li et al. examined the dynamic molecular map of cambium differentiation during primary and secondary growth in trees, using a single

sample (Li et al., 2023). Liu et al. investigated the spatiotemporal transcriptome of
orchids at different developmental stages, using one sample for each stage (Liu et al.,
2022a). Liu et al. explored cell heterogeneity in peanut tissues, using one sample for
each site (Liu et al., 2022b). The list of references is as follows:

**References**

- [1] Fu, Y., et al. Spatial transcriptomics uncover sucrose post-phloem transport during
maize kernel development. *Nat. Commun.* **14**, 7191 (2023). doi: 10.1038/s41467-
023-43006-7
- [2] Du, J., et al. High-resolution anatomical and spatial transcriptome analyses reveal
two types of meristematic cell pools within the secondary vascular tissue of poplar
stem. *Molecular Plant* **16**, 809-828 (2023). doi: 10.1016/j.molp.2023.03.005
- [3] Li, R., Wang, Z., Wang, J. W., & Li, L. Combining single-cell RNA sequencing
with spatial transcriptome analysis reveals dynamic molecular maps of cambium
differentiation in the primary and secondary growth of trees. *Plant Commun.* **4**,
100665 (2023). doi: 10.1016/j.xplc.2023.100665
- [4] Liu, C., et al. A spatiotemporal atlas of organogenesis in the development of orchid
flowers. *Nucleic Acids Res.* **50**, 9724-9737 (2022a). doi: 10.1093/nar/gkac773
- [5] Liu, Y., et al. Spatial transcriptome analysis on peanut tissues shed light on cell
heterogeneity of the peg. *Plant Biotechnol. J.* **20**, 1648-1650 (2022b). doi:
10.1111/pbi.13884

7. The discussion part of this manuscript is similar to the result part.

**Response:** Really appreciated for your comments. Yes, we also identified this issue. We
have moved the analytical and discussion content from the Results section to the
Discussion section, integrated it with the existing analysis and discussion, and
supplemented it with a comparison and analysis of previous research results. The
Discussion section has been extensively rewritten, and the general situation is as
follows:

- 1) Deleted the discussion on the specific expression of marker genes in each cluster
and moved it to the Discussion section.
- 2) Removed the discussion on core nodes like HuCMB1 and HuMED32 in the GRN
and moved it to the Discussion section.
- 3) The introduction to the gene set scoring analysis method has been added, along
with corresponding references.
- 4) The analysis approach for the pseudotime section has been explained in more detail,
clarifying the complex ideas behind the two pseudotime analyses for better reader
comprehension.
- 5) The discussion section now includes additional information on the metabolite
content of each component detected by mass spectrometry. This covers differences
in metabolites among components and comparative analyses with known
metabolites in *H. undatus*.
- 6) Comparative analyses with single-cell analysis results of *Fragaria vesca* infection
with *Botrytis cinerea*, as found in the literature, have been incorporated.
- 7) A summary of currently available single-cell sequencing and spatial
transcriptomics technologies has been added. The achievements in the plant field
using these technologies have been discussed and compared with our work.
- 8) VIGS and RT-qPCR experiments have been included, providing preliminary

evidence that HuCMB1 is involved in the fruit senescence process and can regulate
flavonoid synthesis.

9) Combining the background from previous research and the results of this study,
the discussion section now includes a comparative analysis of the impact of
flavonoids on senescence.

8. The figure legend of Figure 5 is too simple.

Response: The schematic diagram of Figure 5 has been explained in more detail.

9. The abbreviation that the first time appears should be explained. For example, in line
113, the CK and Post samples should be explained.

Response: The abbreviations have been annotated where they are first used in the
manuscript.

Reviewer #3 (Remarks to the Author):

This work is the first single cell-expression atlas of a fleshy fruit pericarp during the
senescence process. There is a lack of cell type information in fleshy fruit studies: there
is no information about specific fruit cells, specifically senescence-related ones, so far.
The authors performed technical optimization for further analysis of tissues. Thus, the
work meets the expected standards in the field (fruit ripening and senescence) giving
specific and key information for the fruit senescence process.

Some points should be incorporated or clarified by the authors according to my review:

Introduction

-Check if the genus *Hylocereus* has changed to *Selenicereus*

Response: Really appreciated for your kindly comments. Yes, the genus "*Hylocereus*"
has been updated to "*Selenicereus*." An explanation regarding the name of
"*Hylocereus*" and "*Selenicereus*" has been added when introducing the genus name for
the first time in the Introduction section. Considering that the genus name used for the
reference genome in the data analysis is "*Hylocereus*," and the gene names in both the
data deposited in NCBI and throughout the manuscript still use "Hu" as a prefix for
consistency with NCBI data, we have retained "Hu" as the gene prefix in this paper.

-Why use pitaya fruit as a model for this pioneering study? There are other fruits that
have been well studied in their maturation and senescence process, and besides the
reasons said by the authors (lines 76-80), other major reasons should be stated.

Response: In addition to the reasons mentioned in this paper, another factor is our
longstanding commitment to unraveling the mechanisms behind the delayed
senescence of fruits through the use of a novel superoxide anion scavenger—Trypsin.
Trypsin has demonstrated effective senescence-delaying effects on fruits and
vegetables such as pitaya, cucumber, and mango, with pitaya showing the most optimal
response. This is possibly attributed, at least in part, to the abundant antioxidants
present in pitaya, leading to a significant response to Trypsin. Therefore, pitaya was

chosen as the primary experimental material. Our exploration of the spatiotemporal
trajectory of gene expression patterns during the senescence process of pitaya not only
contributes to understanding its intrinsic senescence mechanisms but also provides a
scientific basis for the development of new fruit and vegetable preservatives.

Previously, considering that this paper did not delve into the content related to
Trypsin, we omitted the aforementioned reasons and solely expounded on the material
inherent to pitaya. In light of your comments, we acknowledge that the development of
preservatives is a topic of interest for researchers studying fruit senescence mechanisms.
Consequently, we have incorporated a brief introduction to the preliminary foundation
of Trypsin, aiming to offer new perspectives for researchers involved in the
development of innovative preservatives.

Our earlier drafts laid the groundwork for this paper. The list is provided below,
with references 3 and 4 already included in the manuscript's bibliography (references 4
and 5 in the manuscript's reference list).

**References**

- [1] Wang, J., Jia, J.Y., Sun, J.J., Pang, X.Y., Li, B.R., Yuan, J.F., Chen, E.Y., **Li, X.***.
Trypsin preservation: CsUGT91C1 regulates trilobatin biosynthesis in *Cucumis*
*sativus* during storage. *Plant Growth Regul.* **100**, 633-646 (2023). DOI:
10.1007/s10725-023-00962-w
[2] Wang, J., Tian, P.P., Sun, J.J., Li, B.R., Jia, J.Y., Yuan, J.F., **Li, X.***, Gu, S.B.* , Pang,
X.Y.*. CsMYC2 is involved in the regulation of phenylpropanoid biosynthesis
induced by trypsin in cucumber (*Cucumis sativus*) during storage. *Plant Physiol*
*Biochem.* **196**, 65-74 (2023). DOI: 10.1016/j.plaphy.2023.01.041.
[3] **Li, X.**, Zhang, Y.Y., Zhao, S.J., Li, B.R., Cai, L.N., Pang, X.Y. Omics analyses
indicate the routes of lignin related metabolites regulated by trypsin during storage
of pitaya (*Hylocereus undatus*). *Genomics* **113**, 3681-3695 (2021). DOI:
10.1016/j.ygeno.2021.08.005
[4] Pang, X.Y., Zhao, S.J., Zhang, M., Cai, L.N., Zhang, Y.Y., & **Li, X.***. Catechin
gallate acts as a key metabolite induced by trypsin in *Hylocereus undatus* during
storage indicated by omics. *Plant Physiol. Bioch.* **158**, 497-507 (2021). DOI:
10.1016/j.plaphy.2020.11.036
[5] **Li, X.**, et al. Transcriptomic analysis reveals hub genes and subnetworks related to
ROS metabolism in *Hylocereus undatus* through novel superoxide scavenger
trypsin treatment during storage. *BMC Genomics* **21**, 437 (2020). DOI:
10.1186/s12864-020-06850-1
[6] Pang, X.Y., Li, X.L., Liu, X.R., Cai, L.N., Li, B.R., & **Li, X.***. Transcriptomic
analysis reveals Cu/Zn SODs acting as hub genes of SODs in *Hylocereus undatus*
induced by trypsin during storage. *Antioxidants* **9**, 162. (2020). DOI:
10.3390/antiox9020162
[7] **Li, X.**, et al. Transcriptomic analysis reveals key genes related to antioxidant
mechanisms of *Hylocereus undatus* quality improving by trypsin during storage.
*Food Funct.* **10**, 8116-8128 (2019). DOI: 10.1039/c9fo00809h
[8] **Li, X.**, Zhong, Y.L., & Zhao, C.Y. Trypsin binding with copper ions scavenges

superoxide: Molecular dynamics-based mechanism investigation. *Int. J. Env. Res.*
*Pub. He.* **15**, 139-152 (2018a). DOI: 10.3390/ijerph15010139
[9] Li, X., et al. Trypsin slows the ageing of mice due to its novel superoxide
scavenging activity. *Appl. Biochem. Biotech.* **181**, 1549-1560 (2017). DOI:
10.1007/s12010-016-2301-7

Results

-Define CK and Post samples (the first time that are mentioned, line 113). For instance,
give information about the days after anthesis and the main physiological characteristics
of the fruit developmental stages used.

**Response:** Really appreciated for your kindly comments. Indeed, it is essential to
provide clear definitions for CK and Post samples. In the Results section, we have
incorporated a brief explanation, with detailed descriptions now included in the
Methods section. The relevant passage in the main text has been revised to "In the CK
(mature) and Post (senescent sample after storage, details see 'Methods' section)
samples, 80% and 77% of viable cells were retrieved, respectively."

The information regarding the key physiological characteristics has been added to
the Methods section, and the description is as follows:

Fruit from the 35th days after artificial pollination (DAAP) of *H. undatus* (Vietnam
No. 1) were collected from Miaoshui Base, Luoge Road, Ruyang County, Luoyang City,
Henan Province, China. The harvested fruit were used as mature samples. They were
stored under conditions of 25 °C and 85% relative humidity. During the storage period,
the levels of superoxide anion and flavonoids in the exocarp were monitored every three
817 days. The senescent sample was identified typically on the ninth day, characterized by
818 the eruption of superoxide anions, the second peak in flavonoid levels, and the
819 simultaneous appearance of pronounced dehydration-induced wrinkling or disease
spots on the fruit phenotype (Lines 1006-1014).

-Line 143: cluster 5 seems to belong to the Post group since it has 666 vs. 530 of the
CK group according to Table S2.

**Response:** Yes, we agree with your comments that the cell count of Cluster 5 of the
Post sample was higher than that in the CK sample. Corresponding modifications have
been made in the main text.

-Line 148: How do you select the 38 marker genes? Maybe a brief explanation about it
could contribute to the reading.

**Response:** Firstly, cluster-enriched genes with significantly higher expression levels in
specific cell clusters compared to all other cell clusters were identified. These genes
needed to meet the following criteria: expression in the specific cell cluster in over 40%
of cells, expression in the remaining cell clusters in less than 20% of cells, statistical
significance ($q \leq 0.01$), and a high fold change ($\log_2FC \geq 1$). Then, ranked in descending
order based on \log_2FC , the top three genes from each cluster were selected as marker
genes.

The selection criteria for cluster-enriched genes were described in the main text.
We apologize for the omission of the selection criteria for marker genes. This
information has been added to the “Methods” section (Lines 1080-1081).

-Line 182: Are you sure Fig. S2C refers to the 5 distinct components?

Response: We sincerely apologize for an error in our manuscript. The results to be
referenced at this point should be from Figure S2B, not Figure S2C. The “fig. S2C” has
been corrected to “fig. S2B” in this section.

-Is the clustering analysis (lines 207-216) according to higher transcriptional levels of
the cell clusters? It should indicate the clustering criteria.

Response: In the heatmaps generated by SingleR and SciBet, clusters were grouped
based on the correlation data between single-cell transcriptome and spatial
transcriptome clusters. Clusters with close correlations were clustered together. We
apologize for not clearly articulating this principle in the manuscript. The relevant
description has been added to the “Methods” section (Lines 1174-1176).

-Lines 225, 227: The figure S2F does not exist.

Response: We apologize for the inconvenience. The original figure S2B has been
deleted, figure S2 has been renumbered. The relevant details in the manuscript have
been corrected accordingly. Really appreciated for your kindly comments.

-Line 231: cell cluster 2 is in the ME not EX cells (Fig. 2E).

Response: We apologize for the offset in the annotated numbers in the figure. In Figure
2, the cluster represented by the number 2 is indeed located within the curve range of
EX, but the annotation mistakenly shifted to the ME region. We have rectified this issue
and adjusted the annotation for number 2 to the correct position (Fig. R10). Thank you
for bringing this to our attention.

**Figure R10 Corrected UMAP plot (A) and corrected partition plot (B).**

-Line 242: after storage? Indicate storage conditions (in Methods)

Response: The storage conditions for the samples have been included in the Methods
section. The relevant content is as follows: “They were stored under conditions of 25 °C
and 85% relative humidity” (Line 1009).

-Lines 281-282: indicate (and discuss) that a proportion of the EX component goes to
Post samples (Fig. 3E).

Response: Although ME and EX are the primary cell clusters in the Post and CK
samples, a small fraction of the EX components was certainly attributed to the Post
samples. This information has been added to the manuscript for the convenience of
readers' understanding.

-Line 463: “after 6 days” 6 days of what? Please briefly explain the conditions.

Response: The words “after 6 days” have been changed to “after 6 days of storage”.
The storage conditions for the samples have been included in the Methods section. The
relevant content is as follows: “They were stored under conditions of 25 °C and 85%
relative humidity” (Line 1009).

-Line 531: Fig. 4L ?

Response: I sincerely apologize for the error in the figure numbering. It has been
corrected to Fig. 4K at this location.

-Fig. 3 legend: in (F) indicate the meaning of identities (0, 1, 2, 3, 4)

Response: The UMAP plot in Figure 3F illustrated the expression localization of the
top 1 gene of subclusters 0, 1, 2, 3, 4 from the heatmap in each subcluster, consistent
with the distribution of CK and Post in Figure 3C.

In Figures 3C and D, it was observed that within these five subclusters (0, 1, 2, 3,
and 4), subclusters 1 and 2 belonged to the CK sample and were located in the left cell
aggregation of UMAP, while subclusters 0, 3, and 4 belonged to the Post sample and
were situated in the right cell aggregation of the UMAP plot. For clarity, labels for CK
and Post were added to the UMAP plot in Figure 3F to aid reader comprehension.

The relevant information has been added to the figure legend.

-Fig. 5 legend: authors could add the meanings of the acronyms (EX, ME, EN(F),)

Response: Thank you for your kindly comments. The meanings of abbreviations have
been added to the figure legend of Figure 5.

Discussion

-The discussion seems very technical: authors explain how the work was done but it
lacks discussion with other works or previous information.

-Authors could include in the Discussion section the work of Bai et al. 2022, Hort. Res.
(<https://doi.org/10.1093/hr/uhab055>) in terms of defense- and senescence-related genes
(single cell atlas in Botrytis-infected strawberry leaves).

Response: Yes, highly appreciated for your kindly comments. While supplementing the
analysis and discussion section, we also came across this important reference, which
provided valuable insights for our study.

Additionally, there is another literature on the analysis of lychee shoot apices
differentiation using single-cell techniques (Yang et al., 2023), which is also of

significant relevance to our study.

The references mentioned above have been added to the sections of “Introduction”
and “Discussion” of the manuscript, and the list is as follows:

**References**

[1] Bai, Y., Liu, H., Lyu, H., Su, L., Xiong, J. & Cheng, Z. M. Development of a single-
cell atlas for woodland strawberry (*Fragaria vesca*) leaves during early *Botrytis*
*cinerea* infection using single cell RNA-seq. *Hortic Res.* **9**, uhab055. (2022).
doi: 10.1093/hr/uhab055.

[2] Yang, M. C., Wu, Z. C., Chen, R. Y., Abbas, F., Hu, G. B., Huang, X. M., Guan, W.
S., Xu, Y. S. & Wang, H. C. Single-nucleus RNA sequencing and mRNA
hybridization indicate key bud events and LcFT1 and LcTFL1-2 mRNA
transportability during floral transition in litchi. *J Exp Bot.* **74**, 3613–3629. (2023).
doi: 10.1093/jxb/erad103.

-Line 673: authors mention “initiate seed development” What results or references exist
for that? Seed development starts before mesocarp senescence in most fleshy fruit.

**Response:** Really appreciated for your kindly comments. In this study, the proposition
regarding the initiation of seed development, in addition to the accumulation of
senescence-related proteins such as SAG12, was based on the results of pseudotime
analysis. The heatmap from the pseudotime analysis displayed elevated expression of
seed development-related gene *HuSUS* (gene ID HU02G00890) (Table S6).
Additionally, literature has reported the involvement of SAG12 in seed development
(James et al., 2018; Myat et al., 2022). However, we lack further experimental evidence
to substantiate the expression patterns of genes related to seed development. Therefore,
the mention of "initiate seed development" has been removed. In future studies, we
aspire to conduct more extensive research on seed development.

**References**

[1] James, M. et al. SAG₁₂, a major cysteine protease involved in nitrogen allocation
during senescence for seed production in *Arabidopsis thaliana*. *Plant Cell Physiol.*
**59**, 2052-2063 (2018).

[2] Myat, A. A. et al. Overexpression of GhKTI12 enhances seed yield and biomass
production in *nicotiana tabacum*. *Genes (Basel)* **13**, 426 (2022).

**Methods**

-What is the origin of fruit samples? Do the authors know about the growing conditions
of the pitaya fruit used? The authors should give more related information.

**Response:** We apologize for the oversight in not providing detailed information about
the samples. The relevant information has been added to the Methods section, and the
content is as follows:

**Preparation of mature and senescent samples**

Fruit from the 35th days after artificial pollination (DAAP) of *H. undatus* (Vietnam No.
1) were collected from Miaoshui Base, Luoge Road, Ruyang County, Luoyang City,
Henan Province, China. The harvested fruit were used as mature samples. They were
stored under conditions of 25 °C and 85% relative humidity. During the storage period,
the levels of superoxide anion and flavonoids in the exocarp were monitored every three

975 days. The senescent sample was identified typically on the ninth day, characterized by
976 the eruption of superoxide anions, the second peak in flavonoid levels, and the
977 simultaneous appearance of pronounced dehydration-induced wrinkling or disease
spots on the fruit phenotype (Lines 1006-1014).

-Line 799: define “fresh” and “overripe” stages in biological or physiological terms
(e.g., Days After Anthesis, etc).

Response: I apologize for any confusion. The terms "fresh" and "overripe" have been
replaced with "mature" and "senescent" for consistency throughout the manuscript.
Criteria for determining the states of maturity and senescence have been included in the
Methods section, and the content is as follows:

Fruit from the 35th days after artificial pollination (DAAP) of *H. undatus* (Vietnam
No. 1) were collected. The harvested fruit were used as mature samples. Then, they
were stored under conditions of 25 °C and 85% relative humidity. During the storage
period, the levels of superoxide anion and flavonoids in the exocarp were monitored
every three days. The senescent sample was identified typically on the ninth day,
characterized by the eruption of superoxide anions, the second peak in flavonoid levels,
and the simultaneous appearance of pronounced dehydration-induced wrinkling or
disease spots on the fruit phenotype (Lines 1006-1014).

To verify the changes in flavonoids during senescence, we conducted additional
VIGS experiments (Fig. R11). By silencing the RNA of the senescence-related key
gene *HuCMB1*, as identified in this study, we monitored the temporal changes in
flavonoid levels in *H. undatus* pericarp after *HuCMB1* silencing. Combining
phenotypic changes, we preliminarily confirmed the dual-peak trend of flavonoids
during the senescence process. The results of this study clearly demonstrated the silence
of the key senescence transcription factor HuCMB1 significantly impacting the
biosynthesis of flavonoids. Also, through RT-qPCR experiments, we confirmed that
the genes related to flavonoid synthesis were regulated by HuCMB1, indicating its
crucial regulatory role in senescence.

Figure R11 Silencing of *HuCMB1* led to faster senescence of *H. undatus*. (Added as Figure 5 in the manuscript)

A, Expression of four senescence-related genes at different components of pericarp and time points of senescence by RT-qPCR. * represents $p < 0.05$, and ** represents $p < 0.01$; **B**, Schematic showing the domain structure of *HuCMB1*; **C-E**, Changes in fruit phenotype (**C**), weight loss rate (**D**), and flavonoid levels in the exocarp (**E**) after silencing the *HuCMB1* gene. **F**, RT-qPCR was used to analyze the expression of *HuCMB1*, *HuERD6-2*, *HuSAG12*, and *HuMED32* in *H. undatus* after 9 days of storage under *HuCMB1* gene silencing.

(details)

-Line 432: change “Pentose” to “pentose”

Response: The word “Pentose” has been replaced by “pentose”.

-Line 751: change “hours” to “h”

Response: The word “hours” has been replaced by “h”.

-Line 860: delete “a” before “at”

Response: Yes, there is a grammatical error in this passage. The corrected sentence is as follows: Libraries were prepared to achieve a sequencing depth of approximately 250 M read-pairs per sample or higher.

Reviewer #4 (Remarks to the Author):

The authors used microarray-based spatial transcriptomics and single-cell RNA-sequencing to reveal tissue architecture in pericarp of *Hylocereus undatus*, constructed the single-cell expression atlas of the pericarp of pitaya, revealing exocarp and mesocarp cells undergoing the most significant changes during the fruit senescence process. The research is interesting, while some points should be taken in consideration.

1. Whether the ripening process of pitaya fruit is the same as tomato that has post-ripening stage. Is postharvest storage post-ripening or senescence? How to define senescence of pitaya fruit? What are the criteria or indicators of senescence for this fruit? The authors should give some explanation, as this is important for all the experiments in this article.

Response: Really appreciated for your kindly comments. Yes, providing detailed descriptions of the senescence samples of *H. undatus* is crucial. Unlike climacteric fruits such as tomatoes, which undergo a respiratory burst, *H. undatus* is a non-climacteric fruit. Therefore, *H. undatus* does not exhibit the typical physiological changes associated with a respiratory burst during the post-harvest senescence process.

It is generally acknowledged that *H. undatus* enters the senescence stage immediately after harvesting. In other words, the samples at the start of the post-harvest storage period are considered mature samples, marking the onset of the senescence process.

During the storage period, the levels of superoxide anion and flavonoids in the exocarp were monitored every three days. The senescent sample was identified typically on the ninth day, characterized by the eruption of superoxide anions, the second peak in flavonoid levels, and the simultaneous appearance of pronounced dehydration-induced wrinkling or disease spots on the fruit phenotype. These descriptions have been added to the Methods section of the main text (Lines 1006-1014).

To validate the changes in flavonoids during the senescence process, we conducted additional VIGS experiments (see the response to question 2). By silencing the RNA of the senescence-related key gene *HuCMB1*, as identified in this study, we monitored the temporal changes in flavonoid levels in *H. undatus* pericarp after *HuCMB1* silencing. Combining phenotypic changes, we preliminarily confirmed the dual-peak trend of flavonoids during the senescence process. The results of this study clearly demonstrate the silence of the key senescence transcription factor *HuCMB1* significantly inhibits the biosynthesis of flavonoids. Also, through RT-qPCR experiments, we confirmed that the genes related to flavonoid synthesis are significantly suppressed in the silenced lines of *HuCMB1*. Therefore, flavonoids are likely to serve as a potential indicator of fruit senescence.

2. Line291: Validation experiments, one gene is not enough, and should take one gene every cell group for gene or RNA localization, of course, if possible, combine to do some fusion of fluorescent protein for cellular localization. If the authors only use one gene to do this, it should be studied in depth, such as using gene editing technology to verify whether the gene is functional. If gene editing is not possible, transient gene regulation techniques can be considered.

Response: Really appreciated for your kindly comments. Yes, we fully agree with your opinions. Therefore, we supplemented the study with RT-qPCR experiments to validate the expression site-specificity and temporal specificity of the four senescence-related key genes identified in this work. We also conducted additional VIGS experiments to preliminarily validate the function of *HuCMB1*. A brief summary is provided below:

From the pseudotime heatmap, it is evident that the expression peaks of the four

key senescence-related genes, *HuMED32*, *HuERD6-2*, *HuSAG12*, and *HuCMB1*, occur
 at the onset, mid-stage, late stage, and endpoint of the senescence process, respectively.
 Through RT-qPCR, the expression changes of these four key senescence-related genes
 were assessed at three time points during *H. undatus* senescence in the exocarp,
 mesocarp, and endocarp (Fig. R12A).

 The RT-qPCR results indicated that the expression of these four genes was
 specifically localized in different components, consistent with the single-cell
 transcriptome results (Table R6). The expression changes at different stages of
 senescence aligned with the pseudotime heatmap from scRNA-seq results (Fig. R12B).
 A description of the results has been added to the Results section, as detailed below:

The gene *HuCMB1* exhibited the highest expression in the EX component and was
 highly expressed during the mid and late stages of senescence ($p < 0.01$). The well-
 known senescence marker gene *HuSAG12* was exclusively highly expressed in the ME
 component during the late stage of senescence ($p < 0.05$). *HuERD6-2* showed high
 expression only in the EX component during the mid-stage of senescence ($p < 0.01$).
 The gene *HuMED32* was highly expressed in the EX component during the late stage
 of senescence ($p < 0.05$) (Fig. R12AB and Figure 5A, Tables R1 and S12).

 **Figure R12 Expression of senescence related genes** (Figure R12A was added as Figure 5A in the
 manuscript. Figure R12B was shown in the Fig. 4I and S3D) **A**, Expression of four senescence-
 related genes at different components of pericarp and time points of senescence by RT-qPCR. *
 represents $p < 0.05$, and ** represents $p < 0.01$; **B**, Heatmap showed the expression of senescence
 related genes along a pseudotime progression.

 **Table R6 Expression of senescence related hub genes in components of pericarp in scRNA-**
 **seq profiles.** (Shown as sheet 2 of the Table S12 in the manuscript)

Compon ents	Clust ers	Gene id	Gene Name	avg_Log 2FC	pct.1(%)	pct.2(%)	Signific ant	Regul ate
ME	1			1.200329 43	85.2	31.8	yes	up
	8			3.718588 37	98.5	35	yes	up
	0	HU08G02 237	HuSAG 12	- 1.025316 5	52.6	37	yes	down
EX	2			- 3.825551 1	11.2	43.2	yes	down

	3			- 2.194015 5	18.7	41.5	yes	down
	5			- 2.173106 1	21.6	41	yes	down
	7			- 2.234789 3	17.8	41.1	yes	down
	9			- 3.909085 5	4.3	40.5	yes	down
EN	4			- 2.201000 1	15.9	41.7	yes	down
	11			- 0.898904 9	23.6	39.5	yes	down
VB	10			- 3.083416 5	12	40	yes	down
	1			-0.51863	12.3	20.5	yes	down
ME	6			- 0.521777 5	13.4	19.9	yes	down
	8			- 0.597675 9	13	19.8	yes	down
	7	HU05G02 127	HuERD 6-2	0.968398 2	38.3	17.7	yes	up
EX	3			- 0.407859 5	9.4	20.4	yes	down
	5			- 0.462027 8	11.5	20.1	yes	down
EN	4			1.286084 3	40.7	17.1	yes	up
	0			0.504605 14	83.7	57.6	yes	up
ME	1			0.869305 3	81	58.3	yes	up
	6			0.487063 41	78.6	59.9	yes	up
	8			0.582794 52	83.5	59.9	yes	up
	3	HU10G00 469	HuCM B1	- 0.628560 2	41.4	63.6	yes	down
	5			-0.273793	54.8	62.1	yes	down
EX	7			- 0.390901 8	55.1	62	yes	down
	9			- 1.411496 1	32.4	62.5	yes	down

VB	10			- 1.796187 2	22.1	62.6	yes	down
EN	4			- 1.827219 4	27.7	64.9	yes	down
	11			-2.382495	18.9	62.2	yes	down
EX	2	HU08G00 932	HuME D32	0.225712 06	12.5	8.1	yes	up

Following that, we supplemented the study with VIGS experiments (Figure R13). By silencing the RNA of the senescence-related key gene *HuCMB1*, as identified in this study, we monitored the temporal changes in flavonoid levels in *H. undatus* pericarp after *HuCMB1* silencing. Combining phenotypic changes, we preliminarily validated the functionality of the *HuCMB1* gene. Additionally, through RT-qPCR experiments, we confirmed that *HuMED32* is regulated by *HuCMB1*, indicating its crucial regulatory role in senescence. For detailed results and analysis, please refer to the main text of the manuscript.

Figure R13 Silencing of *HuCMB1* led to faster senescence of *H. undatus*. (Added as Figure 5 in the manuscript)

A, Expression of four senescence-related genes at different components of pericarp and time points of senescence. * represents $p < 0.05$, and ** represents $p < 0.01$; **B**, Schematic showing the domain structure of *HuCMB1*; **C-E**, Changes in fruit phenotype (**C**), weight loss rate (**D**), and flavonoid levels in the exocarp (**E**) after silencing the *HuCMB1* gene. **F**, RT-qPCR was used to analyze the expression of *HuCMB1*, *HuERD6-2*, *HuSAG12*, and *HuMED32* in *H. undatus* after 9 days of storage under *HuCMB1* gene silencing.

3. Line959: In the ROS detection experiment, ROS would erupt on the wound surface during fruit cutting. How to ensure that ROS detected were not caused by the wound?

Response: Really appreciated for your kindly comments. Yes, this is a highly
specialized issue. In the experiment, we have also observed this phenomenon. When
preparing slices, as soon as the cutting began, it was inevitable that ROS would be
generated at the wound site, especially with the production rate of superoxide anions
occurring on a millisecond timescale. The purpose of our experiment was to detect ROS
generation, so it was not feasible to suppress this cellular response by adding ROS
scavengers. However, in our experiments, through repeated pre-experimental practice
by students, we achieved stability in the operational procedures. This allowed us to
maintain consistency in the processing of all samples at different time points. Therefore,
the ROS generated during the slicing preparation operation was essentially consistent
across all samples, and this ROS generated during sample preparation could be
considered as background signal. As shown in Figure S4G, we observed significant
differences in the accumulation levels and localization of ROS at different time points.
The background ROS signal induced by the procedure was significantly lower than the
levels of ROS generated over time in the experiment, and it did not mask the variations
in ROS among different samples.

In some of our previous papers, we also addressed the detection of ROS such as
superoxide anions and hydrogen peroxide. The background signal of ROS induced by
the procedures was similarly well addressed, without affecting the analysis of ROS
differences among samples. The list of relevant papers is as follows:

1165 **References**

- 1. **Li, X.**, Imlay, J. A. Improved measurements of scant hydrogen peroxide enable
experiments that define its threshold of toxicity for *Escherichia coli*. *Free Radical*
*Biology and Medicine*. *Free Radical Bio Med.* 2018, **120**, 217-227. DOI:
10.1016/j.freeradbiomed.2018.03.025
- 2. **Li, X.**, et al. Transcriptomic analysis reveals key genes related to antioxidant
mechanisms of *Hylocereus undatus* quality improving by trypsin during storage.
*Food & Function*. 2019. **10**, 8116-8128. DOI: 10.1039/c9fo00809h
- 3. Pang, X. Y., Li, X. L., Liu, X. R., Cai, L. N., Li, B. R., **Li, X.** Transcriptomic
analysis reveals Cu/Zn SODs acting as hub genes of SODs in *Hylocereus undatus*
induced by trypsin during storage. *Antioxidants* 2020, **9**, 162. DOI:
10.3390/antiox9020162
- 4. **Li, X.**, et al. Transcriptomic analysis reveals hub genes and subnetworks related to
ROS metabolism in *Hylocereus undatus* through novel superoxide scavenger
trypsin treatment during storage. *BMC Genomics*. 2020, **21**, 437. DOI:
10.1186/s12864-020-06850-1

4. Line91: “using the Cytoscape plugin SCODE”. Correct citation (original references)
should be given to software, databases, etc., and such issues should be examined in
whole article.

Response: Really appreciated for your kindly comments. Yes, citations for the relevant
literature need to be included in the corresponding places in the manuscript. References
have been added at lines 99, 650, 653, 657, 675, 678, 753, 757, 907, and so on.

The reference cited in line 99 (original 91) was related to the research background
of SCODE.

The references cited in lines 65-66 were related to the research background of
single-cell transcriptome sequencing technology on the fruit.

The references cited in line 650, 653, and 657 were related to the research
background of single cells transcriptomics.

The references cited in line 675, and 678 were related to the research background
of database of marker genes.

The references cited in line 753 and 757 were related to the research background
of SingleR, SciBet, RCTD and CARD.

The references cited in line 907 were related to the research background of ROS
involved in the senescence.

Furthermore, references of “methods”, such as UPLC-MS/MS, RT-qPCR, Virus-
Induced Gene Silencing (VIGS), were cited.

5. Lines62-64: “Also to date, there have been no reports on the expression
characteristics and variation patterns of different cell types during the senescence and
decay processes of fruit.” The authors should check carefully to give this statement,
make sure that there was definitely no report in other fruit.

Response: Drawing such a conclusive statement might be too assertive, and even
though I have thoroughly reviewed the literature, there might be reports that I have
overlooked. I can confirm that, to date, while researchers have utilized single-cell
transcriptome sequencing technology to construct single-cell atlases for fruits and
vegetables, such as strawberries (*Fragaria ananassa*) (Bai et al., 2022) and lychees
(*Litchi chinensis*) (Yang et al. 2023), the tissues analyzed were leaves and shoot apices,
and there have been no reports utilizing single-cell sequencing to investigate fruit.

The description in the manuscript has been modified to: "Also to date, there have
been no reports **utilizing scRNA-seq technology** on the expression characteristics and
variation patterns of different cell types during the senescence and decay processes of
fruit".

The references mentioned above have been added to the manuscript, and the list is
as follows:

[1] Bai, Y., Liu, H., Lyu, H., Su, L., Xiong, J. & Cheng, Z. M. Development of a single-
cell atlas for woodland strawberry (*Fragaria vesca*) leaves during early *Botrytis*
*cinerea* infection using single cell RNA-seq. *Hortic Res.* **9**, uhab055. (2022).
doi: 10.1093/hr/uhab055.

[2] Yang, M. C., Wu, Z. C., Chen, R. Y., Abbas, F., Hu, G. B., Huang, X. M., Guan, W.
S., Xu, Y. S. & Wang, H. C. Single-nucleus RNA sequencing and mRNA
hybridization indicate key bud events and LcFT1 and LcTFL1-2 mRNA
transportability during floral transition in litchi. *J Exp Bot.* **74**, 3613–3629. (2023).
doi: 10.1093/jxb/erad103.

6. Discussion: The authors need add some discussion about currently available fruit
single cell sequencing or spatiotemporal transcriptome researches. Therefore, it can
better reflect the innovation of the author's research.

Response: We summarized the current available technologies for single-cell
sequencing and spatial transcriptomics, elucidating the work conducted in the field of
plants using these techniques. We compared and discussed our work in light of these
studies. The above-mentioned content has been incorporated into the Discussion
section, with specific details as follows:

Currently, there are various platforms for scRNA-seq, including BD Rhapsody (BD
Biosciences, USA), Chromium (10x Genomics, USA), among others, providing
multiple possibilities for high-throughput analysis (Ke et al., 2022). Drop-based high-
throughput and low-cost cell processing platforms, such as Drop-seq or the Chromium
10x platform, dominate the field of plant single-cell transcriptomics. Leveraging these
platforms, studies have been conducted on various plant tissues, including Arabidopsis
roots and leaves, rice stems, and sheaths (Seyfferth et al., 2021; Khozyainova et al.,
2023). However, to date, there have been no reports on the scRNA-seq of fruits.

[revised manuscript text omitted]

If you have any question on this manuscript, please no hesitate to contact me. Thank
you very much!

Prof. Robert Henry

Reviewer #1 (Remarks to the Author):

Your manuscript "Single-Cell and Spatial RNA Sequencing Reveal the Spatiotemporal Trajectories of Fruit Senescence " is recommended for acceptance. All suggestions have been well modified.

Reviewer #2 (Remarks to the Author):

For previous comment 1: I agreed with the authors' opinion that the annotation results obtained from different algorithms might not have been identical because their calculation principles differed.

In this version, the authors abandoned the MIA method, which is not suitable for the sample size in this study, and the annotation results are clearer now. However, the authors should explain the reasons for categorizing Clusters 4 and 11 as EN cells; obviously, they are highly associated with ENF cells in both the SciBet and Single R methods (Figure 2D).

For previous comment 2: No additional questions.

For previous comment 3: Thanks for the detailed explanation and now I understand the re-clustering analysis of the EX and ME cells.

I have some other questions:

(1) In Figure 3A, in the CK sample, the proportion of EX cells was 64.97%, while the proportion of ME cells was only 10.87%. This statistical result seems to be inconsistent with the microscope images of the CK sample in Figure 3B, in which the ME region is obviously larger than the EX region.

(2) In the total of 5788 ME cells, why do 5174 cells (89.39%) originate from the Post sample? In Figure 3B, the ME cell region is even larger in CK than that in Post. Why did the study get such few ME cells from the CK sample?

Could you provide some possible reasons?

For previous comment 4: The pseudotime analysis is usually performed in cell populations with developmental associations. The authors selected EX and ME cells to conduct pseudotime analysis. Whether there are developmental relationship between EX cells and ME cells? it seems that they are two distinct cell types, and they probably have different developmental and senescence processes. Although the CK and Post samples have a time relationship, I think it is not proper to perform the pseudotime analysis of these mixed cell populations, unless the authors could provide much more evidence.

For previous comment 5: More validated RT-qPCR analysis of senescence-related genes has been provided in this version. It would be better if the positive signals could be indicated by arrows in the Figure 3G.

For previous comment 6: The spatial transcriptome data in this study is important because it's crucial for the cell type annotation. The authors used only one sample but without detailed information. For example, is the sample used here for the spatial transcriptome CK or Post? Is it proper to use a one-time point sample to annotate all these cell populations in CK and Post samples? The authors should provide more explanations.

For previous comment 7: The discussion section in this version has not been well improved as expected. It is still similar to the result part. Please improve this section and highlight the novel discovery and potential value.

For previous comment 8: The figure legend in Fig.6 is more informative now.

For previous comment 9: No additional questions.

10. In line 89, the authors described that "Through the application of three different algorithms", while in line 222, the authors described that "Based on the results from the four algorithms". They are not consistent.

11. In Figure 3E, "Simple" is a wrong label. The authors should double-check the figures and words throughout the manuscript.

Reviewer #2 Attachment on the following page

Reviewer #2 (Remarks to the Author):

In this manuscript entitled “Single-Cell and Spatial RNA Sequencing Reveal the Spatiotemporal Trajectories of Fruit Senescence”, thorough integrating spatial transcriptomics and single-cell RNA sequencing, the authors constructed a single-cell expression atlas of the pericarp pitaya (*Hylocereus undatus*) and established cellular differentiation and gene expression trajectories during fruit senescence. Although the authors provide interesting data analysis results, most of them are descriptive and not rigorous enough to support the conclusions.

Revision: The authors had greatly modified this manuscript. However, there are still several questions that need to be addressed, and new insights should be well discussed and highlighted in the discussion section.

Major issues

1. Cell type annotation is the first important step in single cell RNA sequencing data analysis and the foundation for all subsequent analyses. To annotate the 13 clusters identified in single-cell transcriptomic data, the authors utilized three categories of five different methods (SciBet, SingleR, MIA, RCTD, and CARD) to perform the association analysis between spatial transcriptomics and single cell transcriptomics. However, it is very confusing that the association analysis results by five methods are not consistent.

For examples, from line 207 to line 209, the authors reported “Based on analysis using SingleR (Single Cell Recognition) and SciBet (SingleCell Identifier Based on E-test), we observed that cell clusters 0, 1, 6, and 8 from the scRNA-seq data clustered together, suggesting that they belong to the ME component.” However, Cluster 0 in SciBet method is highly associated with ENF, not the ME, though cluster 0 in Single R method is associated with ME (Figure 2D). In addition, in MIA method, all 13 clusters are highly associated with ENF and EX, but the authors classified cluster 2,3,5,9 as EX. Similarly, the authors assigned cluster 10 as VB, and cluster 5 as EX (in line 232), but the two clusters are highly associated with VB in Single R method (Figure 2D). Thus, the cell type annotation results are little confusing and the annotation basis is not rigorous enough to support the conclusion, and the authors should solve this problem.

Response: Considering that the five methods belonged to three major categories of algorithms, even if they fell under the category of deconvolution algorithms, such as SingleR and SciBet, their calculation principles differed. Therefore, the results obtained from these five algorithms might not have been identical. Regarding cluster 0, SingleR showed a high correlation of cluster 0 with ME. In the results from SciBet, the correlation of cluster 0 with ME was the second highest among the five components. Additionally, the results from CARD and RCTD also indicated a high correlation of cluster 0 with the ME component. Another factor for assessing the correlation between clusters and components was the position of clusters in the UMAP plot, where clusters belonging to the same component would have been close together. Therefore, considering these factors, cluster 0 was assigned to the mesocarp (ME) component.

Regarding MIA, we agree with your opinions. MIA is an analytical method that integrates single-cell expression profiles and spatial transcriptomic data through multimodal analysis. It calculates the overlap between spatial transcriptomic region-

specific differentially expressed genes and marker genes for cell types identified in single-cell transcriptomics. It uses the hypergeometric distribution to infer the enrichment of specific cell types in particular tissue regions, and it calculates p-values with all genes as the background. From Figure 1C, it could be observed that the expression localization of marker genes was not absolute but relatively high in certain cell clusters and lower in others. The situation was similar for region-specific differentially expressed genes in spatial transcriptomics. Due to the statistical principles involved, MIA's calculation results might have been more accurate with larger data sample sizes. However, given the limited sample size in this study, the results from MIA were considered unreliable and were excluded from the analysis. Currently, single-cell transcriptomics is still a relatively new technology, with fewer publications utilizing spatial transcriptomics. Therefore, algorithms suitable for joint analysis of two omics data types are still in the early stages of development, and their applications are not yet widespread. In the future, we are willing to explore the applicability of the MIA algorithm by applying it to more single-cell data and identifying its optimal use cases.

Reviewer:

For previous comment 1: I agreed with the authors' opinion that the annotation results obtained from different algorithms might not have been identical because their calculation principles differed.

In this version, the authors abandoned the MIA method, which is not suitable for the sample size in this study, and the annotation results are clearer now. However, the authors should explain the reasons for categorizing Clusters 4 and 11 as EN cells; obviously, they are highly associated with ENF cells in both the SciBet and Single R methods (Figure 2D).

2. From line 261 to line 273, in Figure 3B, the authors described the microscopic images of different layers of *H. undatus* pericarp in CK group and Post group. However, we find it difficult to obtain the effective information from the images. The authors should label different cell layers and mark the corresponding characteristics in these images.

Response: Yes, annotating the various cellular layers helps readers understand the information in the images. The various cellular layers of the CK and Post groups have been labeled in Figure 3B. Descriptions of the characteristics of each layer have been provided in the Results section.

Figure R6 Optical microscope images of CK and Post samples (Figure R6 was shown as revised Figure 3B in the manuscript).

Reviewer:

For previous comment 2: No additional questions.

3. In line 276, the authors re-clustered EX and ME cells, and got five distinct subgroups. But the authors did not perform cell type recognition of these subgroups. From line 286 to line 289, the authors described that “Based on the scRNA-seq profiles, the UMAP overview of the 13 clusters in CK and Post samples showed the expression localization of the top gene in each of the five subgroups (Fig. 3F). Among them, Hu08G02237 exhibited strong expression specificity in the mesocarp.” However, this statement is confusing. In Figure 3F, heat map and UMAP only showed that Hu08G02237 is the marker gene of subcluster 0, not the mesocarp. In addition, I cannot get the significance of the re-clustering analysis.

Response: Really appreciated for your kindly comments. We apologize for not clearly explaining our research approach. We conducted two pseudotime analyses. In the first pseudotime analysis, we selected two cell types, EX and ME, for computation. However, the pseudotime trajectory was not clear. In the results of the pseudotime analysis for all EX and ME cells, we performed GO and KEGG analysis of pseudotime-correlated genes (Figure S3C, Table S5) and identified significant enrichments in senescence, resistance, phenylpropanoid pathway, and reactive oxygen species metabolism. This implied that these genes play crucial roles in the senescence trajectory. Thus, four gene sets related to these four functions were established. By gene set scoring, 9,019 cells highly correlated with these genes were selected. Subsequently, pseudotime analysis of these functionally clarified cells demonstrated a clearer developmental trajectory during senescence (Figure 4B). Therefore, the first pseudotime analysis revealed the major functions of target cells, while the second pseudotime analysis, focusing on cells highly correlated with the identified functions, revealed the cellular developmental trajectory.

Regarding the localization of Hu08G02237 in the subcluster, this conclusion was derived from the cell tracing results of the subgroups described below. I apologize for any confusion in our previous manuscript. The explanation of the tracing results and relevant data tables has been included in the manuscript for better understanding.

For this subcluster analysis, only cells from the exocarp and mesocarp were selected. We supplemented the basis for the cell sources in the subcluster analysis. The cell subgroup dimensionality reduction clustering analysis in this study used the FindClusters function of the Seurat software (Chen et al., 2020; Zhang et al., 2021), clustering the specified cell groups into 5 subclusters. The traceability statistics of cell numbers have been added to the supplementary materials of the manuscript as Table S5. Single-cell sequencing technology marked cells using barcodes, with each cell having its own barcode (a nucleotide sequence). Through barcode information, the origin of each cell could be traced, indicating whether the cell was from the CK or Post sample (specific barcode information is available in Table R3). The results of cell counting were presented in Figure 3C, where cells in subclusters 1 and 2 primarily originated from the CK group, and cells in subclusters 0, 3, and 4 primarily originated from the Post group (Table S5).

Based on the traceability results from the aforementioned barcodes, 99.81% of cells in subclusters 1 and 2 $[(3892+386)/(3892+386+8)]$ originated from the CK sample, and 99.94% of cells in subclusters 0, 3, and 4 $[(6848+264+46)/(6848+264+46+2+2)]$ originated from the Post sample (Table S5). The UMAP results showed that cells from CK and Post samples formed two major clusters, with CK cells clustering on the left and Post cells on the right.

For cells specifically from the mesocarp (ME), the traceability analysis showed a total of 5788 cells, with 5174 cells (89.39%) originating from the Post sample. The UMAP results indicated that Hu08G02237 exhibited specific expression in the Post sample, mainly located in the mesocarp. However, considering that there were also some exocarp cells in the Post sample, the expression specificity statement was modified to: Hu08G02237 exhibited expression specificity in the Post sample, mainly located in the mesocarp.

Table S5 Information about the cellular sources for 5 cell subclusters of ME and EX.

Subcluster	0	1	2	3	4	Sum
CK	0	3892	386	2	2	4282
Post	6848	8	0	264	46	7166

Table R3 Example of Specific Information for Barcode in these subclusters.

Barcode	Sample	Subcluster
Post_AAACCCAAGGCAGCTA-1	Post	0
Post_AAACCCACAATACCCA-1	Post	0
Post_AAACCCATCCGGCAAC-1	Post	0
Post_AAACCCATCGCAATGT-1	Post	0
Post_AAACGAAAGCTGCCAC-1	Post	0
Post_AAACGAACATGACACT-1	Post	0
Post_AAACGAAGTCCAAGAG-1	Post	0
Post_AAACGAAGTTGCGGCT-1	Post	0
Post_AAACGCTAGCGTCAAG-1	Post	0
Post_AAACGCTGTGCATCTA-1	Post	0
CK_GATGGAGGTTTACCTT-1	CK	2
CK_GATGTTGAGGCCTAGA-1	CK	2
CK_GATTCTTGTGTCATTG-1	CK	2
CK_GATTTCTGTTGCATGT-1	CK	2
CK_GCAGCTGCATCATGAC-1	CK	2
CK_GCAGGCTTCGTGGCTG-1	CK	2
CK_GCATTAGCAGTCGCAC-1	CK	2
CK_GCCGTGACACCAGCCA-1	CK	2
CK_GCGATCGCACTGAGGA-1	CK	2
CK_GGAACCCTCTTCTGTA-1	CK	2

References

- [1] Chen Y. P. et al. Single-cell transcriptomics reveals regulators underlying immune cell diversity and immune subtypes associated with prognosis in nasopharyngeal

- carcinoma. *Cell Res.* **30**, 1024-1042. (2020). DOI: 10.1038/s41422-020-0374-x
- [2] Zhang, T. Q., Chen, Y., Liu, Y., Lin, W. H. & Wang, J. W. Single-cell transcriptome atlas and chromatin accessibility landscape reveal differentiation trajectories in the rice root. *Nat. Commun.* **12**, 2053 (2021). DOI: 10.1038/s41467-021-22352-4

Reviewer:

For previous comment 3: Thanks for the detailed explanation and now I understand the re-clustering analysis of the EX and ME cells.

I have some other questions:

- (1) In Figure 3A, in the CK sample, the proportion of EX cells was 64.97%, while the proportion of ME cells was only 10.87%. This statistical result seems to be inconsistent with the microscope images of the CK sample in Figure 3B, in which the ME region is obviously larger than the EX region.
- (2) In the total of 5788 ME cells, why do 5174 cells (89.39%) originate from the Post sample? In Figure 3B, the ME cell region is even larger in CK than that in Post. Why did the study get such few ME cells from the CK sample?

Could you provide some possible reasons?

4. The pseudotime analysis is confusing. The pseudotime analysis is usually performed in cell populations with developmental associations. In line 378, the authors selected 9019 cells from all cells based on gene set scoring results and performed subcluster analysis and developmental trajectories analysis. The 9019 cells contained multiple cell types and their developmental relationships are complicated. This analysis may be not scientific and meaningful.

Response: Yes, we performed two pseudotime analyses, and we apologize for not clearly explaining the research approach. In summary, the first pseudotime analysis revealed the primary functions of the target cells. The second pseudotime analysis, focusing on cells with high correlation to the selected relevant functions, elucidated the trajectory of cell differentiation. The description of the approaches for the two pseudotime analyses has been added to the Discussion section for better reader understanding. The specific approaches are outlined as follows.

In the first pseudotime analysis, we selected the EX and ME cell types for computation. However, the pseudotime trajectory was not clear. Upon analyzing the pseudotime results for all EX and ME cells, we conducted GO and KEGG analyses for pseudotime-related genes, revealing significant enrichment in four functions: senescence, resistance, phenylpropanoid pathway, and reactive oxygen metabolism. This suggested a crucial role for these genes in the senescence trajectory. Subsequently, four gene sets were established, and through gene set scoring, we filtered out 9,019 cells highly correlated with these gene sets.

The majority of these 9,019 cells were EX and ME cells (Table R4). To ensure the completeness of the results, we retained other cell types highly correlated with the target gene set without deleting them. On the other hand, the selection of such highly correlated cells effectively removed cells unrelated or minimally related to the target gene set, resulting in a clearer pseudotime analysis of the cell differentiation trajectory.

In the second pseudotime analysis, based on the analysis results from gene set scoring, cells more relevant to the senescence process were selected, and irrelevant cells

were excluded. The pseudotime analysis of the filtered 9,019 cells with high correlation to senescence provided a clearer differentiation trajectory (Fig. 4B). The relevant literature on gene set scoring has been included in the manuscript (Ren et al., 2021; Hao et al., 2021).

The early-response genes to senescence were identified through the pseudotime heatmap, representing the most crucial findings of this study. We supplemented the experiments with RT-qPCR and VIGS, using RT-qPCR to confirm the specific expression of key genes in selected components and the high expression period during the senescence process. The VIGS experiments provided preliminary validation of the functions of the selected senescence-related genes. This indicates the effectiveness of the pseudotime analysis results in this study. For images related to the VIGS results and RT-qPCR results for core senescence-related genes, please refer to the following (Fig. R8).

Table R4 Cell Count for each cluster in gene set scoring results

Components	Clusters	Num of cells high related with gene sets			
		Resistance	Senescence	Phenylpropanoids	ROS
	0	47	240	207	848
	1	92	540	56	536
ME	6	166	164	167	351
	8	55	573	35	188
Sum of ME cells		360	1517	465	1923
	2	640	89	847	442
	3	311	231	423	326
EX	5	161	212	203	126
	7	552	171	497	251
	9	121	69	271	30
Sum of EX cells		1785	772	2241	1175
Sum of ME and EX cells		2145	2289	2706	3098
Percent of EX and ME cells in all components		64.43%	68.76%	81.28%	93.06%
EN	4	958	738	535	212
	11	136	147	46	11
Sum of EN		1094	885	581	223
ENF	12	14	19	4	0
VB	10	76	136	38	8

References

- [1] Ren, X., et al. COVID-19 immune features revealed by a large-scale single-cell transcriptome atlas. *Cell*. **184**, 1895-1913 (2021). doi: 10.1016/j.cell.2021.01.053
- [2] Hao, Y. H., et al. Integrated analysis of multimodal single-cell data. *Cell*. **184**, 3573-3587.e29 (2021). doi: 10.1016/j.cell.2021.04.048.

Reviewer:

For previous comment 4: The pseudotime analysis is usually performed in cell populations with developmental associations. The authors selected EX and ME cells to conduct pseudotime analysis. Whether there are developmental relationship between EX cells and ME cells? it seems that they are two distinct cell types, and they probably have different developmental and senescence processes. Although the CK and Post samples have a time relationship, I think it is not proper to perform the pseudotime analysis of these mixed cell populations, unless the authors could provide much more evidence.

Minor issues

5. More validated expression evidence for the marker genes of the identified cell types should be provided. The RNA FISH images in Figure 3G are not clear enough to understand.

Response: Really appreciated for your kindly comments. We agree with your perspective. In Figure 3G, the bright-field microscopy images serve as the foundation for understanding the spatial localization of gene expression in RNA-FISH images. We have added labels for the various cellular layers in the images, allowing readers to comprehend the stained cell nuclei in both DAPI and FAM-stained images, as well as the positioning of RNA for marker genes.

Figure R7. RNA FISH indicated that the predominant location of *HuSAG12* was in the mesocarp. (Shown as revised Figure 3G in the manuscript)

Components of EX, ME, and VB were labelled in the light field image. *HuSAG12* probes were labeled with FAM (green). Nuclei were stained with DAPI (blue). Scale bar: 40 μ m.

Based on the current FAM staining results, we can observe the expression localization of *HuSAG12* RNA, although the effectiveness is not optimal. To further validate the gene expression localization, we employed RT-qPCR to confirm the expression changes of the four key genes related to senescence that were identified in this study during the senescence process (Fig. R8).

The results from RT-qPCR demonstrated that the specific localization of the four genes in different components aligned with the findings from single-cell transcriptomics (Fig. R8A and Table R5). Moreover, the expression changes at different stages of senescence were consistent with the pseudotime heatmap results from scRNA-seq (Fig. R8A and B). The description of the results has been integrated into the Results

section, as outlined below:

The expression of the gene *HuCMB1* was highest in the EX component and significantly elevated in the mid and late stages of senescence ($p < 0.01$). The well-established senescence marker gene *HuSAG12* exhibited exceptionally high expression solely in the late senescence phase of the ME component ($p < 0.05$). *HuERD6-2*, on the other hand, showed high expression in the EX component specifically during the mid-senescence period ($p < 0.01$). The gene *HuMED32* demonstrated elevated expression in the late senescence phase of the EX component ($p < 0.05$) (Fig. R8AB and Fig. 5A, Table R5 and S12).

Figure R8 Expression of senescence related genes (Figure R8A was added as Figure 5A in the manuscript. Figure R8B was shown in the Fig. 4I and S3D). **A**, Expression of four senescence-related genes at different components of pericarp and time points of senescence by RT-qPCR. * represents $p < 0.05$, and ** represents $p < 0.01$; **B**, Heatmap showed the expression of senescence related genes along a pseudotime progression.

Table R5 Expression of senescence related hub genes in components of pericarp in scRNA-seq profiles. (Shown as sheet 2 of the Table S12 in the manuscript)

Components	Clusters	Gene id	Gene Name	avg_Log2FC	pct.1(%)	pct.2(%)	Significant	Regulate	
ME	1			1.200329 43	85.2	31.8	yes	up	
	8			3.718588 37	98.5	35	yes	up	
	0			- 5	1.025316	52.6	37	yes	down
EX	2	HU08G02 237	HuSAG 12	- 1	3.825551	11.2	43.2	yes	down
	3			- 5	2.194015	18.7	41.5	yes	down
	5			- 1	2.173106	21.6	41	yes	down
	7			- 3	2.234789	17.8	41.1	yes	down
	9			- 3.909085	4.3	40.5	yes	down	

				5				
	4			-	2.201000	15.9	41.7	yes down
EN				1				
	11			-	0.898904	23.6	39.5	yes down
				9				
VB	10			-	3.083416	12	40	yes down
				5				
	1			-	-0.51863	12.3	20.5	yes down
				5				
ME	6			-	0.521777	13.4	19.9	yes down
				5				
	8			-	0.597675	13	19.8	yes down
				9				
	7	HU05G02	HuERD	0.968398	38.3	17.7	yes up	
		127	6-2	2				
EX	3			-	0.407859	9.4	20.4	yes down
				5				
	5			-	0.462027	11.5	20.1	yes down
				8				
EN	4			1.286084	40.7	17.1	yes up	
				3				
	0			0.504605	83.7	57.6	yes up	
				14				
ME	1			0.869305	81	58.3	yes up	
				3				
	6			0.487063	78.6	59.9	yes up	
				41				
	8			0.582794	83.5	59.9	yes up	
				52				
	3			-	0.628560	41.4	63.6	yes down
				2				
EX	5	HU10G00	HuCM	-0.273793	54.8	62.1	yes down	
		469	B1					
	7			-	0.390901	55.1	62	yes down
				8				
	9			-	1.411496	32.4	62.5	yes down
				1				
VB	10			-	1.796187	22.1	62.6	yes down
				2				
	4			-	1.827219	27.7	64.9	yes down
EN				4				
	11			-	-2.382495	18.9	62.2	yes down
				06				
EX	2	HU08G00	HuME	0.225712	12.5	8.1	yes up	
		932	D32					

In this study, early response genes were identified through the pseudotime heatmap. We conducted additional VIGS experiments to provide preliminary validation of the functions of the selected senescence-related genes (Fig. R9). By silencing the RNA of the identified key senescence gene, *HuCMB1*, through VIGS, we monitored the temporal changes in flavonoid levels in *H. undatus* pericarp after *HuCMB1* silencing. Combining these results with phenotypic changes, we achieved a preliminary validation of the functionality of the *HuCMB1* gene. Additionally, through RT-qPCR experiments, we confirmed that *HuMED32* was significantly regulated by *HuCMB1*, playing a crucial regulatory role in senescence (Fig. R9 and Fig. 5). For a detailed description and analysis of the results, please refer to the main text of the manuscript.

Figure R9 Silencing of *HuCMB1* led to faster senescence of *H. undatus*. (Figure R9 was added as Figure 5 in the manuscript)

A, Expression of four senescence-related genes at different components of pericarp and time points of senescence by RT-qPCR. * represents $p < 0.05$, and ** represents $p < 0.01$; **B**, Schematic showing the domain structure of *HuCMB1*; **C-E**, Changes in fruit phenotype (**C**), weight loss rate (**D**), and flavonoid levels in the exocarp (**E**) after silencing the *HuCMB1* gene. **F**, RT-qPCR was used to analyze the expression of *HuCMB1*, *HuERD6-2*, *HuSAG12*, and *HuMED32* in *H. undatus* after 9 days of storage under *HuCMB1* gene silencing.

Reviewer:

For previous comment 5: More validated RT-qPCR analysis of senescence-related genes has been provided in this version. It would be better if the positive signals could be indicated by arrows in the Figure 3G.

6. The spatial transcriptomics was performed only on one pericarp section. Lacking the biological replicates.

Response: Really appreciated for your kindly comments. Yes, we agree with your opinion. For biological experiments, having repeated samples is highly meaningful. Unfortunately, due to budget constraints, we only prepared a single pericarp slice. In our future work, we will strive to secure more funding support to use repeated samples

for increased data accuracy.

Regarding the data in this study, the spatial transcriptome primarily provided spatial information, while the cellular transcriptome was based on the higher-resolution single-cell transcriptome sequencing data. The interplay between these two transcriptomes mutually reinforced and contributed to more reliable results in this study.

Furthermore, currently, spatial transcriptome data is generally considered stable and reliable. Some published articles utilizing spatial transcriptome technology have also employed single samples. For instance, Fu et al. analyzed the developmental process of maize kernels, using one sample each for 12 DAP and 24 DAP (Fu et al., 2023). Du et al. studied six developmental stages of poplar stem tissues, using one sample for each stage (Du et al., 2023). Li et al. examined the dynamic molecular map of cambium differentiation during primary and secondary growth in trees, using a single sample (Li et al., 2023). Liu et al. investigated the spatiotemporal transcriptome of orchids at different developmental stages, using one sample for each stage (Liu et al., 2022a). Liu et al. explored cell heterogeneity in peanut tissues, using one sample for each site (Liu et al., 2022b). The list of references is as follows:

References

- [1] Fu, Y., et al. Spatial transcriptomics uncover sucrose post-phloem transport during maize kernel development. *Nat. Commun.* **14**, 7191 (2023). doi: 10.1038/s41467-023-43006-7
- [2] Du, J., et al. High-resolution anatomical and spatial transcriptome analyses reveal two types of meristematic cell pools within the secondary vascular tissue of poplar stem. *Molecular Plant* **16**, 809-828 (2023). doi: 10.1016/j.molp.2023.03.005
- [3] Li, R., Wang, Z., Wang, J. W., & Li, L. Combining single-cell RNA sequencing with spatial transcriptome analysis reveals dynamic molecular maps of cambium differentiation in the primary and secondary growth of trees. *Plant Commun.* **4**, 100665 (2023). doi: 10.1016/j.xplc.2023.100665
- [4] Liu, C., et al. A spatiotemporal atlas of organogenesis in the development of orchid flowers. *Nucleic Acids Res.* **50**, 9724-9737 (2022a). doi: 10.1093/nar/gkac773
- [5] Liu, Y., et al. Spatial transcriptome analysis on peanut tissues shed light on cell heterogeneity of the peg. *Plant Biotechnol. J.* **20**, 1648-1650 (2022b). doi: 10.1111/pbi.13884

Reviewer:

For previous comment 6: The spatial transcriptome data in this study is important because it's crucial for the cell type annotation. The authors used only one sample but without detailed information. For example, is the sample used here for the spatial transcriptome CK or Post? Is it proper to use a one-time point sample to annotate all these cell populations in CK and Post samples? The authors should provide more explanations.

7. The discussion part of this manuscript is similar to the result part.

Response: Really appreciated for your comments. Yes, we also identified this issue. We have moved the analytical and discussion content from the Results section to the Discussion section, integrated it with the existing analysis and discussion, and supplemented it with a comparison and analysis of previous research results. The

Discussion section has been extensively rewritten, and the general situation is as follows:

- 1) Deleted the discussion on the specific expression of marker genes in each cluster and moved it to the Discussion section.
- 2) Removed the discussion on core nodes like HuCMB1 and HuMED32 in the GRN and moved it to the Discussion section.
- 3) The introduction to the gene set scoring analysis method has been added, along with corresponding references.
- 4) The analysis approach for the pseudotime section has been explained in more detail, clarifying the complex ideas behind the two pseudotime analyses for better reader comprehension.
- 5) The discussion section now includes additional information on the metabolite content of each component detected by mass spectrometry. This covers differences in metabolites among components and comparative analyses with known metabolites in *H. undatus*.
- 6) Comparative analyses with single-cell analysis results of *Fragaria vesca* infection with *Botrytis cinerea*, as found in the literature, have been incorporated.
- 7) A summary of currently available single-cell sequencing and spatial transcriptomics technologies has been added. The achievements in the plant field using these technologies have been discussed and compared with our work.
- 8) VIGS and RT-qPCR experiments have been included, providing preliminary evidence that HuCMB1 is involved in the fruit senescence process and can regulate flavonoid synthesis.
- 9) Combining the background from previous research and the results of this study, the discussion section now includes a comparative analysis of the impact of flavonoids on senescence.

Reviewer:

For previous comment 7: The discussion section in this version has not been well improved as expected. It is still similar to the result part. Please improve this section and highlight the novel discovery and potential value.

8. The figure legend of Figure 5 is too simple.

Response: The schematic diagram of Figure 5 has been explained in more detail.

Reviewer:

For previous comment 8: The figure legend in Fig.6 is more informative now.

9. The abbreviation that the first time appears should be explained. For example, in line 113, the CK and Post samples should be explained.

Response: The abbreviations have been annotated where they are first used in the manuscript.

Reviewer:

For previous comment 9: No additional questions.

10. In line 89, the authors described that “Through the application of three different algorithms”, while in line 222, the authors described that “Based on the results from the four algorithms”. They are not consistent.

11. In Figure 3E, “Simple” is a wrong label. The authors should double-check the

figures and words throughout the manuscript.

Reviewer #3 (Remarks to the Author):

The authors have made a better version of the manuscript according to the reviews. For my part, I have no further observations except for the change you should make on line 65: change *Fragaria ananassa* to *Fragaria vesca*.

Reviewer #4 (Remarks to the Author):

The authors have addressed almost all my questions, while two new points still puzzle me.

1. As shown in Figure R12A, it was found that HuCMB1 exhibited the highest expression in the ME component, not EX component, while the author demonstrated that "The gene HuCMB1 exhibited the highest expression in the EX component and was highly expressed during the mid and late stages of senescence ($p < 0.01$)."

I don't know if I'm understanding this correctly, or if I'm overlooking or misunderstanding something.

2. Figure R13F: How did the author confirmed that HuMED32 is regulated by HuCMB1? Could the author explain it in detail?

List of Responses

Dear editors and reviewers,

We have made some corrections based on your comments. We have highlighted the changes made in the manuscript by using the track changes mode in Word. The responses are as follows:

REVIEWER COMMENTS

Reviewer #1 (Remarks to the Author):

Your manuscript "Single-Cell and Spatial RNA Sequencing Reveal the Spatiotemporal Trajectories of Fruit Senescence " is recommended for acceptance. All suggestions have been well modified.

Response: Thanks for these comments.

Reviewer #2 (Remarks to the Author):

For previous comment 1: I agreed with the authors' opinion that the annotation results obtained from different algorithms might not have been identical because their calculation principles differed.

In this version, the authors abandoned the MIA method, which is not suitable for the sample size in this study, and the annotation results are clearer now. However, the authors should explain the reasons for categorizing Clusters 4 and 11 as EN cells; obviously, they are highly associated with ENF cells in both the SciBet and Single R methods (Figure 2D).

Response: Yes, it is difficult to completely separate EN and ENF cells. In both the SciBet and SingleR results, clusters 4, 11, and 12 were clustered together. On the UMAP map, clusters 4 and 11 were displayed together, but there was still a small portion of cells in clusters 4 and 11 adjacent to cluster 12. Moreover, it can be seen from the results of the spatial transcriptome that EN and ENF were indeed very close.

From the data in Table R1 below, it can be seen that in both SciBet and SingleR results, the clustering data of cluster 12 was only focused in ENF. Therefore, cluster 12 might be best identified as ENF.

In the violin plot of Figure R1, the marker gene of cluster 12 was only independently expressed in cluster 12. It is difficult to find the marker genes of cluster 4 that were only independently expressed in cluster 4. They were all co-expressed in clusters 4 and 11, and even had a certain expression level in cluster 12. The same applied to the marker genes of cluster 11. In addition to Hu03G02180 being specifically expressed in cluster 11, other marker genes were co-expressed in clusters 11 and 4.

Therefore, when classifying the cell types of each cell cluster, we grouped cluster 4 and cluster 11 together. Due to the clear classification of cluster 12 as ENF, clusters 4 and 11 were classified as the second highest scoring EN components in the results of SciBet and SingleR.

The identification criteria for clusters 4 and 11 belonging to the EN component have been added to the results section (Lines 213-232).

Figure R1 Violin plot displaying the expression of marker genes of clusters 4, 11, and 12 in each cluster (Added as Figure S2F in the manuscript).

Table R1 Scores of clusters 4, 11, and 12 by SingleR and SciBet

		EN	ENF	EX	ME	VB
SingleR	cluster12	0	23	0	0	1
	cluster4	214	900	110	10	10
	cluster11	54	94	51	10	32
SciBet	cluster12	0	23	0	0	1
	cluster4	106	1014	112	3	9
	cluster11	52	112	0	50	19

For previous comment 2: No additional questions.

For previous comment 3: Thanks for the detailed explanation and now I understand the re-clustering analysis of the EX and ME cells.

I have some other questions:

(1) In Figure 3A, in the CK sample, the proportion of EX cells was 64.97%, while the proportion of ME cells was only 10.87%. This statistical result seems to be inconsistent with the microscope images of the CK sample in Figure 3B, in which the ME region is obviously larger than the EX region.

Response: The situation regarding the reliability of sample preparation is as follows. Our samples were taken and sequenced by a professional sample preparation team. Although the technicians of the team were collecting fruit samples for the first time, they had completed a large number of single-cell sequencing experiments on plant samples such as leaves and roots. The paper on animal cell samples previously completed by the team has been published

in journals such as Nature Communications (Zhao et al., 2023; Ma et al., 2023; Shao et al., 2023). Therefore, the sample preparation process should be stable and reliable.

Regarding the differences in the number of cells, there were no literature references available for the single-cell sample preparation of fruit samples. We speculate that the situation is as follows: microscopic photos provide information about the distribution of various types of cells, but it is difficult to determine the exact cell counts. We can see that the ME cells have a large number of thin-walled cells, which occupy a significant amount of space in the CK sample, but the actual number of cells might be lower.

At the same time, we also consider that the proportions of different types of cells are relative values. Due to the healthier condition of the CK sample, a large number of EX cells were preserved. In contrast, the EX cells in the Post sample may be in a poorer state, resulting in only a few EX cells being obtained during the protoplast preparation. Therefore, this may contribute to a lower proportion of ME cells in the CK sample and a lower proportion of EX cells in the Post sample.

(2) In the total of 5788 ME cells, why do 5174 cells (89.39%) originate from the Post sample? In Figure 3B, the ME cell region is even larger in CK than that in Post. Why did the study get such few ME cells from the CK sample?

Could you provide some possible reasons?

Response: Yes, this result is indeed strange. In general, we would speculate that the number of cells in the CK sample should be greater than that in the senescence sample. However, in actual experiments, the number of cells obtained was indeed higher in the Post sample than in the CK sample. There might be several reasons for this:

1. The samples in the Post group were only in a senescence state, not decaying. The cells obtained were only affected by the senescence state, and there were changes in gene expression. It might not lead to a decrease in the number of live cells in the Post group.
2. We mentioned in the results section that “Notably, the pericarp of *H. undatus* is characterized by its high content of lignin and polysaccharides, necessitating the optimization of protocols for protoplast isolation and impurity removal. Due to the fragility of protoplasts, a filtration step was employed to eliminate damaged cellular fragments and organelles, after which the purified protoplasts were loaded into the 10x Genomics Chromium Controller.” In this step, it is possible that due to the higher polysaccharide content in the CK sample, the CK sample obtained fewer live cells than the Post sample.
3. A total of 720.94 Gb Raw Data was obtained through sequencing. Cells with a high proportion of mitochondria and cells containing dual cells in a droplet were filtered out. This description has been added in the Methods section (Line 1017). Perhaps, the data filtering process resulted in CK samples obtaining fewer effective cell numbers than Post samples.

However, since the operating parameters of the samples in the two groups in all processes were consistent, if the difference between the two groups was caused by the above steps, the root cause was still due to the significant difference between the samples themselves. Since single-cell sequencing of fruit samples has not been reported before, these explanations can only be our speculation. If more data on fruits are published in the future, we can further explore whether the differences in cell numbers between samples are consistent with our speculation.

References

- [1] Zhao, W., et al. Single-cell analysis of gastric signet ring cell carcinoma reveals cytological and immune microenvironment features. *Nat Commun.* **14**, 2985 (2023). doi: 10.1038/s41467-023-38426-4.
- [2] Ma, Y., Guo, C., Wang, X., Wei, X., & Ma, J. Impact of chemotherapeutic agents on liver microenvironment: oxaliplatin create a pro-metastatic landscape. *J Exp Clin Cancer Res*, **42**, 237 (2023). doi: 10.1186/s13046-023-02804-z.
- [3] Shao, X. X., et al. The asymmetrical ESR1 signaling in muscle progenitor cells determines the progression of adolescent idiopathic scoliosis. *Cell Discovery*, **9**, 44 (2023). doi: 10.1038/s41421-023-00531-5

For previous comment 4: The pseudotime analysis is usually performed in cell populations with developmental associations. The authors selected EX and ME cells to conduct pseudotime analysis. Whether there are developmental relationship between EX cells and ME cells? it seems that they are two distinct cell types, and they probably have different developmental and senescence processes. Although the CK and Post samples have a time relationship, I think it is not proper to perform the pseudotime analysis of these mixed cell populations, unless the authors could provide much more evidence.

Response: Yes, we agree with your point of view. For in-depth research on animal cells, pseudotime analysis is an effective method for studying the developmental trajectory of cells. Moreover, in plant tissues, in addition to the development of organs such as roots and flowers, there is also a significant differentiation of cellular states. Our attempts and some other studies have shown that pseudotime analysis is also a good means of revealing the direction of cell differentiation (Liu et al., 2022a, 2022b).

Similar to our analysis of different types of cells in fruit peels, researchers have also grouped different types of cells together using pseudotime analysis to explore the differentiation trajectory of cells in tissues. For example, xylem vessel, xylem parenchyma cells and sieve element-companion cells were grouped into different branches and revealed two-directional differentiation trajectories of cambium cells (Liu et al., 2022a). The trajectory of cell states was also used to study the cell differentiation from meristematic cells to vegetative cells (tepals and lip) and reproductive ones (column) using pseudotime analysis (Liu et al., 2022b). This is similar to the three states of cells in our work: mature cells (state 1), resistant cells (state 3), and senescent cells (state 2).

Our research results in fruits showed that although EX and ME are two different cell types and do not have a developmental relationship, as cell senescence progresses, ME and EX cells in CK and Post samples exhibited significant differentiation in the cellular state (Fig. 4C). When attempting to analyze the trajectory of cell differentiation, of course, a single type of cell was the first research objective. However, the results for a single cell type were not conclusive, as neither EX nor ME analysis has shown effective information about the differentiation of cell states in the senescent process.

Perhaps due to the lack of significant development of mature fruit cells after harvesting, single-cell analysis has not yet been conducted. Fruit are different from previously reported plant root and stem cells, and there are significant differences from animal cells. Therefore, based on the methods used by some researchers in the literature to analyze cell differentiation, we conducted more than 20 pseudotime analyses (Figure R2) using different cells and parameters from different perspectives, including analyzing EX cells separately or ME cells separately, and ultimately found the best solution to elucidate the process of fruit senescence. We cautiously chose the current method of analyzing EX and ME together, and the results were the clearest, which can describe the differentiation of EX and ME in the process of fruit senescence.

As stated in the analysis and discussion section, finally, two pseudotime analyses were conducted. The first pseudotime analysis revealed the major functions of target cells, and the second pseudotime analysis, focusing on highly correlated cells with identified functions, elucidated the cell differentiation trajectory.

```
0410_185454-ME_cp_08_subcluster0.1_20230402_145753
0413_065032-EX_cp_08_subcluster0.1_20230413_060412
0413_092116_ME+EX_cp_08_resolution0.8_20230317_123609
0501_205244_Hu_ME+EX_subcluster0.2_20230501_204536
0502_073644_Hu_ME+EX_subcluster0.1_20230502_072228
0517_191826_ME_cp_08_resolution0.8_20230317_123609
0701_081019_EX_cp_08_resolution0.8_20230317_123609
0704_092859_regulated3329_resolution0.1_20230704_085412
0704_115838_regulated3329_resolution0.5_20230704_094414
0706_063317_regulated_EXME_resolution0.1_20230706_062001
0706_090903_ME+EX_cp_08_resolution0.8_20230317_123609
0706_135829_regulated_EXME_resolution0.2_20230706_091132
0706_201425_regulated3329_resolution0.1_20230705_065957
0706_202118_CKPost_senescence3329_resolution0.8_20230706_200637
0706_202210_CK_senescence3329_resolution0.8_20230706_200637
0707_071838_regulated3329_resolution0.5_20230705_070630
0707_140442_regulated_EXME_ROS_resolution0.1_20230707_135114
0707_140506_regulated_EXME_ROS_resolution0.2_20230707_135126
0707_140534_regulated_EXME_ROS_resolution0.8_20230707_135149
0707_182716_regulated3329_ROS_resolution0.1_20230707_175018
0707_182747_regulated3329_ROS_resolution0.5_20230707_175251
0708_074905_regulated_EXME_ROS_4_resolution0.1_20230707_182436
0708_080651_regulated3329_ROS_4_resolution0.1_20230708_074430
0708_135428_regulated_EXME_ROS_4_resolution0.8_20230708_074716
0708_163802_regulated_EXME_ROS_4_resolution0.2_20230707_182443
0709_073155_regulated3329_ROS_4_resolution0.2_20230708_074440
0709_073534_regulated_EXME_ROS_4_resolution0.5_20230707_182456
0709_122418_regulated3329_ROS_4_resolution0.5_20230708_074446
```

Figure R2 List of attempted and successful results of our pseudotime analysis.

References

- [1] Liu, Y. Y., et al. Spatial transcriptome analysis on peanut tissues shed light on cell heterogeneity of the peg. *Plant Biotechnology Journal*, **20**, 1648-1650 (2022a) doi: 10.1111/pbi.13884
- [2] Liu, C., et al. A spatiotemporal atlas of organogenesis in the development of orchid flowers. *Nucleic Acids Research*, **50**, 9724-9737 (2022b) doi: 10.1093/nar/gkac773

For previous comment 5: More validated RT-qPCR analysis of senescence-related genes has been provided in this version. It would be better if the positive signals could be indicated by arrows in the Figure 3G.

Response: The position of DAPI stained cell nuclei and the expression of *HuSAG12* stained by FAM have been indicated by arrows in Figure 3G. The explanation of the arrows has also been added to the legend of Figure 3.

Figure R3 RNA FISH indicated the predominant location of *HuSAG12*. (Shown as Figure 3G in the manuscript).

Yellow arrows indicated the localization of DAPI stained nuclei or the localization of RNA expression stained with FAM probes

For previous comment 6: The spatial transcriptome data in this study is important because it's crucial for the cell type annotation. The authors used only one sample but without detailed information. For example, is the sample used here for the spatial transcriptome CK or Post? Is it proper to use a one-time point sample to annotate all these cell populations in CK and Post samples? The authors should provide more explanations.

Response: We overlooked this crucial information. In our work, CK samples were used for spatial transcriptome analysis. This description has been added to the Methods section (Lines 1092-1093).

The single-cell sequencing results showed that the CK sample contained all 13 cell clusters, and there were no new cell clusters in the Post sample compared to the CK sample, only changes in the expression of genes in each cell cluster. Therefore, the spatial transcriptome data of CK samples can be well combined with the single-cell results of CK samples for analysis. These explanations have also been added to the Results section (Line 163).

For previous comment 7: The discussion section in this version has not been well improved as expected. It is still similar to the result part. Please improve this section and highlight the novel discovery and potential value.

Response: The novel discovery and potential value have been highlighted and summarized in the discussion section, as follows:

Analyzing the temporal trajectory of post-harvest fruit cell differentiation is crucial for advancing research into the mechanisms of post-harvest preservation of fruit. This study found that the senescence processes of fruit exhibited a very clear timeline of ROS-resistance-senescence, and also identified a spatial transition of cell function along the mesocarp-exocarp-mesocarp. Transcription factors of the MADS family such as *HuCMB1* were found to be highly expressed at different stages of senescence and in different types of peel cells, indicating their likely involvement in distinct biological processes of different cell types. While the MADS family has been previously reported to be involved in flower development, this study revealed their participation in fruit resistance and senescence, particularly by exploring the temporal trajectory of *HuCMB1* gene regulation in senescence, providing new evidence for understanding the mechanism of fruit senescence. Thus far, preservation research has primarily focused on macroscopic environmental factors and signaling pathways post-harvest. However, the results of this study suggested that future fruit preservation strategies may involve controlling and utilizing the functionality of specific cell populations to achieve anti-senescence effects. This research will contribute to a better understanding of the roles of different types of fruit cells in the senescence process, thereby promoting more in-depth research into post-harvest fruit senescence mechanisms at the single-cell level. (Lines 947-964)

For previous comment 8: The figure legend in Fig.6 is more informative now.

For previous comment 9: No additional questions.

10. In line 89, the authors described that “Through the application of three different algorithms”, while in line 222, the authors described that “Based on the results from the four algorithms”. They are not consistent.

Response: The error on line 89 has been corrected.

11. In Figure 3E, “Simple” is a wrong label. The authors should double-check the figures and words throughout the manuscript.

Response: We apologize for the labeling error. We have corrected the labeling in Figure 3E. In addition, when we proofread the entire text and images again, we found that there was a deviation in the numerical numbering of the cell clusters in Figure 1B. The markings on the images have been rechecked and Figure 1 has been updated.

Reviewer #3 (Remarks to the Author):

The authors have made a better version of the manuscript according to the reviews. For my part, I have no further observations except for the change you should make on line 65: change *Fragaria ananassa* to *Fragaria vesca*.

Response: The “*Fragaria ananassa*” has been changed to “*Fragaria vesca*”.

Reviewer #4 (Remarks to the Author):

The authors have addressed almost all my questions, while two new points still puzzle me.

1. As shown in Figure R12A, it was found that *HuCMB1* exhibited the highest expression in the ME component, not EX component, while the author demonstrated that “The gene *HuCMB1* exhibited the highest expression in the EX component and was highly expressed during the mid and late stages of senescence ($p < 0.01$).” I don't know if I'm understanding this correctly, or if I'm overlooking or misunderstanding something.

Response: I'm sorry that we mistakenly reported ME as EX in the “Response” file of R1 of the manuscript. The description of the results in the manuscript R1 was correct. The text was as follows: “The RT-qPCR results revealed that the *HuCMB1* gene exhibited specific high expression in the mesocarp and showed elevated expression in the mid and late stages of senescence ($p < 0.01$). (Lines 569-571)”

2. Figure R13F: How did the author confirmed that *HuMED32* is regulated by *HuCMB1*? Could the author explain it in detail?

Response: In the results of single-cell analysis, the top node in the gene regulatory network (GRN) constructed using SCODE for the senescence-related transcription factor network was *HuCMB1*, which was expressed in the early stage of senescence. In the pseudotime heatmap, *HuMED32* emerged as a key gene associated with senescence, also showing early expression during senescence.

Previous studies have shown that *HuCMB1* belongs to the MADS family, which contains MADS (M) domain and keratin-like (K) domain (Henschel et al., 2002; Kaufmann et al., 2005). We proposed a hypothesis that *HuCMB1* might interact with target gene promoter sequences through the MADS domain (Santelli and Richmond, 2000), and bind to the MED complex through the K domain (Henschel et al., 2002; Kaufmann et al., 2005; Yang et al., 2003), acting as an enhancer to influence the expression of senescence-related genes.

On the one hand, we identified the expression trends of *HuCMB1* and *HuMED32* in various cellular components during the senescence process through RT-qPCR experiments. The results showed that the expression changes of *HuCMB1* and *HuMED32* were similar in the mesocarp (Figure R4A). Additionally, we observed high expression of *HuMED32* in the late stages of senescence in the exocarp, which may be due to the involvement of *HuMED32* in another senescence -related regulatory mechanism specific to the exocarp.

On the other hand, we constructed an RNA silencing line of *HuCMB1* and found that after *HuCMB1* silencing, *HuMED32* was also negatively regulated, but the amplitude was not significant (Figure R4F). We speculated that the MED complex is the foundation, and after

binding to this complex, HuCMB1 can act as an enhancer for MED complex mediated Pol II regulation of target gene expression. When HuCMB1 regulates the expression of target genes, it is likely to require the involvement of MED complex including the key unit HuMED32.

Figure R4 Silencing of *HuCMB1* led to faster senescence of *H. undatus*. (Shown as Figure 5 in the manuscript)

A, Expression of four senescence-related genes at different components of pericarp and time points of senescence. * represents $p < 0.05$, and ** represents $p < 0.01$; **B**, Schematic showing the domain structure of *HuCMB1* gene; **C-E**, Changes in fruit phenotype (**C**), weight loss rate (**D**), and flavonoid levels in the exocarp (**E**) after silencing the *HuCMB1* gene. **F**, RT-qPCR was used to analyze the expression of *HuCMB1*, *HuERD6-2*, *HuSAG12*, and *HuMED32* in *H. undatus* after 9 days of storage under *HuCMB1* gene silencing.

References

- [1] Henschel K, Kofuji R, Hasebe M, Saedler H, Munster T, Theissen G: Two ancient classes of MIKC-type MADS-box genes are present in the moss *Physcomitrella patens*. *Mol Biol Evol* 2002, 19:801-814
- [2] Kaufmann K, Melzer R, Theissen G: MIKC-type MADS-domain proteins: structural modularity, protein interactions and network evolution in land plants. *Gene* 2005, 347:183-198
- [3] Santelli E, Richmond TJ: Crystal structure of MEF2A core bound to DNA at 1.5 angstrom resolution. *J Mol Biol* 2000, 297:437-449.
- [4] Yang Y, Fanning L, Jack T: The K domain mediates heterodimerization of the Arabidopsis floral organ identity proteins, *APETALA3* and *PISTILLATA*. *Plant J* 2003, 33:47-59

If you have any questions on this manuscript, please no hesitate to contact me. Thank you very much!

Prof. Robert Henry

Reviewer #2 (Remarks to the Author):

The concerns were well addressed, and there are no additional comments.

Reviewer #4 (Remarks to the Author):

The article "Single-Cell and Spatial RNA Sequencing Reveal the Spatiotemporal Trajectories of Fruit Senescence " is recommended for acceptance.